# On the Necessity of Collaboration for Online Model Selection with Decentralized Data

**Junfan Li**[1]  **Zheshun Wu**[1]  **Zenglin Xu**[2,3,4*]  **Irwin King**[5]

[1]School of Computer Science and Technology, Harbin Institute of Technology, Shenzhen
[2]Pengcheng Lab
[3]Artificial Intelligence Innovation and Incubation (AI[3]) Institute, Fudan University
[4]Shanghai Academy of AI for Science
[5]Department of Computer Science and Engineering, The Chinese University of Hong Kong

lijunfan@hit.edu.cn
wuzhsh23@gmail.com
zenglin@gmail.com
king@cse.cuhk.edu.hk

## Abstract

We consider online model selection with decentralized data over $M$ clients, and study the necessity of collaboration among clients. Previous work proposed various federated algorithms without demonstrating their necessity, while we answer the question from a novel perspective of computational constraints. We prove lower bounds on the regret, and propose a federated algorithm and analyze the upper bound. Our results show (i) collaboration is unnecessary in the absence of computational constraints on clients; (ii) collaboration is necessary if the computational cost on each client is limited to $o(K)$, where $K$ is the number of candidate hypothesis spaces. We clarify the unnecessary nature of collaboration in previous federated algorithms for distributed online multi-kernel learning, and improve the regret bounds at a smaller computational and communication cost. Our algorithm relies on three new techniques including an improved Bernstein's inequality for martingale, a federated online mirror descent framework, and decoupling model selection and prediction, which might be of independent interest.

## 1   Introduction

Model selection which is a fundamental problem for offline machine learning focuses on how to select a suitable hypothesis space for a machine learning algorithm [Mitchell, 1997, Bartlett et al., 2002, Mohri et al., 2018]. Model selection for online machine learning is called online model selection (OMS), such as model selection for online supervised learning [Foster et al., 2017, Zhang and Liao, 2018, Zhang et al., 2021, Li and Liao, 2022], model selection for online active learning [Karimi et al., 2021], and model selection for contextual bandits [Foster et al., 2019, Pacchiano et al., 2020, Ghosh and Chowdhury, 2022]. We consider model selection for online supervised learning. Let $\mathcal{F} = \{\mathcal{F}_1, \ldots, \mathcal{F}_K\}$ contain $K$ hypothesis spaces and $\ell(\cdot, \cdot)$ be a loss function. For a sequence of examples $\{(\mathbf{x}_t, y_t)\}_{t=1,\ldots,T}$, we aim to adapt to the case that the optimal hypothesis space $\mathcal{F}_{i^*} \in \mathcal{F}$ is given by an oracle and we run an online learning algorithm in $\mathcal{F}_{i^*}$. OMS can be defined by minimizing the *regret*, i.e.,

$$\min_{f_1, \ldots, f_T} \left( \sum_{t=1}^{T} \ell(f_t(\mathbf{x}_t), y_t) - \min_{f \in \mathcal{F}_{i^*}} \sum_{t=1}^{T} \ell(f(\mathbf{x}_t), y_t) \right),$$

---

*Corresponding author.

38th Conference on Neural Information Processing Systems (NeurIPS 2024).

where $f_t \in \cup_{i=1}^{K} \mathcal{F}_i$ is the hypothesis used by an OMS algorithm at the $t$-th round. The optimal value of the regret depends on the complexity of $\mathcal{F}_{i^*}$ [Foster et al., 2017, 2019].

In this work, we consider online model selection with decentralized data (OMS-DecD) over $M$ clients, in which each client observes a sequence of examples $\left\{ \left( \mathbf{x}_t^{(j)}, y_t^{(j)} \right) \right\}_{t=1,\ldots,T}, j = 1, \ldots, M,$ and but does not share personalized data with others. There is a central server that coordinates the clients by sharing personalized models or gradients [Konečný et al., 2016, Kairouz et al., 2021, Zeng et al., 2023a,b]. OMS-DecD captures some real-world applications where the data may be collected by sensors on $M$ different remote devices or mobile phones [Li et al., 2020, Patel et al., 2023, Kwon et al., 2023], or a local device can not store all of data due to low storage and thus it is necessary to store the data on more local devices [Slavakis et al., 2014, Bouboulis et al., 2018]. OMS-DecD can be defined by minimizing the following regret,

$$\min_{f_t^{(j)}, t=1,\ldots,T, j=1,\ldots,M} \left( \sum_{j=1}^{M} \sum_{t=1}^{T} \ell \left( f_t^{(j)} \left( \mathbf{x}_t^{(j)} \right), y_t^{(j)} \right) - \min_{f \in \mathcal{F}_{i^*}} \sum_{j=1}^{M} \sum_{t=1}^{T} \ell \left( f \left( \mathbf{x}_t^{(j)} \right), y_t^{(j)} \right) \right),$$

where $f_t^{(j)} \in \cup_{i=1}^{K} \mathcal{F}_i$ is the hypothesis adopted by the $j$-th client at the $t$-th round. Solving OMS-DecD must achieve two goals: **G**1 minimizing the regret, and **G**2 providing privacy protection.

A trivial approach is to use a *noncooperative algorithm* that independently runs a copy of an OMS algorithm on the $M$ clients. It naturally provides strong privacy protection, that is, it achieves **G**2, but suffers a regret bound that increases linearly with $M$. It is unknown whether it achieves **G**1. Another approach is *federated learning* which is a framework of cooperative learning with privacy protection and is provably effective in stochastic convex optimization [McMahan et al., 2017, Woodworth et al., 2020b, Wang et al., 2021, Reddi et al., 2021]. It is natural to ask:

**Question 1.** *Whether collaboration is effective for OMS-DecD.*

The question reveals the hardness of OMS-DecD and is helpful to understand the limitations of federated learning. Previous work studied a special instance of OMS-DecD called distributed online multi-kernel learning (OMKL) where $\mathcal{F}_i$ is a reproducing kernel Hilbert space (RKHS), and proposed three federated OMKL algorithms including vM-KOFL, eM-KOFL [Hong and Chae, 2022] and POF-MKL [Ghari and Shen, 2022]. The three algorithms also suffer regret bounds that increase linearly with $M$, and thus can not answer the question. If $K = 1$, then OMS-DecD is equivalent to distributed online learning [Mitra et al., 2021, Kwon et al., 2023, Patel et al., 2023]. A noncooperative algorithm that independently runs online gradient descent (OGD) on each client achieves the two goals simultaneously [Patel et al., 2023]. Collaboration is unnecessary in the case of $K = 1$.

In summary, previous work can not answer the question well. On one hand, previous work can not answer the question in the case of $K > 1$. On the other hand, in the case of $K = 1$, previous work has answered the question only using the statistical property of algorithms, i.e., the worst-case regret, but omitted the computational property which is very important for real-world applications.

## 1.1 Main Results

In this paper, we will answer the question from a new perspective of computational constraints on the problem (Section 5.5). Our main results are as follows.

(1) **An upper bound on the regret.** We propose a federated algorithm, FOMD-OMS, and prove an upper bound on the regret (Theorem 2). Besides, if $\mathcal{F}_1, ..., \mathcal{F}_K$ are RKHSs, then our algorithm improves the regret bounds of FOP-MKL [Ghari and Shen, 2022] and eM-KOFL [Hong and Chae, 2022] at a smaller computational and communication cost. Table 1 summarizes the results.

(2) **Lower bounds on the regret.** We separately prove a lower bound on the regret of any (possibly cooperative) algorithm and any noncooperative algorithm (Theorem 3).

(3) **A new perspective of computational constraints for Question 1.** By the upper bound and lower bounds, we conclude that (i) collaboration is unnecessary when there are no computational constraints on clients, thereby generalizing the result for distributed online learning, i.e., $K = 1$; (ii) collaboration is necessary if the computational cost on each client is limited to $o(K)$ where irrelevant parameters are omitted. Our results clarify the unnecessary nature of collaboration in previous federated algorithms for distributed OMKL.

## 1.2 Technical Challenges

There are two main technical challenges on designing a federated online model selection algorithm.

The first challenge lies in obtaining high-probability regret bounds that adapt to the complexity of individual hypothesis space, a fundamental problem in online model selection [Foster et al., 2017]. While acquiring expected regret bounds that adapt to the complexity of individual hypothesis spaces is straightforward, the crux is to derive high-probability bounds from expected bounds. To this end, we introduce a new Bernstein's inequality for martingale (Lemma 1), which might be of independent interest.

The second challenge involves achieving a per-round communication cost of $o(K)$. To tackle this challenge, we propose two techniques: (i) decoupling model selection and prediction; (ii) an algorithmic framework, named FOMD-No-LU, which might be of independent interest. Specifically, when clients execute model selection, server must broadcast an aggregated probability distribution, denoted by $\mathbf{p} \in \mathbb{R}^K$, to clients, naturally incurring a $O(K)$ download cost. Our algorithm conducts model selection on server and makes predictions on clients, thereby eliminating the need to broadcast the aggregated probability distribution to clients. Additionally, if we use the local updating approach [Mitra et al., 2021, Patel et al., 2023], then server must broadcast $K$ aggregated models to clients, also resulting in a $O(K)$ download cost [Ghari and Shen, 2022]. By utilizing FOMD-No-LU, our algorithm only broadcasts the selected models to clients and can achieve a $o(K)$ download cost.

## 2 Related work

Previous work has studied the necessity of collaboration for distributed bandit convex optimization [Patel et al., 2023], where a federated algorithmic framework named FEDPOSGD was proposed. Although the regret bounds of FEDPOSGD are smaller than some noncooperative algorithms, there is not a lower bound on the regret of any noncooperative algorithm [Patel et al., 2023]. Moreover, the regret analysis of FEDPOSGD is based on the analysis for federated online gradient descent that is not applicable to our algorithm, FOMD-OMS. The regret analysis of FOMD-OMS requires the analysis for federated online mirror descent with negative entropy regularizer.

Our work is also different from federated bandits, such as federated $K$-armed bandits [Wang et al., 2020] and federated linear contextual bandits [Huang et al., 2021]. For OMS-DecD, we do not assume that the examples $(\mathbf{x}_t^{(j)}, y_t^{(j)})$, $t = 1, \ldots, T$, on each client are independent and identically distributed (i.i.d.). In contrast, in both federated $K$-armed bandits or federated linear contextual bandits, the rewards must be i.i.d., thereby making collaboration effective. This is similar to the approach used in federated stochastic optimization. However, this may not hold true for OMS-DedD. Therefore, it is a distinctive problem for OMS-DecD to study whether collaboration is effective.

## 3 Problem Setting

**Notations** Let $\mathcal{X} = \{\mathbf{x} \in \mathbb{R}^d | \|\mathbf{x}\|_2 < \infty\}$ be an instance space, $\mathcal{Y} = \{y \in \mathbb{R} : |y| < \infty\}$ be an output space, and $\{(\mathbf{x}_t, y_t)\}_{t \in [T]}$ be a sequence of examples, where $[T] = \{1, \ldots, T\}$, $\mathbf{x}_t \in \mathcal{X}$ and $y_t \in \mathcal{Y}$. Let $S = \{s_1, s_2, \ldots\}$ be a finite set, $\mathrm{Uni}(S)$ be the uniform distribution over the elements in $S$ and $s_{[T]}$ be the abbreviation of the sequence $s_1, s_2, \ldots, s_T$. Denote by $\mathbb{P}[A]$ the probability that an event $A$ occurs, $a \wedge b = \min\{a, b\}$, $a \vee b = \max\{a, b\}$ and $\log(a) = \log_2(a)$. Let $\psi_t(\cdot) : \Omega \to \mathbb{R}, t \in [T]$ be a sequence of time-variant strongly convex regularizers defined on a domain $\Omega$. The Bregman divergence denoted by $\mathcal{D}_{\psi_t}(\cdot, \cdot)$, associated with $\psi_t(\cdot)$ is defined by

$$\forall \mathbf{u}, \mathbf{v} \in \Omega, \quad \mathcal{D}_{\psi_t}(\mathbf{u}, \mathbf{v}) = \psi_t(\mathbf{u}) - \psi_t(\mathbf{v}) - \langle \nabla \psi_t(\mathbf{v}), \mathbf{u} - \mathbf{v} \rangle.$$

### 3.1 Online Model Selection (OMS)

Let $\mathcal{F} = \{\mathcal{F}_1, \ldots, \mathcal{F}_K\}$ contain $K$ hypothesis spaces where

$$\mathcal{F}_i = \left\{ f(\mathbf{x}) = \mathbf{w}^\top \phi_i(\mathbf{x}) : \phi_i(\mathbf{x}) \in \mathbb{R}^{d_i}, \|\mathbf{w}\|_2 \le U_i \right\}. \tag{1}$$

Let $\mathcal{F}_{i^*} \in \mathcal{F}$ be the optimal but unknown hypothesis space for a given $\{(\mathbf{x}_t, y_t)\}_{t \in [T]}$. OMS can be defined as follows: generating a sequence of hypotheses $f_{[T]}$ that minimizes the following *regret*,

$$\forall i \in [K], \quad \text{Reg}(\mathcal{F}_i) = \sum_{t=1}^{T} \ell(f_t(\mathbf{x}_t), y_t) - \min_{f \in \mathcal{F}_i} \sum_{t=1}^{T} \ell(f(\mathbf{x}_t), y_t),$$

where $f_t \in \cup_{i=1}^{K} \mathcal{F}_i$. The optimal hypothesis space $\mathcal{F}_{i^*}$ must contain a good hypothesis and has a low complexity [Foster et al., 2017, 2019], and is defined by

$$\mathcal{F}_{i^*} = \arg\min_{\mathcal{F}_i \in \mathcal{F}} \left[ \min_{f \in \mathcal{F}_i} \sum_{t=1}^{T} \ell(f(\mathbf{x}_t), y_t) + \Theta\left(\sqrt{T \cdot \mathfrak{C}_i}\right) \right],$$

where $\mathfrak{C}_i$ measures the complexity of $\mathcal{F}_i$, such as $U_i$ and $d_i$.

OMS is more challenge than online learning, since we not only learn the optimal hypothesis space, but also learn the optimal hypothesis in the space. Next we give some examples of OMS.

**Example 1** (Online Hyper-parameters Tuning). *Let $\mathcal{F}_i$ consist of linear functions of the form*

$$\mathcal{F}_i = \{f(\mathbf{x}) = \langle \mathbf{w}, \mathbf{x} \rangle, \|\mathbf{w}\|_2 \leq U_i\},$$

*where $U_i > 0$ is a regularization parameter. Let $\mathcal{U} = \{U_i, i \in [K] : U_1 < U_2 < \ldots < U_K\}$. The hypothesis spaces are nested, i.e., $\mathcal{F}_1 \subseteq \mathcal{F}_2 \subseteq \ldots \subseteq \mathcal{F}_K$. The optimal regularization parameter $U_{i^*} \in \mathcal{U}$ corresponds to the optimal hypothesis space $\mathcal{F}_{i^*} \in \mathcal{F}$.*

**Example 2** (Online Kernel Selection [Shen et al., 2019, Li and Liao, 2022]). *Let $\kappa_i(\cdot, \cdot) : \mathbb{R}^d \times \mathbb{R}^d \to \mathbb{R}$ be a positive semidefinite kernel function, and $\phi_i : \mathbb{R}^d \to \mathbb{R}^{d_i}$ be the associated feature mapping. $\mathcal{F}_i$ is the RKHS associated with $\kappa_i$, i.e.,*

$$\mathcal{F}_i = \{f(\mathbf{x}) = \langle \mathbf{w}, \phi_i(\mathbf{x}) \rangle : \|\mathbf{w}\|_2 \leq U_i\}.$$

*The optimal kernel function $\kappa_{i^*} \in \{\kappa_1, \ldots, \kappa_K\}$ corresponds to the optimal RKHS $\mathcal{F}_{i^*} \in \mathcal{F}$.*

**Example 3** (Online Pre-trained Classifier Selection [Karimi et al., 2021]). *Generally, $\mathcal{F}_i$ can be a well-trained machine learning model. Let $\mathcal{F}$ contain $K$ pre-trained classifiers. For a new instance $\mathbf{x}_t$, we select a (combinational) pre-trained classifier and make a prediction. The selection of a pre-trained classifier has an important implication in practical scenarios.*

### 3.2 Online Model Selection with Decentralized Data (OMS-DecD)

We formally define OMS-DecD as follows. Assuming that there are $M$ clients and a server. At any round $t$, each client observes an instance $\mathbf{x}_t^{(j)}$, and selects a hypothesis $f_t^{(j)} \in \cup_{i=1}^{K} \mathcal{F}_i$, $j \in [M]$. Then clients output predictions $\{f_t^{(j)}(\mathbf{x}_t^{(j)})\}_{j=1}^{M}$. The goal is to minimize the following regret

$$\forall i \in [K], \quad \text{Reg}_D(\mathcal{F}_i) = \sum_{t=1}^{T} \sum_{j=1}^{M} \ell\left(f_t^{(j)}(\mathbf{x}_t^{(j)}), y_t^{(j)}\right) - \min_{f \in \mathcal{F}_i} \sum_{t=1}^{T} \sum_{j=1}^{M} \ell\left(f(\mathbf{x}_t^{(j)}), y_t^{(j)}\right),$$

where $y_t^{(j)}$ is the label or true output. Each client can not share personalized data with others, but can share personalized models or gradients via the central server.

## 4 FOMD-No-LU

In this section, we propose a federated algorithmic framework, FOMD-No-LU (Federated Online Mirror Descent without Local Updating) for online collaboration.

Let $\Omega$ be a convex and bounded decision set. At any round $t$, each client $j \in [M]$ selects a decision $\mathbf{u}_t^{(j)} \in \Omega$, and then observes a loss function $l_t^{(j)}(\cdot) : \Omega \to \mathbb{R}$. The client computes the loss $l_t^{(j)}(\mathbf{u}_t^{(j)})$ and an estimator of the gradient denoted by $\tilde{g}_t^{(j)}$ (or the gradient denoted by $g_t^{(j)}$). To reduce the communication cost, we adopt the intermittent communication (IC) protocol [Woodworth et al., 2021] where clients communicate with server every $N$ rounds. Assuming that $T = N \times R$ where $N, R \in \mathbb{Z}$, the IC protocol limits the rounds of communication to $R$.

We divide $[T]$ into $R$ disjoint sub-intervals denoted by $\{T_r\}_{r=1}^R$, in which

$$T_r = \{(r-1)N + 1, (r-1)N + 2, \dots, rN\}. \tag{2}$$

For any $t \in T_r$, all clients always select the initial decision, i.e.,

$$\forall j \in [M], \ \forall t \in T_r, \quad \mathbf{u}_t^{(j)} = \mathbf{u}_{(r-1)N+1}^{(j)}. \tag{3}$$

At the end of the $rN$-round, all of clients send $\frac{1}{N} \sum_{t \in T_r} \tilde{g}_t^{(j)}, j \in [M]$ to server. Then server updates the decision within online mirror descent framework [Bubeck and Cesa-Bianchi, 2012],

$$\bar{g}_t = \frac{1}{M} \sum_{j=1}^M \left( \frac{1}{N} \sum_{t \in T_r} \tilde{g}_t^{(j)} \right), \tag{4}$$

$$\nabla_{\bar{\mathbf{u}}_{t+1}} \psi_t(\bar{\mathbf{u}}_{t+1}) = \nabla_{\mathbf{u}_t} \psi_t(\mathbf{u}_t) - \bar{g}_t, \tag{5}$$

$$\mathbf{u}_{t+1} = \arg\min_{\mathbf{u} \in \Omega} \mathcal{D}_{\psi_t}(\mathbf{u}, \bar{\mathbf{u}}_{t+1}). \tag{6}$$

(4)-(5) is called model averaging [McMahan et al., 2017] and shows the collaboration among clients. Finally, server may broadcast $\mathbf{u}_{t+1}$ to all clients. Let the initial decision $\mathbf{u}_1^{(j)} = \mathbf{u}_1$ for all $j \in [M]$, then it must be $\mathbf{u}_t^{(j)} = \mathbf{u}_t$ for all $t \in [T]$. Thus clients do not transmit $\mathbf{u}_t^{(j)}$ to server. The pseudo-code of FOMD-No-LU is shown in Algorithm 1.

## 4.1 Regret Bound

**Theorem 1.** *Let* $\mathcal{E} = \{N, 2N, \dots, RN\}$ *where* $N = \frac{T}{R}$ *and* $R \in [T]$. *Assuming that* $l_t^{(j)}(\cdot)$, $t \in [T], j \in [M]$, *are convex loss functions. Let* $g_t^{(j)} = \nabla_{\mathbf{u}_t^{(j)}} l_t^{(j)}(\mathbf{u}_t^{(j)})$ *and* $\tilde{g}_t^{(j)}$ *be an estimator of* $g_t^{(j)}$. *At any round* $t \in \mathcal{E}$, *let* $\mathbf{q}_{t+1}$ *and* $\mathbf{r}_{t+1}$ *be two auxiliary decisions defined as follows,*

$$\nabla_{\mathbf{q}_{t+1}} \psi_t(\mathbf{q}_{t+1}) = \nabla_{\mathbf{u}_t} \psi_t(\mathbf{u}_t) - 2 \sum_{j=1}^M \frac{\tilde{g}_t^{(j)} - g_t^{(j)}}{M}, \quad \nabla_{\mathbf{r}_{t+1}} \psi_t(\mathbf{r}_{t+1}) = \nabla_{\mathbf{u}_t} \psi_t(\mathbf{u}_t) - \frac{2}{M} \sum_{j=1}^M g_t^{(j)}.$$

*Then FOMD-No-LU guarantees that,*

$$\forall \mathbf{v} \in \Omega, \quad \sum_{t=1}^T \sum_{j=1}^M \frac{l_t^{(j)}(\mathbf{u}_t^{(j)}) - l_t^{(j)}(\mathbf{v})}{NM} \le \underbrace{\sum_{t \in \mathcal{E}} \left[ \mathcal{D}_{\psi_t}(\mathbf{v}, \mathbf{u}_t) - \mathcal{D}_{\psi_t}(\mathbf{v}, \mathbf{u}_{t+1}) + \frac{\mathcal{D}_{\psi_t}(\mathbf{u}_t, \mathbf{r}_{t+1})}{2} \right]}_{\Xi_1} +$$

$$\underbrace{\sum_{t \in \mathcal{E}} \frac{\mathcal{D}_{\psi_t}(\mathbf{u}_t, \mathbf{q}_{t+1})}{2} + \frac{1}{M} \sum_{t \in \mathcal{E}} \sum_{j=1}^M \left\langle \tilde{g}_t^{(j)} - g_t^{(j)}, \mathbf{u}_t - \mathbf{v} \right\rangle}_{\Xi_2}.$$

It is intriguing that the regret bound comprises two components: the first part, $\Xi_1$, cannot be reduced by collaboration, while the second part, $\Xi_2$, highlights the benefits of collaboration. $\Xi_1$ is the regret induced by exact gradients, while $\Xi_2$ is the regret induced by estimated gradients and shows how collaboration controls the regret. It is worth mentioning that Theorem 1 gives a general regret bound, from which various types of regret bounds can be readily derived by instantiating the decision set $\Omega$ and the regularizer $\psi_t(\cdot)$. For instance, if $\Omega = \mathcal{F}_i$ where $\mathcal{F}_i$ follows Example 1, $\psi_t(\mathbf{v}) = \frac{1}{2\lambda} \|\mathbf{v}\|_2^2$ and $\mathbb{E}[\|\tilde{g}_t^{(j)}\|_2^2] \le C \|g_t^{(j)}\|_2^2$, then FOMD-No-LU becomes a federated online descent descent. It is easy to give a $O(MU_i \sqrt{(1 + \frac{C}{M})T})$ expected regret from Theorem 1. Besides, $N > 1$ increases the regret and shows the trade-off between communication cost and regret bound.

Theorem 1 requires a novel analysis on how the bias of estimators, i.e., $\sum_{j=1}^M \|\tilde{g}_t^{(j)} - g_t^{(j)}\|_2^2$, is controlled by cooperation. To this end, we introduce two virtual decisions $\mathbf{q}_{t+1}$ and $\mathbf{r}_{t+1}$ that are updated by $2 \sum_{j=1}^M \frac{\tilde{g}_t^{(j)} - g_t^{(j)}}{M}$ and $2 \sum_{j=1}^M \frac{g_t^{(j)}}{M}$, respectively. Previous federated online mirror descent uses exact gradients $g_t^{(j)}, j \in [M]$ [Mitra et al., 2021]. Thus its analysis is different from ours.

| **Algorithm 1** FOMD-No-LU | **Algorithm 2** FOMD-OMS ($R = T$) |
|---|---|
| **Require:** $\Omega$. | **Require:** $T, J, \eta_1, \{U_i, \lambda_{1,i}, i \in [K]\}$ |
| **Ensure:** $\mathbf{u}_1^{(j)}, j \in [M]$ | **Ensure:** $f_{1,i}^{(j)} = 0, p_{1,i}, i \in [K], j \in [M]$ |
| 1: **for** $r = 1, 2, \ldots, R$ **do** | 1: **for** $t = 1, 2, \ldots, T$ **do** |
| 2:   **for** $t = (r-1)N+1, \ldots, rN$ **do** | 2:   **for** $j = 1, \ldots, M$ **do** |
| 3:      **for** $j = 1, \ldots, M$ in parallel **do** | 3:      Server samples $O_t^{(j)}$ following (7) |
| 4:         Client selects $\mathbf{u}_{(r-1)N+1}^{(j)} \in \Omega$ | 4:      Server broadcasts $f_{t,i}^{(j)}, i \in O_t^{(j)}$ to client |
| 5:         Client observes loss function $l_t^{(j)}(\cdot)$ | 5:   **end for** |
| 6:         Client computes estimated gradient $\tilde{g}_t^{(j)}$ | 6:   **for** $j = 1, \ldots, M$ in parallel **do** |
| 7:      **end for** | 7:      Client outputs $f_{t,A_{t,1}}^{(j)}(\mathbf{x}_t^{(j)})$ |
| 8:      **if** $t == rN$ **then** | 8:      Client computes $\nabla_{t,i}^{(j)}, c_{t,i}^{(j)}, i \in O_t^{(j)}$ |
| 9:         Clients transmit $\frac{1}{N}\sum_{t \in T_r} \tilde{g}_t^{(j)}, j \in [M]$ | 9:      Client transmits $\nabla_{t,i}^{(j)}, c_{t,i}^{(j)}, i \in O_t^{(j)}$ |
| 10:        Server computes $\mathbf{u}_{t+1}$ following (4)-(6) | 10:  **end for** |
| 11:        Server may broadcast $\mathbf{u}_{t+1}$ | 11:  Server computes $\mathbf{p}_{t+1}$ following (8) |
| 12:      **end if** | 12:  Server computes $\mathbf{w}_{t+1,i}, i \in [K]$ following (9) |
| 13:   **end for** | 13: **end for** |
| 14: **end for** | |

## 4.2 Comparison with Previous Work

In fact, FOMD-No-LU adopts the batching technique [Dekel et al., 2011], that is, it divides $[T]$ into $R$ sub-intervals and executes (3) during each sub-intervals. The batching technique (also known as mini-batch) has been used in the multi-armed bandit problem [Arora et al., 2012] and distributed stochastic convex optimization [Karimireddy et al., 2020, Woodworth et al., 2020a]. We use the batching technique for the first time to distributed online learning.

FOMD-No-LU is different from FedOMD (federated online mirror descent) [Mitra et al., 2021]. (i) FedOMD only transmits exact gradients, while FOMD-No-LU can transmit estimated gradients. Thus the regret bound of FedOMD did not contain $\Xi_2$ in Theorem 1. (ii) FedOMD uses local updating, such as local OGD [Patel et al., 2023] and local SGD [McMahan et al., 2017, Reddi et al., 2021]. Thus FedOMD induces client drift, i.e., $\mathbf{u}_t^{(j)} \neq \mathbf{u}_t$. Besides, if we use FedOMD to design a federated online model selection algorithm, then the download cost is in $O(MK)$ bits.

# 5 A Federated Algorithm for OMS-DecD

In this section, we just consider the case $R = T$, that is, there is no communication constraints. Due to space constraints, we have deferred the algorithm and result of $R < T$ to the appendix.

At a high level, our algorithm comprises two components both of which are critical for achieving a communication cost in $o(K)$: (i) decoupling model selection and online prediction; (ii) collaboratively updating decisions within the framework of FOMD-No-LU.

## 5.1 Decoupling Model Selection and Prediction

**Model Selection on Server** At any round $t$, server maintains $K$ hypotheses $\{f_{t,i}^{(j)} \in \mathcal{F}_i\}_{i=1}^K$ and a probability distribution $\mathbf{p}_t^{(j)}$ over the $K$ hypotheses for all $j \in [M]$. The model selection process aims to select a hypothesis from $\{f_{t,i}^{(j)}\}_{i=1}^K$ and then predicts the output of $\mathbf{x}_t^{(j)}$. An intuitive idea is that, for each $j \in [M]$, the client samples a hypothesis following $\mathbf{p}_t^{(j)}$. However, such an approach requires that server broadcasts $\mathbf{p}_t^{(j)}$ to clients, and will cause a download cost in $O(K)$.

The sampling operation (or model selection process) can be executed on server. Specifically, server just broadcasts the selected hypotheses, and thus saves the communication cost. For each $j \in [M]$, server selects $J \in [2, K]$ hypotheses denoted by $f_{t,A_{t,a}}^{(j)}, a \in [J]$ where $A_{t,a} \in [K]$. For simplicity, let $O_t^{(j)} = \{A_{t,1}, \ldots, A_{t,J}\}$. We instantiate $\mathbf{u}_t = \mathbf{p}_t$ in FOMD-No-LU. Then FOMD-No-LU ensures

$\mathbf{p}_t^{(j)} = \mathbf{p}_t$ for all $j \in [M]$. We sample $A_{t,1}, \ldots, A_{t,J}$ in order and follow (7).

$$\begin{cases} A_{t,1} \sim \mathbf{p}_t, \\ A_{t,a} \sim \mathrm{Uni}([K] \setminus \{A_{t,1}, \ldots, A_{t,a-1}\}), \ a \in [2, J]. \end{cases} \tag{7}$$

Server samples $O_t^{(j)}$ for all $j \in [M]$ and thus must independently execute (7) $M$ times which only pays an additional computational cost in $O(M \log K)$. The factor $\log K$ arises from the process of sampling a number from $\{1, ..., K\}$. Server only sends $f_{A_{t,a}}^{(j)}$, $a \in [J]$ to the $j$-th client. It is worth mentioning that server does not send $\mathbf{p}_t$. The total download cost is $O(\sum_{j=1}^{M} \sum_{a=1}^{J} (d_{A_{t,a}} + \log K))$. If $J$ is independent of $K$, then the download cost is only $O(M \log K)$.

**Prediction on Clients** For each $j \in [M]$, the $j$-th client receives $f_{A_{t,a}}^{(j)}$, $a \in [J]$, and uses $f_{t,A_{t,1}}^{(j)}$ to output a prediction, i.e.,

$$\hat{y}_t^{(j)} = f_{t,A_{t,1}}^{(j)} \left( \mathbf{x}_t^{(j)} \right) = \left\langle \mathbf{w}_{t,A_{t,1}}^{(j)}, \phi_{A_{t,1}}(\mathbf{x}_t^{(j)}) \right\rangle,$$

where we assume that $f_{t,i}^{(j)}$ is parameterized by $\mathbf{w}_{t,i}^{(j)} \in \mathbb{R}^{d_i}$ (see (1)). After observing the true output $y_t^{(j)}$, the client suffers a loss $\ell(f_{t,A_{t,1}}^{(j)}(\mathbf{x}_t^{(j)}), y_t^{(j)})$.

It is worth mentioning that the other $J - 1$ hypotheses $f_{t,A_{t,a}}^{(j)}$, $a \geq 2$ are just used to obtain more information on the loss function. We will explain more in the following subsection. Thus we do not cumulate the loss $\ell(f_{t,A_{t,a}}^{(j)}(\mathbf{x}_t^{(j)}), y_t^{(j)})$, $a \geq 2$.

### 5.2 Online Collaboration Updating

**Updating sampling probabilities** For each $j \in [M]$, let $\mathbf{c}_t^{(j)} = (c_{t,1}^{(j)}, \ldots, c_{t,K}^{(j)})$ where $c_{t,i}^{(j)} = \ell(f_{t,i}^{(j)}(\mathbf{x}_t^{(j)}), y_t^{(j)})$ is the loss of $f_{t,i}^{(j)}$, $i \in [K]$. The $j$-th client will send $c_{t,i}^{(j)}$, $i \in O_t^{(j)}$, to server. Since $c_{t,i}^{(j)}$, $i \notin O_t^{(j)}$ can not be observed, it is necessary to construct an estimated loss vector $\tilde{\mathbf{c}}_t^{(j)} = (\tilde{c}_{t,1}^{(j)}, \ldots, \tilde{c}_{t,K}^{(j)})$ where $\tilde{c}_{t,i}^{(j)} = \frac{c_{t,i}^{(j)}}{\mathbb{P}[i \in O_t^{(j)}]} \cdot \mathbb{I}_{i \in O_t^{(j)}}$, $i \in [K]$. It is easy to prove that $\mathbb{E}_t \left[ \tilde{c}_{t,i}^{(j)} \right] = c_{t,i}^{(j)}$ and $\mathbb{E}_t \left[ (\tilde{c}_{t,i}^{(j)})^2 \right] \leq \frac{K-1}{J-1} (c_{t,i}^{(j)})^2$ where $\mathbb{E}_t[\cdot] := \mathbb{E} \left[ \cdot | O_{[t-1]}^{(j)} \right]$. Thus sampling $A_{t,a}$, $a \geq 2$ reduces the variance of the estimators which is equivalent to obtain more information on the true loss.

Server aggregates $\tilde{\mathbf{c}}_t^{(j)}$, $j \in [M]$ and updates $\mathbf{p}_t$ following (4)-(6). Let $\Delta_K$ be the $(K-1)$-dimensional simplex, $\Omega = \Delta_K$ and $\tilde{g}_t^{(j)} = \tilde{\mathbf{c}}_t^{(j)}$. Then the server executes (8).

$$\begin{cases} \nabla_{\bar{\mathbf{p}}_{t+1}} \psi_t(\bar{\mathbf{p}}_{t+1}) = \nabla_{\mathbf{p}_t} \psi_t(\mathbf{p}_t) - \frac{1}{M} \sum_{j=1}^{M} \tilde{\mathbf{c}}_t^{(j)}, \\ \mathbf{p}_{t+1} = \arg\min_{\mathbf{p} \in \Delta_K} \mathcal{D}_{\psi_t}(\mathbf{p}, \bar{\mathbf{p}}_{t+1}), \\ \psi_t(\mathbf{p}) = \sum_{i=1}^{K} \frac{C_i}{\eta_t} p_i \ln p_i, \end{cases} \tag{8}$$

where $\psi_t(\mathbf{p})$ is the weighted negative entropy regularizer [Bubeck et al., 2017], $\eta_t > 0$ is a time-variant learning rate and $C_i > 0$ satisfies that $\max_{t,j} c_{t,i}^{(j)} \leq C_i$. It is obvious that server does not broadcast $\mathbf{p}_{t+1}$.

**Updating hypotheses** For each $j \in [M]$ and $i \in [K]$, let $\nabla_{t,i}^{(j)} = \nabla_{\mathbf{w}_{t,i}^{(j)}} \ell \left( \left\langle \mathbf{w}_{t,i}^{(j)}, \phi_i(\mathbf{x}_t^{(j)}) \right\rangle, y_t^{(j)} \right)$. Since $\nabla_{t,i}^{(j)}$, $i \notin O_t^{(j)}$ are unknown, it is necessary to construct an estimator of the gradient, denoted by $\tilde{\nabla}_{t,i}^{(j)} = \frac{\nabla_{t,i}^{(j)}}{\mathbb{P}[i \in O_t^{(j)}]} \cdot \mathbb{I}_{i \in O_t^{(j)}}$ for all $j \in [M], i \in [K]$. Clients send $\{\nabla_{t,i}^{(j)}, i \in O_t^{(j)}\}, j \in [M]$ to server. Then server aggregates $\{\tilde{\nabla}_{t,i}^{(j)}, i \in [K]\}, j \in [M]$ and updates the hypotheses following

(4)-(6). For each $i \in [K]$, let $\Omega = \mathcal{F}_i$ and $\tilde{g}_t^{(j)} = \tilde{\nabla}_{t,i}^{(j)}$. Server executes (9).

$$
\begin{cases}
\nabla_{\bar{\mathbf{w}}_{t+1,i}} \psi_{t,i}(\bar{\mathbf{w}}_{t+1,i}) = \nabla_{\mathbf{w}_{t,i}} \psi_{t,i}(\mathbf{w}_{t,i}) - \dfrac{1}{M} \sum_{j=1}^{M} \tilde{\nabla}_{t,i}^{(j)}, \quad i = 1, ..., K, \\[2mm]
\mathbf{w}_{t+1,i} = \underset{\mathbf{w} \in \mathcal{F}_i}{\arg\min}\, \mathcal{D}_{\psi_{t,i}}(\mathbf{w}, \bar{\mathbf{w}}_{t+1,i}), \\[2mm]
\psi_{t,i}(\mathbf{w}) = \dfrac{1}{2\lambda_{t,i}} \|\mathbf{w}\|_2^2,
\end{cases}
\tag{9}
$$

where $\psi_{t,i}(\mathbf{w})$ is the Euclidean regularizer and $\lambda_{t,i}$ is a time-variant learning rate.

We name this algorithm FOMD-OMS (FOMD-No-LU for OMS-DecD) and show it in Algorithm 2.

### 5.3   Regret bounds

To obtain high-probability regret bounds that adapt to the complexity of individual hypothesis space, we establish a new Bernstein's inequality for martingale.

**Lemma 1.** *Let $X_1, \ldots, X_n$ be a bounded martingale difference sequence w.r.t. the filtration $\mathcal{H} = (\mathcal{H}_k)_{1 \leq k \leq n}$ and with $|X_k| \leq a$. Let $Z_t = \sum_{k=1}^{t} X_k$ be the associated martingale. Denote the sum of the conditional variances by $\Sigma_n^2 = \sum_{k=1}^{n} \mathbb{E}\left[X_k^2 | \mathcal{H}_{k-1}\right] \leq v$, where $v \in [0, B]$ is a random variable and $B \geq 2$ is a constant. Then for any constant $a > 0$, with probability at least $1 - 2\lceil \log B \rceil \delta$,*

$$
\max_{t=1,\ldots,n} Z_t < \frac{2a}{3} \ln \frac{1}{\delta} + \sqrt{\frac{2}{B} \ln \frac{1}{\delta}} + 2\sqrt{v \ln \frac{1}{\delta}}.
$$

Note that $v$ is a random variable in Lemma 1, while it is a constant in standard Bernstein's inequality for martingale (see Lemma A.8 [Cesa-Bianchi and Lugosi, 2006]). Lemma 1 is derived from the standard Bernstein's inequality along with the well-known peeling technique [Bartlett et al., 2005].

**Assumption 1.** *For each $i \in [K]$, there is a constant $b_i$ such that $\|\phi_i(\mathbf{x})\|_2 \leq b_i$ where $\phi_i(\cdot)$ is defined in* (1).

**Lemma 2.** *Under Assumption 1, for each $i \in [K]$, there are two constants $C_i > 0, G_i > 0$ that depend on $U_i$ or $b_i$ such that $\max_{t,j} c_{t,i}^{(j)} \leq C_i$ and $\max_{t,j} \|\nabla_{t,i}^{(j)}\|_2 \leq G_i$.*

**Theorem 2.** *Let $R = T$. Under Assumption 1, denote by $A_m = \operatorname{argmin}_{i \in [K]} C_i$ and $C = \max_{i \in [K]} C_i$. Assuming that $\ell(\cdot, \cdot)$ is convex and $K \geq J \geq 2$. Let $g_{K,J} = \frac{K-J}{J-1}$,*

$$
\forall i \in A_m,\ p_{1,i} = \frac{1}{|A_m|}\left(1 - \frac{\sqrt{K}}{\sqrt{T}}\right) + \frac{1}{\sqrt{KT}} \quad \text{and} \quad \forall i \notin A_m,\ p_{1,i} = \frac{1}{\sqrt{KT}},
$$

$$
\forall t \in [T],\ \eta_t = \frac{\sqrt{\ln(KT)}}{2\sqrt{\left(1 + \frac{g_{K,J}}{M}\right)T}} \wedge \frac{1}{2g_{K,J}}, \quad \lambda_{t,i} = \frac{U_i}{2G_i\sqrt{\left(1 + \frac{g_{K,J}}{M}\right) \cdot \left(g_{K,J}^2 \vee t\right)}}.
$$

*With probability at least $1 - \Theta\left(M \log(T) + \log(KT/M)\right) \cdot \delta$, the regret of FOMD-OMS satisfies:*

$$
\forall i \in [K],\ \operatorname{Reg}_D(\mathcal{F}_i) = O\left(MB_{i,1}\sqrt{\left(1 + \frac{g_{K,J}}{M}\right)T} + B_{i,2} \cdot g_{K,J} \ln \frac{1}{\delta} + B_{i,3}\sqrt{g_{K,J}MT \ln \frac{1}{\delta}}\right),
$$

*where $B_{i,1} = U_i G_i + C_i\sqrt{\ln(KT)}$, $B_{i,2} = MC + U_i G_i$ and $B_{i,3} = U_i G_i + \sqrt{CC_i}$.*

Both $C_i$ and $G_i$ depend on $U_i$ or $b_i$ (see Lemma 2). Let $\mathfrak{C}_i = \Theta(U_i G_i + C_i)$. Thus $\mathfrak{C}_i$ measures the complexity of $\mathcal{F}_i$. Then our regret bound adapts to $\sqrt{\mathfrak{C}\mathfrak{C}_i}$ where $\mathfrak{C} = \max_{i \in [K]} \mathfrak{C}_i$, while previous regret bounds depend on $\mathfrak{C}$ [Ghari and Shen, 2022, Hong and Chae, 2022], that is, they can not adapt to the complexity of individual hypothesis space. If $\mathfrak{C}_{i^*} \ll \mathfrak{C}$, then our regret bound is much better.

The regret bound in Theorem 2 is also called multi-scale regret bound [Bubeck et al., 2017]. However, previous regret analysis can not yield a high-probability multi-scale bound. The reason is the lack of the new Bernstein's inequality for martingale (Lemma 1). If we use the new Freedman's inequality for martingale [Lee et al., 2020], then a high-probability bound can still be obtained, but is worse than the bound in Theorem 2 by a factor of order $O(\operatorname{poly}(\ln T))$.

## 5.4 Time Complexity and Communication Complexity Analysis

For each $j \in [M]$, the $j$-th client makes prediction and computes gradients in time $O(\sum_{i \in O_t^{(j)}} d_i)$. Server samples $O_t^{(j)}, j \in [M]$, aggregates gradients and updates global models. The per-round time complexity on server is $O(\sum_{j=1}^{M} \sum_{i \in O_t^{(j)}} d_i + \sum_{i=1}^{K} d_i + JM \log K)$.

**Upload** At any round $t \in [T]$, the $j$-th client transmits $c_{t,i}^{(j)}, \nabla_{t,i}^{(j)}, i \in O_t^{(j)}$ and the corresponding indexes to server. It requires $J(\sum_{i \in O_t^{(j)}} d_i + 1)$ floating-point numbers and $J$ integers. If we use 32 bits to represent a float, and use $\log K$ bits to represent an integer in $[K]$. Each client transmits $(32J(\sum_{i \in O_t^{(j)}} d_i + 1) + J \log K)$ bits to server.

**Download** Server broadcasts $\mathbf{w}_{t,i} \in \mathbb{R}^{d_i}, i \in O_t^{(j)}$ and the corresponding indexes to clients. The total download cost is $(32MJ(\sum_{i \in O_t^{(j)}} d_i + 1) + MJ \log K)$ bits.

## 5.5 Answers to Question 1

Before discussing Question 1, we give two lower bounds on the regret.

**Theorem 3** (Lower Bounds). *Assuming that $5 \leq K \leq \min\{d, T\}$. For each $i \in [K]$, let $\mathcal{F}_i = \{f_i(\mathbf{x}) = \mathbf{e}_i^\top \mathbf{x}\}$ and $\mathcal{D}_i = [\min_{\mathbf{x} \in \mathcal{X}} f_i(\mathbf{x}), \max_{\mathbf{x} \in \mathcal{X}} f_i(\mathbf{x})]$, where $\mathbf{e}_i$ is the standard basis vector in $\mathbb{R}^d$. Denote by $\sup$ the supremum over all examples.*

*(i) There are no computational constraints on clients. Let $\ell(v, y) = |v - y|$. The regret of any algorithm for OMS-DecD satisfies:* $\lim_{T \to \infty} \sup \max_{i \in [K]} \text{Reg}_D(\mathcal{F}_i) \geq 0.25M\sqrt{T \ln K}$;

*(ii) The per-round time complexity on each client is limited to $O(J)$. Let $\ell(v, y) = 1 - v \cdot y$. The regret of any, possibly randomized, noncooperative algorithm with outputs in $\cup_{i \in [K]} \mathcal{D}_i$ for OMS-DecD satisfies:* $\sup \mathbb{E}[\max_{i \in [K]} \text{Reg}_D(\mathcal{F}_i)] \geq 0.1M\sqrt{KTJ^{-1}}$, *where the expectation is taken over the randomization of algorithm.*

The assumption that the outputs of any noncooperative algorithm belong to $\cup_{i = \in [K]} \mathcal{D}_i$ is natural, and can be removed in the case of $J = 1$. Next we define a noncooperative algorithm, NCO-OMS.

**Definition 1** (NCO-OMS). *NCO-OMS independently samples $O_t^{(j)}$ following (7) and executes*

$$\forall j \in [M], \quad \nabla_{\bar{\mathbf{p}}_{t+1}} \psi_t(\bar{\mathbf{p}}_{t+1}) = \nabla_{\mathbf{p}_t^{(j)}} \psi_t\left(\mathbf{p}_t^{(j)}\right) - \tilde{\mathbf{c}}_t^{(j)}, \qquad \mathbf{p}_{t+1}^{(j)} = \arg\min_{\mathbf{p} \in \Delta_K} \mathcal{D}_{\psi_t}(\mathbf{p}, \bar{\mathbf{p}}_{t+1}).$$

$$\nabla_{\bar{\mathbf{w}}_{t+1,i}} \psi_{t,i}(\bar{\mathbf{w}}_{t+1,i}) = \nabla_{\mathbf{w}_{t,i}^{(j)}} \psi_{t,i}\left(\mathbf{w}_{t,i}^{(j)}\right) - \tilde{\nabla}_{t,i}^{(j)}, \quad \mathbf{w}_{t+1,i}^{(j)} = \arg\min_{\mathbf{w} \in \mathcal{F}_i} \mathcal{D}_{\psi_{t,i}}(\mathbf{w}, \bar{\mathbf{w}}_{t+1,i}),$$

*where the definitions of $\tilde{\mathbf{c}}_t^{(j)}$ and $\tilde{\nabla}_{t,i}^{(j)}$ follow FOMD-OMS.*

It is easy to prove the regret of NCO-OMS satisfies: with probability at least $1 - \Theta\left(M \log(KT)\right) \cdot \delta$,

$$\forall i \in [K], \ \text{Reg}_D(\mathcal{F}_i) = O\left(M\left(B_{i,1}\sqrt{(1 + g_{K,J})T} + B_{i,2}g_{K,J} \ln \frac{1}{\delta} + B_{i,3}\sqrt{g_{K,J}T \ln \frac{1}{\delta}}\right)\right),$$

where $B_{i,1} = U_i G_i + C_i \sqrt{\ln(KT)}$, $B_{i,2} = C + U_i G_i$ and $B_{i,3} = U_i G_i + \sqrt{CC_i}$. We leave the pseudo-code of NCO-OMS and the corresponding regret analysis in appendix.

Next we discuss Question 1 by considering two cases.

**Case 1**: There are no computational constraints on clients. Collaboration is unnecessary.

Let $J = \Theta(K)$ in FOMD-OMS and NCO-OMS. By Theorem 2, both FOMD-OMS and NCO-OMS enjoy a $O(MU_i G_i \sqrt{T} + MC_i \sqrt{T \ln(KT)})$ regret. By Theorem 3, FOMD-OMS and NCO-OMS are nearly optimal in terms of the dependence on $M$ and $T$. Thus collaboration is unnecessary.

**Case 2**: The per-round time complexity on each client is limited to $o(K)$. Collaboration is necessary.

Let $J = o(K)$ in FOMD-OMS and Theorem 3. By Theorem 2, FOMD-OMS enjoys a $O(MB_{i,1}\sqrt{T} + B_{i,3}\sqrt{MKTJ^{-1} \ln \delta^{-1}})$ regret, which is smaller than the lower bound on the regret of any noncooperative algorithm (see Theorem 3). Thus collaboration is necessary.

Table 1: Comparison with previous algorithms. $D$ is the number of random features [Rahimi and Recht, 2007]. Time (s) is the per-round time complexity on client.

| Algorithm | Regret bound | Time (s) | Download (bits) |
|---|---|---|---|
| eM-KOFL | $\tilde{O}\left(\mathfrak{C}M\sqrt{T\ln K} + \frac{\mathfrak{c}_i MT}{\sqrt{D}}\right)$ | $O(DK)$ | $O(DM\log K)$ |
| POF-MKL | $\tilde{O}\left(\mathfrak{C}M\sqrt{KT} + \frac{\mathfrak{c}_i MT}{\sqrt{D}}\right)$ | $O(DK)$ | $O(DMK)$ |
| FOMD-OMS | $\tilde{O}\left(\mathfrak{C}_i M\sqrt{T\ln K} + \sqrt{\mathfrak{C}\mathfrak{c}_i MKT} + \frac{\mathfrak{c}_i MT}{\sqrt{D}}\right)$ | $O(D)$ | $O(DM\log K)$ |

## 6    Application to Distributed OMKL

We will apply the proposed FOMD-OMS to a special instance of OMS-DecD, known as distributed OMKL, in which $\mathcal{F}_i$ is a RKHS. Then we contrast our results with those from earlier studies, highlighting the unnecessary nature of collaboration in prior federated algorithms.

**Theorem 4.** *Let $\mathcal{F}_i$ be a RHKS for all $i \in [K]$ and $R \leq T$. With probability at least $1 - \Theta\left(TM\log(R) + T\log(KR/M)\right) \cdot \delta$, the regret of FOMD-OMS satisfies, $\forall i \in [K]$,*

$$\text{Reg}_D(\mathcal{F}_i) = \tilde{O}\left(MB_{i,1}\sqrt{1 + \frac{g_{K,J}}{M}} \cdot \frac{T}{\sqrt{R}} + \frac{B_{i,2}Mg_{K,J}T}{R} + \frac{B_{i,3}T}{\sqrt{R}}\sqrt{Mg_{K,J}} + \frac{U_iG_iMT}{\sqrt{D}}\right),$$

*where the notation $\tilde{O}(\cdot)$ hides polylogarithmic factor in $\delta^{-1}$ and $D = d_i$ follows* (1).

We defer the algorithm in appendix. Let $R = T$ and $J = 2$. We compare FOMD-OMS with eM-KOFL [Hong and Chae, 2022] and POF-MKL [Ghari and Shen, 2022]. Table 1 gives the results.

We observe that FOMD-OMS significantly improves the computational complexity of eM-KOFL and POF-MKL (by a factor of $O(K)$) on each client. The per-round time complexity of the two algorithms is $O(DK)$. Recalling the answer to Question 1 (see Section 5.5), collaboration in eM-KOFL and POF-MKL is unnecessary.

Next we compare the regret bound of the three algorithms. Recalling that $\mathfrak{C}_i \leq \mathfrak{C}$. The regret bounds of eM-KOFL and POF-MKL can not adapt to the complexity of individual hypothesis space. (i) The regret bound of FOMD-OMS is better than that of POF-MKL in relation to its dependence on $M$ and $\mathfrak{C}_i$. (ii) In the case of $K = O\left(\frac{\mathfrak{C}}{\mathfrak{C}_i}M \cdot \ln K\right)$, the regret bound of FOMD-OMS is better than that of eM-KOFL. (iii) In the case of $K = \Omega\left(\frac{\mathfrak{C}}{\mathfrak{C}_i}M \cdot \ln K\right)$, the regret bound of FOMD-OMS is worse than that of eM-KOFL. If $K$ is sufficiently large, the regret bound of eM-KOFL is better than that of FOMD-OMS by a factor of $O(\sqrt{K})$.

## 7    Conclusion

In this paper, we have studied the necessity of collaboration for OMS-DecD from the perspective of computational constraints. We demonstrate that collaboration is unnecessary when there are no computational constrains on clients, while it becomes necessary if the time complexity on each client is limited to $o(K)$. Our work clarifies the unnecessary nature of collaboration in previous algorithms for the first time, gives conditions under which collaboration is necessary, and provides inspirations for studying the problem from constraints beyond computational constrains.

## Acknowledgements

We thank all anonymous reviewers for their valuable comments and suggestions. This work was supported by the Major Key Project of PCL (No. 2022ZD0115301).

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

## A Limitations

The main limitation of this work lies in the first lower bound in Theorem 3 that holds in the case of $T \to \infty$. Although this lower bound nearly matches our upper bound asymptotically and is enough to answer Question 1, it is desired to establish a non-asymptotical lower bound, that is, the lower bound holds for any value of $T$.

## B Broader Impact

This work can be potentially applied to distributed online prediction tasks. We aim to address a fundamental problem whether collaboration among clients is necessary and under what conditions collaboration is necessary. A predictable economic benefit of our work is to give a guidance on the usage of federated learning and save unnecessary communication overhead.

Our work can also instruct the machine learning engineers to tune the hyper-parameter of online learning algorithms. Thus our work can alleviate the burden of machine learning engineers and improve the utility of online leaning algorithms in industrial applications.

## C Experiments

In this section, we aim to verify the following three goals which are our main results.

**G**1 Collaboration is unnecessary if we allow the computational cost on each client to be $O(K)$.
   We set $R = T$ and $J = K$ in FOMD-OMS. In this case, the per-round running time on each client is $O(K)$. We aim to verify that FOMD-OMS enjoys similar prediction performance with the noncooperative algorithm, NCO-OMS with $J = K$ (see Definition 1).

**G**2 Collaboration is necessary if we limit the computational cost on each client to $o(K)$.
   We set $R = T$ and $J = 2$ in FOMD-OMS. In this case, the per-round running time on each client is $O(1)$. We aim to verify that FOMD-OMS enjoys better prediction performance than NCO-OMS with $J = 2$.

**G**3 FOMD-OMS improves the regret bounds of algorithms for distributed OMKL.
   FOMD-OMS with $R = T$ and $J = 2$ enjoys similar prediction performance with eM-KOFL [Hong and Chae, 2022], and enjoys better prediction performance than POF-MKL [Ghari and Shen, 2022] at a smaller computational cost on each client.

   Although there are more baseline algorithms, such as vM-KOFL [Hong and Chae, 2022], pM-KOFL [Hong and Chae, 2022] and OFSKL [Ghari and Shen, 2022], we do not compare with the three algorithms since they do not perform as well as eM-KOFL and POF-MKL.

### C.1 Experimental setting

We will execute three experiments and each one verifies a goal. For simplicity, we do not measure the actual communication cost and use serial implementation to simulate the distributed implementation.

To verify **G**1 and **G**2, we use the instance of online model selection given in Example 1. The first experiment verifies **G**1. We construct 10 nested hypothesis spaces (i.e., $K = 10$) as follows

$$\forall i \in [10], \quad \mathcal{F}_i = \{f(\mathbf{x}) = \langle \mathbf{w}, \mathbf{x} \rangle, \|\mathbf{w}\|_2 \le U_i\},$$

where $U_i = \frac{i}{10}$. We set $R = T$ and $J = K$ in FOMD-OMS. Since $J = K$, we have $O_t^{(j)} = [K]$ and $\mathbb{P}\left[i \in O_t^{(j)}\right] = 1$. The learning rates $\eta_t, \lambda_{t,i}, i \in [K]$ of FOMD-OMS follow Theorem 2. For NCO-OMS, we set $J = K$ and set the learning rate $\eta_t, \lambda_{t,i}, i \in [K]$ following Theorem 2 in which $M = 1$, i.e.,

$$\forall t \in [T], \ \eta_t = \frac{\sqrt{\ln(KT)}}{2\sqrt{T}}, \quad \lambda_{t,i} = \frac{U_i}{2G_i\sqrt{t}}.$$

We use the square loss function $\ell(f(\mathbf{x}), y) = (f(\mathbf{x}) - y)^2$. For both FOMD-OMS and NCO-OMS, we tune $G_i = (U_i + 1) \times \{1, 2, 4, 6, 8, 10\}$ and set $C_i = (U_i + 1)^2$.

Table 2: Basic information of datasets.

| Dataset | Number of Instances | Number of features |
|---|---|---|
| bank | 8,190 | 32 |
| ailerons | 13,750 | 40 |
| calhousing | 14,000 | 8 |
| elevators | 16,590 | 18 |
| TomsHardware | 28,170 | 96 |
| Twitter | 50,000 | 77 |
| Year | 51,630 | 90 |
| Slice | 53,500 | 384 |

The second experiment verifies **G**2. We set $R = T$ and $J = 2$ in FOMD-OMS. The learning rates of FOMD-OMS also follow Theorem 2. For NCO-OMS, we also set $J = 2$ and set the learning rate $\eta_t, \lambda_{t,i}, i \in [K]$ following Theorem 2 in which $M = 1$, i.e.,

$$\forall t \in [T], \quad \eta_t = \frac{\sqrt{\ln(KT)}}{2\sqrt{(K-1)T}} \wedge \frac{1}{2(K-2)}, \quad \lambda_{t,i} = \frac{U_i}{2G_i\sqrt{(K-1) \cdot ((K-2)^2 \vee t)}}.$$

Similar to the first experiment, we tune $G_i = (U_i + 1) \times \{1, 2, 4, 6, 8, 10\}$ and set $C_i = (U_i + 1)^2$.

The third experiment verifies **G**3. We consider online kernel selection (as known as online multi-kernel learning) which is an instance of online model selection given in Example 2. We select the Gaussian kernel with 8 different kernel widths (i.e., $K = 8$),

$$\forall i \in [8], \quad \kappa_i(\mathbf{x}, \mathbf{v}) = \exp\left(-\frac{\|\mathbf{x} - \mathbf{v}\|_2^2}{2\sigma_i^2}\right), \sigma_i = 2^{i-2},$$

and construct the corresponding hypothesis space $\mathcal{F}_i$ and $\mathbb{H}_i$ following (21) in which we set $U_i = U$ and $D_i = D$ for all $i \in [K]$ and tune $U \in \{1, 2, 4\}$. Note that $U_i$ is same for all $i \in [K]$. We replace the initial distribution $\mathbf{p}_1$ in Theorem 2 with a uniform distribution $(\frac{1}{K}, \ldots, \frac{1}{K})$. We set $D = 100$ for FOMD-OMS, eM-KOFL and POF-MKL. $D$ is the number of random features. We set $R = T$, $J = 2$ and $C = U + 1$ in FOMD-OMS. Thus the per-round time complexity on each client is $O(D)$ and the per-round communication cost is $O(MD + M \log K)$. There are three hyper-parameters in eM-KOFL, i.e., $\eta_g$, $\eta_l$ and $\lambda$. $\eta_g$ is the global learning rate, $\eta_l$ is the local learning rate and $\lambda$ is a regularization parameter. There are $2M + 3$ hyper-parameters in POF-MKL, i.e., $\eta_g, \eta_j, \xi_j, j \in [M]$, $m$, $\lambda$ in which $M/m$ plays the same role with $J$ in FOMD-OMS. $\eta_g$ is the global learning rate, $\eta_j$ is the local learning rate, $\xi_j$ is called exploration rate and $\lambda$ is a regularization parameter. Since $J = 2$ in FOMD-OMS, we can set $m = M/2$ for FOMD-OMS. Following the original paper [Ghari and Shen, 2022], we set $\xi_j = 1$. For a fair comparison, we change the learning rates of FOMD-OMS, eM-KOFL and POF-MKL. Following the parameter setting of eM-KOFL [Hong and Chae, 2022], we tune $\eta_g, \eta_l, \eta_j \in \{0.1, 0.5, 1, 4, 8, 16\}$ and $\lambda \in \{0.1, 0.001, 0.0001\}$ for eM-KOFL and POF-MKL. For FOMD-OMS, we also tune $\eta_t, \lambda_{t,i} \in \{0.1, 0.5, 1, 4, 8, 16\}$.

For all of the three experiments, we set 10 clients, i.e., $M = 10$. We use 8 regression datasets shown in Table 2 from WEKA [2] [Hall et al., 2009] and UCI machine learning repository [3], and rescale the target variables and features of all datasets to fit in [0,1] and [-1,1] respectively. For each dataset, we randomly divide it into 10 subsets and each subset simulates the data on a client. We randomly permutate the instances in the datasets 10 times and report the average results. All algorithms are implemented with R on a Windows machine with 2.8 GHz Core(TM) i7-1165G7 CPU [4].

We use the square loss function and define the mean squared error (MSE) of all algorithms, i.e.,

$$\text{MSE} = \frac{1}{MT}\sum_{j=1}^{M}\sum_{t=1}^{T}\left(f_t^{(j)}\left(\mathbf{x}_t^{(j)}\right) - y_t^{(j)}\right)^2.$$

We record the mean of MSE over 10 random experiments, and the standard deviation of the mean of MSE. We also record the mean of of the total running time on each client, and the standard deviation of the mean of running time.

---

[2] https://waikato.github.io/weka-wiki/datasets/

[3] https://archive.ics.uci.edu/ml/index.php

[4] The code of algorithms is available at https://github.com/JunfLi-TJU/OMS-DecD.git

Table 3: Comparison with the noncooperative algorithm. $\Delta$ is the difference of MSE between NCO-OMS and FOMD-OMS. $a\text{E-}b = a \times 10^{-b}$, $a > 0, b > 0$.

| Algorithm | elevator | | | | bank | | | |
|---|---|---|---|---|---|---|---|---|
| | MSE$\times 10^2$ | $J$ | Time (s) | $\Delta$ | MSE$\times 10^2$ | $J$ | Time (s) | $\Delta$ |
| NCO-OMS | $0.991 \pm 0.002$ | $K$ | $1.31 \pm 0.10$ | 1E-4 | $2.158 \pm 0.022$ | $K$ | $0.88 \pm 0.05$ | 1E-3 |
| FOMD-OMS | $\mathbf{0.980 \pm 0.005}$ | $K$ | $0.65 \pm 0.08$ | | $\mathbf{2.020 \pm 0.005}$ | $K$ | $0.33 \pm 0.08$ | |
| NCO-OMS | $1.168 \pm 0.005$ | $2$ | $0.58 \pm 0.04$ | 1E-3 | $2.321 \pm 0.021$ | $2$ | $0.39 \pm 0.05$ | 2E-3 |
| FOMD-OMS | $\mathbf{1.024 \pm 0.002}$ | $2$ | $0.14 \pm 0.04$ | | $\mathbf{2.118 \pm 0.003}$ | $2$ | $0.08 \pm 0.03$ | |

| Algorithm | TomsHardware | | | | Twitter | | | |
|---|---|---|---|---|---|---|---|---|
| | MSE$\times 10^2$ | $J$ | Time (s) | $\Delta$ | MSE$\times 10^2$ | $J$ | Time (s) | $\Delta$ |
| NCO-OMS | $0.090 \pm 0.004$ | $K$ | $3.02 \pm 0.29$ | 1E-4 | $0.017 \pm 0.000$ | $K$ | $5.11 \pm 0.24$ | 0 |
| FOMD-OMS | $0.083 \pm 0.008$ | $K$ | $1.48 \pm 0.28$ | | $0.017 \pm 0.000$ | $K$ | $2.07 \pm 0.07$ | |
| NCO-OMS | $0.150 \pm 0.002$ | $2$ | $1.11 \pm 0.07$ | 4E-4 | $0.018 \pm 0.000$ | $K$ | $2.24 \pm 0.25$ | 1E-5 |
| FOMD-OMS | $\mathbf{0.107 \pm 0.003}$ | $2$ | $0.44 \pm 0.09$ | | $\mathbf{0.017 \pm 0.000}$ | $2$ | $0.51 \pm 0.05$ | |

| Algorithm | ailerons | | | | calhousing | | | |
|---|---|---|---|---|---|---|---|---|
| | MSE$\times 10^2$ | $J$ | Time (s) | $\Delta$ | MSE$\times 10^2$ | $J$ | Time (s) | $\Delta$ |
| NCO-OMS | $19.506 \pm 0.033$ | $K$ | $1.41 \pm 0.04$ | 3E-4 | $10.166 \pm 0.029$ | $K$ | $1.07 \pm 0.05$ | 3E-4 |
| FOMD-OMS | $19.480 \pm 0.046$ | $K$ | $0.74 \pm 0.08$ | | $10.136 \pm 0.012$ | $K$ | $0.43 \pm 0.05$ | |
| NCO-OMS | $20.323 \pm 0.036$ | $2$ | $0.65 \pm 0.05$ | 5E-3 | $10.372 \pm 0.021$ | $2$ | $0.53 \pm 0.04$ | 1E-3 |
| FOMD-OMS | $\mathbf{19.820 \pm 0.032}$ | $2$ | $0.20 \pm 0.04$ | | $\mathbf{10.227 \pm 0.014}$ | $2$ | $0.12 \pm 0.05$ | |

| Algorithm | year | | | | Slice | | | |
|---|---|---|---|---|---|---|---|---|
| | MSE$\times 10^2$ | $J$ | Time (s) | $\Delta$ | MSE$\times 10^2$ | $J$ | Time (s) | $\Delta$ |
| NCO-OMS | $20.322 \pm 0.040$ | $K$ | $5.94 \pm 0.25$ | 2E-3 | $13.097 \pm 0.009$ | $K$ | $10.40 \pm 0.94$ | 1E-3 |
| FOMD-OMS | $\mathbf{20.096 \pm 0.045}$ | $K$ | $2.25 \pm 0.20$ | | $\mathbf{12.964 \pm 0.007}$ | $K$ | $4.12 \pm 0.18$ | |
| NCO-OMS | $24.334 \pm 0.021$ | $2$ | $2.95 \pm 0.63$ | 1E-2 | $13.364 \pm 0.012$ | $2$ | $3.60 \pm 0.23$ | 3E-3 |
| FOMD-OMS | $\mathbf{22.705 \pm 0.040}$ | $2$ | $0.59 \pm 0.09$ | | $\mathbf{13.038 \pm 0.009}$ | $2$ | $1.41 \pm 0.12$ | |

## C.2 Results of the First and the Second Experiment

We summary the experimental results of the first and the second experiments in Table 3.

In Table 3, $\Delta$ is defined as the difference of MSE between NCO-OM and FOMD-OMS. Thus $\Delta$ shows whether collaboration improves the prediction performance of the noncooperative algorithm. Times (s) records the total running time on all clients.

We first consider the case $J = K$ in which the per-round time complexity on each client is $O(K)$. It is obvious that the MSE of NCO-OMS is similar with that of FOMD-OMS. Although there are four datasets on which FOMD-OMS performs better than NCO-OMS, such as the *elevator*, *bank*, *Year* and *Slice* datsets, the improvement is very limited. Beside, the value of $\Delta$ is very small. Thus collaboration does not significantly improve the prediction performance of the noncooperative algorithm. The results verify the first goal **G**1.

Next we consider the case $J = 2$ in which the per-round time complexity on each client is $O(1)$. It is obvious that FOMD-OMS performs better than NCO-OMS on all datasets. Besides, the value of $\Delta$ in the case of $J = 2$ is much larger than that in the case of $J = K$, such as the *elevators*, *ailerons*, *ailerons* and *Year* datasets. Thus collaboration indeed improves the prediction performance of the noncooperative algorithm. The results verify the second goal **G**2.

Finally we compare the running time of all algorithms. It is obvious that FOMD-OMS with $J = 2$ runs faster than the other algorithms. The results coincide with our theoretical analysis. NCO-OMS runs slower than FOMD-OMS. The reason is that NCO-OMS must solve the sampling probability $\mathbf{p}_t$ using an additional binary search on each client (see Section G.1). In other words, NCO-OMS must execute binary search $M$ times at each round. FOMD-OMS only executes one binary search on server at each round. The improvement on the computational cost is benefit from decoupling model selection and prediction.

## C.3 Results of the Third Experiment

We summary the experimental results of the third experiment in Table 4.

Table 4: Comparison with the state-of-the-art algorithms.

| Algorithm | elevator | | | bank | | |
|---|---|---|---|---|---|---|
| | MSE | $J$ | Time (s) | MSE | $J$ | Time (s) |
| eM-KOFL | **0.00292 ± 0.00013** | - | 2.67 ± 0.05 | **0.01942 ± 0.00066** | - | 1.41 ± 0.06 |
| POF-MKL | 0.00806 ± 0.00026 | - | 3.12 ± 0.14 | 0.02292 ± 0.00036 | - | 1.59 ± 0.13 |
| FOMD-OMS | **0.00318 ± 0.00021** | 2 | 0.52 ± 0.08 | **0.01917 ± 0.00110** | 2 | 0.27 ± 0.06 |

| Algorithm | TomsHardware | | | Twitter | | |
|---|---|---|---|---|---|---|
| | MSE | $J$ | Time (s) | MSE | $J$ | Time (s) |
| eM-KOFL | **0.00048 ± 0.00003** | - | 5.88 ± 0.69 | **0.00007 ± 0.00000** | - | 9.60 ± 0.77 |
| POF-MKL | 0.00188 ± 0.00004 | - | 6.60 ± 0.93 | 0.00020 ± 0.00001 | 2 | 10.44 ± 0.54 |
| FOMD-OMS | 0.00059 ± 0.00003 | 2 | 1.46 ± 0.12 | 0.00010 ± 0.00001 | 2 | 2.23 ± 0.18 |

| Algorithm | ailerons | | | calhousing | | |
|---|---|---|---|---|---|---|
| | MSE | $J$ | Time (s) | MSE | $J$ | Time (s) |
| eM-KOFL | **0.00370 ± 0.00011** | - | 2.40 ± 0.19 | **0.02242 ± 0.00043** | - | 2.28 ± 0.06 |
| POF-MKL | 0.01335 ± 0.00046 | - | 2.66 ± 0.12 | 0.05248 ± 0.00197 | - | 2.68 ± 0.08 |
| FOMD-OMS | 0.00429 ± 0.00021 | 2 | 0.48 ± 0.04 | **0.02373 ± 0.00126** | 2 | 0.39 ± 0.07 |

| Algorithm | year | | | Slice | | |
|---|---|---|---|---|---|---|
| | MSE | $J$ | Time (s) | MSE | $J$ | Time (s) |
| eM-KOFL | **0.01481 ± 0.00108** | - | 9.60 ± 0.51 | **0.05781 ± 0.00230** | - | 12.74 ± 0.95 |
| POF-MKL | 0.01896 ± 0.00036 | - | 10.73 ± 0.29 | 0.08675 ± 0.00402 | - | 14.22 ± 0.54 |
| FOMD-OMS | **0.01534 ± 0.00121** | 2 | 2.26 ± 0.10 | **0.05698 ± 0.00480** | 2 | 4.82 ± 0.21 |

We first compare FOMD-OMS with eM-KOFL. As a whole, the MSE of the two algorithms is similar. On the *TomsHardware*, *Twitter* and *ailerons* datasets, eM-KOFL enjoys slightly better prediction performance than FOMD-OMS. However, the running time of eM-KOFL is much larger than that of FOMD-OMS. The results coincide with the theoretical observations that FOMD-OMS enjoys a similar regret bound with eM-KOFL at a much smaller computational cost on the clients.

Next we compare FOMD-OMS with POF-MKL. Both the MSE and running time of FOMD-OMS are much smaller than that of POF-MKL. The results coincide with the theoretical observations that FOMD-OMS enjoys a smaller regret bound than POF-MKL at a much smaller computational cost on the clients.

Thus the results in Table 4 verifies the third goal **G**3.

Finally, we explain that why POF-MKL performs worse than FOMD-OMS. There are three reasons.

(1) POF-MKL does not use federated learning to learn a global probability distribution denoted by $\mathbf{p}_t$, but learns a personalized probability distribution denoted by $\mathbf{p}_{t,j}$ on each client. Thus POF-MKL converges to the best kernel function at a lower rate.

(2) POF-MKL uniformly samples two kernel functions and then learns two global hypotheses, while FOMD-OMS uses $\mathbf{p}_t$ to sample a kernel function and learns a global hypothesis. Thus POF-MKL can learn a better global hypothesis.

(3) On each client, POF-MKL executes model selection and combines the predictions of $K$ hypotheses using $\mathbf{p}_{t,j}$. Thus the time complexity is in $O(DK)$. FOMD-OMS executes model selection on server, and only uses the sampled hypothesis to make prediction. Thus the time complexity on each client is in $O(D)$.

# D  A Federated Algorithm for OMS-DecD with Communication Constraints

Let $R < T$. Clients communicate with server every $N$ rounds. At the $r$-th communication, clients transmit $\{\frac{1}{N} \sum_{t \in T_r} \nabla_{t,i}^{(j)}, \frac{1}{N} \sum_{t \in T_r} c_{t,i}^{(j)}\}_{i \in O_t^{(j)}}$ to server at the last round in $T_r$. Then server updates sampling probabilities and hypotheses. We give the pseudo-code in Algorithm 3.

---

**Algorithm 3** FOMD-OMS $(R < T)$

---

**Require:** $U, T, R, J$.
**Ensure:** $f_{1,i}^{(j)} = 0, p_{1,i}, i \in [K], j \in [M]$
1: **for** $r = 1, 2, \ldots, R$ **do**
2:    **for** $t \in T_r$ **do**
3:       **if** $t == (r-1)N + 1$ **then**
4:          **for** $j = 1, \ldots, M$ **do**
5:             Server samples $O_t^{(j)}$ following (7)
6:             Server transmits $f_{t,i}^{(j)}, i \in O_t^{(j)}$ to the $j$-th client
7:          **end for**
8:       **end if**
9:       **for** $j = 1, \ldots, M$ in parallel **do**
10:          Output $f_{t,A_{t,1}}^{(j)}(\mathbf{x}_t^{(j)})$
11:          **for** $i \in O_t^{(j)}$ **do**
12:             Computing $\nabla_{t,i}^{(j)}$ and $c_{t,i}^{(j)}$
13:          **end for**
14:       **end for**
15:       **if** $t == rN$ **then**
16:          Clients transmit $\{\frac{1}{N}\sum_{t \in T_r} \nabla_{t,i}^{(j)}, \frac{1}{N}\sum_{t \in T_r} c_{t,i}^{(j)}\}_{i \in O_t^{(j)}}$ to server
17:          Server computes $\mathbf{p}_{t+1}$ following (8)
18:          Server computes $\mathbf{w}_{t+1,i}, i \in [K]$ following (9)
19:       **end if**
20:    **end for**
21: **end for**

---

**Theorem 5.** *Let $R < T$. For any $t \in [R]$, let*

$$\forall i \in A_m, \ p_{1,i} = \frac{1}{|A_m|}\left(1 - \frac{\sqrt{K}}{\sqrt{R}}\right) + \frac{1}{\sqrt{KR}} \quad \text{and} \quad \forall i \notin A_m, \ p_{1,i} = \frac{1}{\sqrt{KR}},$$

$$\forall t \in [R], \ \eta_t = \frac{\sqrt{\ln(KR)}}{2\sqrt{\left(1 + \frac{g_{K,J}}{M}\right)R}} \wedge \frac{1}{2g_{K,J}}, \quad \lambda_{t,i} = \frac{U_i}{2G_i\sqrt{\left(1 + \frac{g_{K,J}}{M}\right)\cdot\left(g_{K,J}^2 \vee t\right)}}.$$

*Under the condition of Theorem 2, with probability at least $1 - \Theta\left(\frac{T}{R}M\log(R) + \frac{T}{R}\log(KR/M)\right)\cdot\delta$, the regret of FOMD-OMS satisfies:* $\forall i \in [K]$,

$$\text{Reg}_D(\mathcal{F}_i) = O\left(MB_{i,1}\sqrt{1 + \frac{g_{K,J}}{M}} \cdot \frac{T}{\sqrt{R}} + \frac{T}{R}\cdot B_{i,2}Mg_{K,J}\ln\frac{1}{\delta} + \frac{B_{i,3}T}{\sqrt{R}}\sqrt{Mg_{K,J}\ln\frac{1}{\delta}}\right).$$

The regret bound depends on $O(\frac{1}{\sqrt{R}})$. Thus FOMD-OMS explicitly balances the prediction performance and the communication cost.

*Proof.* If FOMD-OMS runs on a sequence of examples with length $T = R$, then Theorem 2 gives, with probability at least $1 - \Theta\left(M\log(CR) + \log(CKR/M)\right)\cdot\delta$,

$$\text{Reg}_D(\mathcal{F}_i) = O\left(MB_{i,1}\sqrt{\left(1 + \frac{g_{K,J}}{M}\right)R} + B_{i,2}g_{K,J}\ln\frac{1}{\delta} + B_{i,3}\sqrt{g_{K,J}MR\ln\frac{1}{\delta}}\right).$$

According to Theorem 1, the regret bound of FOMD-OMS with $R < T$ satisfies, with probability at least $1 - \Theta\left(\frac{T}{R}M\log(CR) + \frac{T}{R}\log(CKR/M)\right)\cdot\delta$,

$$\text{Reg}_D(\mathcal{F}_i) = O\left(NMB_{i,1}\sqrt{\left(1 + \frac{g_{K,J}}{M}\right)R} + NB_{i,2}g_{K,J}\ln\frac{1}{\delta} + NB_{i,3}\sqrt{g_{K,J}MR\ln\frac{1}{\delta}}\right)$$

$$= O\left(\frac{T}{\sqrt{R}}MB_{i,1}\sqrt{1 + \frac{g_{K,J}}{M}} + \frac{T}{R}\cdot B_{i,2}g_{K,J}\ln\frac{1}{\delta} + \frac{T}{\sqrt{R}}B_{i,3}\sqrt{g_{K,J}M\ln\frac{1}{\delta}}\right),$$

which concludes the proof. $\qquad\square$

# E   Proof of Theorem 1

We first state a technical lemma.

**Lemma 3** (Boyd and Vandenberghe [2004]). *Assuming that $\psi(\cdot) : \mathcal{X} \to \mathbb{R}$ is a convex and differential function, and $\mathcal{X}$ is a convex domain. Let $f^* = \operatorname{argmin}_{f \in \mathcal{X}} \psi(f)$. Then it must be*

$$\forall g \in \mathcal{X}, \quad \langle \nabla \psi(f^*), g - f^* \rangle \geq 0.$$

Lemma 3 gives the first-order optimality condition.

*Proof of Theorem 1.* We first consider the case $R = T$.

The main idea is to give an lower bound and upper bound on $\langle \bar{g}_t, \mathbf{u}_{t+1} - \mathbf{v} \rangle$, respectively. Next we give an upper bound.

$$
\begin{aligned}
&\langle \bar{g}_t, \mathbf{u}_{t+1} - \mathbf{v} \rangle \\
=& \langle \nabla_{\mathbf{u}_t} \psi_t(\mathbf{u}_t) - \nabla_{\bar{\mathbf{u}}_{t+1}} \psi_t(\bar{\mathbf{u}}_{t+1}), \mathbf{u}_{t+1} - \mathbf{v} \rangle \\
=& \langle \nabla_{\mathbf{u}_t} \psi_t(\mathbf{u}_t) - \nabla_{\mathbf{u}_{t+1}} \psi_t(\mathbf{u}_{t+1}), \mathbf{u}_{t+1} - \mathbf{v} \rangle + \langle \nabla_{\mathbf{u}_{t+1}} \psi_t(\mathbf{u}_{t+1}) - \nabla_{\bar{\mathbf{u}}_{t+1}} \psi_t(\bar{\mathbf{u}}_{t+1}), \mathbf{u}_{t+1} - \mathbf{v} \rangle \\
=& \mathcal{D}_{\psi_t}(\mathbf{v}, \mathbf{u}_t) - \mathcal{D}_{\psi_t}(\mathbf{v}, \mathbf{u}_{t+1}) - \mathcal{D}_{\psi_t}(\mathbf{u}_{t+1}, \mathbf{u}_t) - \langle \nabla_{\mathbf{u}_{t+1}} \mathcal{D}_{\psi_t}(\mathbf{u}_{t+1}, \bar{\mathbf{u}}_{t+1}), \mathbf{v} - \mathbf{u}_{t+1} \rangle \\
\leq& \mathcal{D}_{\psi_t}(\mathbf{v}, \mathbf{u}_t) - \mathcal{D}_{\psi_t}(\mathbf{v}, \mathbf{u}_{t+1}) - \mathcal{D}_{\psi_t}(\mathbf{u}_{t+1}, \mathbf{u}_t).
\end{aligned}
$$

The last inequality comes from Lemma 3.

Then we give a lower bound.

$$
\begin{aligned}
&\langle \bar{g}_t, \mathbf{u}_{t+1} - \mathbf{v} \rangle \\
=& \frac{1}{M} \sum_{j=1}^{M} \left[ \left\langle g_t^{(j)}, \mathbf{u}_{t+1} - \mathbf{v} \right\rangle + \left\langle \tilde{g}_t^{(j)} - g_t^{(j)}, \mathbf{u}_{t+1} - \mathbf{v} \right\rangle \right] \\
=& \frac{1}{M} \sum_{j=1}^{M} \left\langle g_t^{(j)}, \mathbf{u}_t^{(j)} - \mathbf{v} \right\rangle + \underbrace{\frac{1}{M} \sum_{j=1}^{M} \left\langle g_t^{(j)}, \mathbf{u}_{t+1} - \mathbf{u}_t \right\rangle}_{\Xi_1} + \underbrace{\frac{1}{M} \sum_{j=1}^{M} \left\langle \tilde{g}_t^{(j)} - g_t^{(j)}, \mathbf{u}_{t+1} - \mathbf{v} \right\rangle}_{\Xi_2},
\end{aligned}
$$

where $\mathbf{u}_t^{(j)} = \mathbf{u}_t$.

Next we analyze $\Xi_1$ and $\Xi_2$.

To analyze $\Xi_1$, we introduce an auxiliary variable $\mathbf{r}_{t+1}$ defined as follows

$$\nabla_{\mathbf{r}_{t+1}} \psi_t(\mathbf{r}_{t+1}) = \nabla_{\mathbf{u}_t} \psi_t(\mathbf{u}_t) - \frac{2}{M} \sum_{j=1}^{M} g_t^{(j)}.$$

Then we have

$$
\begin{aligned}
\Xi_1 =& \frac{1}{2} \left\langle \frac{2}{M} \sum_{j=1}^{M} g_t^{(j)}, \mathbf{u}_{t+1} - \mathbf{u}_t \right\rangle \\
=& \frac{1}{2} \left\langle \nabla_{\mathbf{u}_t} \psi_t(\mathbf{u}_t) - \nabla_{\mathbf{r}_{t+1}} \psi_t(\mathbf{r}_{t+1}), \mathbf{u}_{t+1} - \mathbf{u}_t \right\rangle \\
=& \frac{1}{2} \left( \mathcal{D}_\psi(\mathbf{u}_{t+1}, \mathbf{r}_{t+1}) - \mathcal{D}_\psi(\mathbf{u}_{t+1}, \mathbf{u}_t) - \mathcal{D}_\psi(\mathbf{u}_t, \mathbf{r}_{t+1}) \right) \\
\geq& -\frac{1}{2} \left( \mathcal{D}_\psi(\mathbf{u}_{t+1}, \mathbf{u}_t) + \mathcal{D}_\psi(\mathbf{u}_t, \mathbf{r}_{t+1}) \right).
\end{aligned}
$$

Before analyzing $\Xi_2$, we also introduce an auxiliary variable $\mathbf{q}_{t+1}$ defined as follows

$$\nabla_{\mathbf{q}_{t+1}} \psi_t(\mathbf{q}_{t+1}) = \nabla_{\mathbf{u}_t} \psi_t(\mathbf{u}_t) - \frac{2}{M} \sum_{j=1}^{M} \left( \tilde{g}_t^{(j)} - g_t^{(j)} \right).$$

Now we can analyze $\Xi_2$. We have

$$\Xi_2 = \frac{1}{2} \left\langle \frac{2}{M} \sum_{j=1}^{M} \left( \tilde{g}_t^{(j)} - g_t^{(j)} \right), \mathbf{u}_{t+1} - \mathbf{u}_t \right\rangle + \underbrace{\left\langle \frac{1}{M} \sum_{j=1}^{M} \left( \tilde{g}_t^{(j)} - g_t^{(j)} \right), \mathbf{u}_t - \mathbf{v} \right\rangle}_{\Xi_3}$$

$$= \frac{1}{2} \left\langle \nabla_{\mathbf{u}_t} \psi_t(\mathbf{u}_t) - \nabla_{\mathbf{q}_{t+1}} \psi_t(\mathbf{u}_{t+1}), \mathbf{u}_{t+1} - \mathbf{u}_t \right\rangle + \Xi_3$$

$$= \frac{1}{2} \left( \mathcal{D}_\psi(\mathbf{u}_{t+1}, \mathbf{q}_{t+1}) - \mathcal{D}_\psi(\mathbf{u}_{t+1}, \mathbf{u}_t) - \mathcal{D}_\psi(\mathbf{u}_t, \mathbf{q}_{t+1}) \right) + \Xi_3$$

$$\geq -\frac{1}{2} \left( \mathcal{D}_\psi(\mathbf{u}_{t+1}, \mathbf{u}_t) + \mathcal{D}_\psi(\mathbf{u}_t, \mathbf{q}_{t+1}) \right) + \Xi_3.$$

Combining the lower bound and upper bound gives

$$\frac{1}{M} \sum_{j=1}^{M} \left[ \left\langle g_t^{(j)}, \mathbf{u}_t^{(j)} - \mathbf{v} \right\rangle \right]$$

$$\leq \mathcal{D}_{\psi_t}(\mathbf{v}, \mathbf{u}_t) - \mathcal{D}_{\psi_t}(\mathbf{v}, \mathbf{u}_{t+1}) + \Xi_3 + \frac{1}{2} \mathcal{D}_\psi(\mathbf{u}_t, \mathbf{q}_{t+1}) + \frac{1}{2} \mathcal{D}_\psi(\mathbf{u}_t, \mathbf{r}_{t+1}).$$

Using the convexity of $l_t^{(j)}$, that is, $l_t^{(j)}(\mathbf{u}_t^{(j)}) - l_t^{(j)}(\mathbf{v}) \leq \left\langle g_t^{(j)}, \mathbf{p}_t^{(j)} - \mathbf{v} \right\rangle$, we further obtain

$$\frac{1}{M} \sum_{j=1}^{M} \left( l_t^{(j)}(\mathbf{u}_t^{(j)}) - l_t^{(j)}(\mathbf{v}) \right)$$

$$\leq \mathcal{D}_{\psi_t}(\mathbf{v}, \mathbf{u}_t) - \mathcal{D}_{\psi_t}(\mathbf{v}, \mathbf{u}_{t+1}) + \frac{1}{2} \mathcal{D}_\psi(\mathbf{u}_t, \mathbf{q}_{t+1}) + \frac{1}{2} \mathcal{D}_\psi(\mathbf{u}_t, \mathbf{r}_{t+1}) + \Xi_3,$$

which concludes the proof.

Now we consider the case $R < T$.

Recalling that $\mathcal{E} = \{N, 2N, 3N \dots, RN\}$ and
$$T_r = \{(r-1)N + 1, (r-1)N + 2, \dots, rN\}, \quad r = 1, \dots, R.$$

For any batch $T_r$, $r = 1, \dots, R$, we define a new loss function $\bar{l}_{rN}^{(j)}(\cdot)$ at the end of this batch,

$$\forall j \in [M], \ \forall \mathbf{u} \in \Omega, \quad \bar{l}_{rN}^{(j)}(\mathbf{u}) = \frac{1}{N} \sum_{\tau \in T_r} l_\tau^{(j)}(\mathbf{u}).$$

During each batch, our algorithmic framework does not change the decision, i.e.,

$$\forall j \in [M], \ t \in T_r, \quad \mathbf{u}_t^{(j)} = \mathbf{u}_{(r-1)N+1}^{(j)}.$$

Thus the regret can be decomposed as follows,

$$\frac{1}{M} \sum_{t=1}^{T} \sum_{j=1}^{M} \left( l_t^{(j)}(\mathbf{u}_t^{(j)}) - l_t^{(j)}(\mathbf{v}) \right) = \frac{1}{M} \sum_{r=1}^{R} \left[ \sum_{t \in T_r} \sum_{j=1}^{M} \left( l_t^{(j)}(\mathbf{u}_{(r-1)N+1}^{(j)}) - l_t^{(j)}(\mathbf{v}) \right) \right]$$

$$= \frac{N}{M} \sum_{r=1}^{R} \left[ \sum_{j=1}^{M} \sum_{t \in T_r} \frac{1}{N} \left( l_t^{(j)}(\mathbf{u}_{(r-1)N+1}^{(j)}) - l_t^{(j)}(\mathbf{v}) \right) \right]$$

$$= \frac{N}{M} \sum_{r=1}^{R} \sum_{j=1}^{M} \left( \bar{l}_{rN}^{(j)}(\mathbf{u}_{(r-1)N+1}^{(j)}) - \bar{l}_{rN}^{(j)}(\mathbf{v}) \right).$$

Now we can use FOMD-No-LU with $R = T$ to the new loss functions $\{\bar{l}_{rN}^{(1)}, \dots, \bar{l}_{rN}^{(M)}\}_{r=1,\dots,R}$, and use Theorem 1 to obtain

$$\frac{1}{NM} \sum_{t=1}^{T} \sum_{j=1}^{M} \left( l_t^{(j)}(\mathbf{u}_t^{(j)}) - l_t^{(j)}(\mathbf{v}) \right) \leq \sum_{t \in \mathcal{E}} \left[ \mathcal{D}_{\psi_t}(\mathbf{v}, \mathbf{u}_t) - \mathcal{D}_{\psi_t}(\mathbf{v}, \mathbf{u}_{t+1}) + \frac{1}{2} \mathcal{D}_{\psi_t}(\mathbf{u}_t, \mathbf{q}_{t+1}) \right] +$$

$$\frac{1}{2} \sum_{t \in \mathcal{E}} \mathcal{D}_{\psi_t}(\mathbf{u}_t, \mathbf{r}_{t+1}) + \frac{1}{M} \sum_{t \in \mathcal{E}} \sum_{j=1}^{M} \left\langle \tilde{g}_t^{(j)} - g_t^{(j)}, \mathbf{u}_t - \mathbf{v} \right\rangle,$$

which concludes the proof. □

# F    Proof of Lemma 1

**Lemma 4** (Bernstein's inequality for martingale). *Let $X_1, \ldots, X_n$ be a bounded martingale differ-ence sequence w.r.t. the filtration $\mathcal{H} = (\mathcal{H}_k)_{1 \le k \le n}$ and with $|X_k| \le a$. Let $Z_t = \sum_{k=1}^{t} X_k$ be the associated martingale. Denote the sum of the conditional variances by*

$$\Sigma_n^2 = \sum_{k=1}^{n} \mathbb{E}\left[X_k^2 | \mathcal{H}_{k-1}\right] \le v.$$

*Then for all constants $a, v > 0$, with probability at least $1 - \delta$,*

$$\max_{t=1,\ldots,n} Z_t < \frac{2}{3} a \ln \frac{1}{\delta} + \sqrt{2v \ln \frac{1}{\delta}}.$$

Note that $v$ must be a constant. Lemma 4 is derived from Lemma A.8 in [Cesa-Bianchi and Lugosi, 2006].

*Proof.* Let $v \in [0, B]$ be a random variable and $B \ge 2$ is a constant. We use the well-known peeling technique [Bartlett et al., 2005]. We divide the interval $[0, B]$ as follows

$$[0, B] \subseteq \left[0, 2^{-\lceil \log B \rceil}\right] \bigcup_{j=-\lceil \log B \rceil + 1}^{\lceil \log B \rceil} \left(2^{j-1}, 2^j\right].$$

First, we consider the case $v > 2^{-\lceil \log B \rceil}$. Let

$$\epsilon = \frac{2}{3} a \ln \frac{1}{\delta} + 2\sqrt{v \ln \frac{1}{\delta}} > \frac{2}{3} a \ln \frac{1}{\delta} + 2\sqrt{2^{-1-\log B} \ln \frac{1}{\delta}} = \frac{2}{3} a \ln \frac{1}{\delta} + \sqrt{\frac{2}{B} \ln \frac{1}{\delta}}.$$

We decompose the random event as follows,

$$\mathbb{P}\left[\max_{t=1,\ldots,n} Z_t > \epsilon, \Sigma_n^2 \le v, v > 2^{-\lceil \log B \rceil}\right]$$

$$= \mathbb{P}\left[\max_{t \le n} Z_t > \epsilon, \Sigma_n^2 \le v, \cup_{i=-\lceil \log B \rceil + 1}^{\lceil \log B \rceil} 2^{i-1} < v \le 2^i\right]$$

$$\le \sum_{i=-\lceil \log B \rceil + 1}^{\lceil \log B \rceil} \mathbb{P}\left[\max_{t \le n} Z_t > \epsilon, \Sigma_n^2 \le v, 2^{i-1} < v \le 2^i\right]$$

$$\le \sum_{i=-\lceil \log B \rceil + 1}^{\lceil \log B \rceil} \mathbb{P}\left[\max_{t \le n} Z_t > \frac{2}{3} a \ln \frac{1}{\delta} + 2\sqrt{2^{i-1} \ln \frac{1}{\delta}}, \Sigma_n^2 \le v, 2^{i-1} < v \le 2^i\right]$$

$$= \sum_{i=-\lceil \log B \rceil + 1}^{\lceil \log B \rceil} \mathbb{P}\left[\max_{t \le n} Z_t > \frac{2}{3} a \ln \frac{1}{\delta} + \sqrt{2 \cdot 2^i \ln \frac{1}{\delta}}, \Sigma_n^2 \le v, 2^{i-1} < v \le 2^i\right]$$

$$\le 2\lceil \log B \rceil \delta,$$

in which we use Lemma 4 for each sub-event.

Then we consider the case $v \le 2^{-\lceil \log B \rceil} \le \frac{1}{B}$. Lemma 4 yields, with probability at least $1 - \delta$,

$$\max_{t=1,\ldots,n} Z_t \le \frac{2}{3} a \ln \frac{1}{\delta} + \sqrt{2^{1-\lceil \log B \rceil} \ln \frac{1}{\delta}} \le \frac{2}{3} a \ln \frac{1}{\delta} + \sqrt{\frac{2}{B} \ln \frac{1}{\delta}}.$$

Combining the two cases, with probability at least $1 - 2\lceil \log B \rceil \delta$,

$$\max_{t=1,\ldots,n} Z_t \le \frac{2a}{3} \ln \frac{1}{\delta} + \sqrt{\frac{2}{B} \ln \frac{1}{\delta}} + 2\sqrt{v \ln \frac{1}{\delta}},$$

which concludes the proof. □

# G Properties of Online Mirror Descent (OMD)

## G.1 OMD with the weighted negative entropy regularizer

Let $\Omega = \Delta_K$ and $\psi_t(\mathbf{p}) = \sum_{i=1}^{K} \frac{C_i}{\eta_t} p_i \ln p_i$. Then we have

$$\forall \mathbf{p} \in \mathbb{R}^K, \quad \nabla_{p_i} \psi_t(\mathbf{p}) = \frac{C_i}{\eta_t}\left(\ln p_i + 1\right), \quad \nabla_{i,i}^2 \psi_t(\mathbf{p}) = \frac{C_i}{\eta_t p_i}.$$

The Bregman divergence associated with the negative entropy regularizer is

$$\mathcal{D}_{\psi_t}(\mathbf{p}, \mathbf{q}) = \frac{1}{\eta_t} \sum_{i=1}^{K} C_i \left( p_i \ln \frac{p_i}{q_i} + q_i - p_i \right). \tag{10}$$

Denote by $\bar{\mathbf{c}}_t = \frac{1}{M} \sum_{j=1}^{M} \tilde{\mathbf{c}}_t^{(j)}$. Recalling that the OMD is defined as follows,

$$\nabla_{\bar{\mathbf{p}}_{t+1}} \psi_t(\bar{\mathbf{p}}_{t+1}) = \nabla_{\mathbf{p}_t} \psi_t(\mathbf{p}_t) - \bar{\mathbf{c}}_t, \quad \mathbf{p}_{t+1} = \arg\min_{\mathbf{p} \in \Delta_K} \mathcal{D}_{\psi_t}(\mathbf{p}, \bar{\mathbf{p}}_{t+1}).$$

Substituting into the gradient of $\psi_t$, the mirror updating can be simplified.

$$\forall i \in [K], \quad \bar{p}_{t+1,i} = p_{t,i} \cdot \exp\left( -\frac{\eta_t \bar{c}_{t,i}}{C_i} \right).$$

Now we use the Lagrangian multiplier method to solve the projection associated with Bregman divergence.

$$L(\mathbf{p}, \lambda) = \frac{1}{\eta_t} \sum_{i=1}^{K} C_i \left( p_i \ln \frac{p_i}{\bar{p}_{t+1,i}} + \bar{p}_{t+1,i} - p_i \right) + \lambda \left( \sum_{i=1}^{K} p_i - 1 \right) - \sum_{i=1}^{K} \beta_i p_i.$$

The KKT conditions are

$$\frac{\partial L}{\partial p_i} = C_i \frac{\ln p_i + 1 - \ln \bar{p}_{t+1,i} - 1}{\eta_t} + \lambda - \beta_i = 0,$$

$$\frac{\partial L}{\partial \lambda} = \left( \sum_{i=1}^{K} p_i - 1 \right) = 0,$$

$$\beta_i p_i = 0.$$

Let $\mathbf{p}_{t+1}$, $\lambda^*$ and $\{\beta_i^*\}_{i=1}^{K}$ be the optimal solution. By the KKT conditions, we have

$$p_{t+1,i} = \bar{p}_{t+1,i} \cdot \exp\left( -\frac{\eta_t(\lambda^* - \beta_i^*)}{C_i} \right),$$

$$\sum_{i=1}^{K} \bar{p}_{t+1,i} \cdot \exp\left( -\frac{\eta_t(\lambda^* - \beta_i^*)}{C_i} \right) = \sum_{i=1}^{K} p_{t,i} \cdot \exp\left( -\frac{\eta_t(\lambda^* - \beta_i^* + \bar{c}_{t,i})}{C_i} \right) = 1,$$

$$\beta_i^* = 0,$$

in which the last equality comes from $p_{t+1,i} > 0$. The reason is that by the fact $p_{1,i} > 0$ for all $i \in [K]$, we can iteratively prove that $p_{t,i} > 0$ and $p_{t+1,i} > 0$, satisfying the condition under which $\beta_i^* = 0$. Then we can obtain the solution $\mathbf{p}_{t+1}$, i.e.,

$$\forall i \in [K], \quad p_{t+1,i} = p_{t,i} \cdot \exp\left( -\frac{\eta_t(\lambda^* + \bar{c}_{t,i})}{C_i} \right). \tag{11}$$

Next we prove that $\lambda^*$ can be found by the binary search.

If $\lambda^* \geq 0$, then $\sum_{i=1}^{K} p_{t,i} \cdot \exp\left( -\frac{\eta_t(\lambda^* + \bar{c}_{t,i})}{C_i} \right) \leq \sum_{i=1}^{K} p_{t,i} \leq 1$.

If $\lambda^* \leq -\max_i \bar{c}_{t,i}$, then $\sum_{i=1}^{K} p_{t,i} \cdot \exp\left( -\frac{\eta_t(\lambda^* + \bar{c}_{t,i})}{C_i} \right) \geq \sum_{i=1}^{K} p_{t,i} \geq 1$.

Thus it must be $-\max_i \bar{c}_{t,i} \leq \lambda^* \leq 0$. For any $0 \geq \lambda_1 \geq \lambda_2 \geq -\max_i \bar{c}_{t,i}$, we can obtain

$$\sum_{i=1}^{K} p_{t,i} \cdot \exp\left( -\frac{\eta_t(\lambda_1 + \bar{c}_{t,i})}{C_i} \right) \leq \sum_{i=1}^{K} p_{t,i} \cdot \exp\left( -\frac{\eta_t(\lambda_2 + \bar{c}_{t,i})}{C_i} \right).$$

Thus $\sum_{i=1}^{K} p_{t,i} \cdot \exp\left( -\frac{\eta_t(\lambda^* + \bar{c}_{t,i})}{C_i} \right)$ is non-increasing w.r.t. $\lambda^*$.
We can use the binary search to find $\lambda^*$.

### G.2 OMD with the Euclidean regularizer

Let $\Omega = \mathcal{F}_i$ and $\psi_{t,i}(\mathbf{w}) = \frac{1}{2\lambda_{t,i}}\|\mathbf{w}\|_2^2$. Then we have

$$\forall \mathbf{w} \in \mathbb{R}^{d_i}, \quad \nabla_{\mathbf{w}}\psi_{t,i}(\mathbf{w}) = \frac{1}{\lambda_{t,i}}\mathbf{w}, \quad \nabla_{\mathbf{w}}^2\psi_{t,i}(\mathbf{w}) = \frac{1}{\lambda_{t,i}}, \quad \mathcal{D}_{\psi_{t,i}}(\mathbf{w}, \mathbf{v}) = \frac{1}{2\lambda_{t,i}}\|\mathbf{w} - \mathbf{v}\|_2^2.$$

Recalling that the OMD is defined as follows,

$$\nabla_{\bar{\mathbf{w}}_{t+1,i}}\psi_{t,i}(\bar{\mathbf{w}}_{t+1,i}) = \nabla_{\mathbf{w}_{t,i}}\psi_t(\mathbf{w}_{t,i}) - \bar{\nabla}_{t,i}, \quad \mathbf{w}_{t+1,i} = \arg\min_{\mathbf{w} \in \mathcal{F}_i} \mathcal{D}_{\psi_{t,i}}(\mathbf{w}, \bar{\mathbf{w}}_{t+1,i}).$$

The mirror updating is as follows,

$$\forall i \in [K], \quad \bar{\mathbf{w}}_{t+1,i} = \mathbf{w}_{t,i} - \lambda_{t,i} \cdot \bar{\nabla}_{t,i},$$

$$\mathbf{w}_{t+1,i} = \min\left\{1, \frac{U_i}{\|\bar{\mathbf{w}}_{t+1,i}\|_2}\right\} \cdot \bar{\mathbf{w}}_{t+1,i}.$$

Thus OMD with the Euclidean regularizer is online gradient descent [Zinkevich, 2003].

## H Proof of Lemma 2

Recalling that $c_{t,i}^{(j)} = \ell(f_{t,i}^{(j)}(\mathbf{x}_t^{(j)}), y_t^{(j)})$, in which

$$f_{t,i}^{(j)}(\mathbf{x}_t^{(j)}) = \langle \mathbf{w}_{t,i}^{(j)}, \phi_i(\mathbf{x}_t^{(j)}) \rangle \leq U_i b_i.$$

Since $|y_t^{(j)}|$ is uniformly bounded for all $j \in [M]$ and $t \in [T]$, there is a constant $C_i$ that depends on $U_i$ and $b_i$ such that $c_{t,i}^{(j)} \leq C_i$.

Recalling that $\nabla_{t,i}^{(j)} = \ell'(f_{t,i}^{(j)}(\mathbf{x}_t^{(j)}), y_t^{(j)}) \cdot \phi_i(\mathbf{x}_t^{(j)})$. Since $\ell(f_{t,i}^{(j)}(\mathbf{x}_t^{(j)}), y_t^{(j)})$ can be upper bounded by $C_i$ and $\|\phi_i(\mathbf{x}_t^{(j)})\|_2 \leq b_i$, there is a constant $G_i$ that depends on $U_i$ and $b_i$ such that $\|\nabla_{t,i}^{(j)}\|_2 \leq G_i$.

## I Proof of Theorem 2

The regret w.r.t. any $f \in \mathcal{F}_i$ can be decomposed as follows.

$$\sum_{t=1}^{T}\sum_{j=1}^{M}\ell\left(f_{t,A_{t,1}}^{(j)}(\mathbf{x}_t^{(j)}), y_t^{(j)}\right) - \sum_{t=1}^{T}\sum_{j=1}^{M}\ell\left(f(\mathbf{x}_t^{(j)}), y_t^{(j)}\right)$$

$$=\sum_{t=1}^{T}\sum_{j=1}^{M}\left[\ell\left(f_{t,A_{t,1}}^{(j)}(\mathbf{x}_t^{(j)}), y_t^{(j)}\right) - \ell\left(f_{t,i}^{(j)}(\mathbf{x}_t^{(j)}), y_t^{(j)}\right) + \ell\left(f_{t,i}^{(j)}(\mathbf{x}_t^{(j)}), y_t^{(j)}\right) - \ell\left(f(\mathbf{x}_t^{(j)}), y_t^{(j)}\right)\right]$$

$$=\underbrace{\sum_{t=1}^{T}\sum_{j=1}^{M}\left[c_{t,A_{t,1}}^{(j)} - c_{t,i}^{(j)}\right]}_{\Xi_4} + \underbrace{\sum_{t=1}^{T}\sum_{j=1}^{M}\left[\ell\left(f_{t,i}^{(j)}(\mathbf{x}_t^{(j)}), y_t^{(j)}\right) - \ell\left(f(\mathbf{x}_t^{(j)}), y_t^{(j)}\right)\right]}_{\Xi_5}.$$

Next we separately give an upper bound on $\Xi_4$ and $\Xi_5$.

### I.1 Analyzing $\Xi_4$

We start with Lemma 1 and instantiate some notations.

$$\Omega = \Delta_K, \quad \mathbf{v} = \mathbf{v} \in \Delta_K,$$

$$\forall t \in [T], \quad g_t^{(j)} = \mathbf{c}_t^{(j)}, \quad \tilde{g}_t^{(j)} = \tilde{\mathbf{c}}_t^{(j)}, \quad \bar{g}_t = \bar{\mathbf{c}}_t, \quad \mathbf{u}_t^{(j)} = \mathbf{p}_t^{(j)}, \quad \mathbf{u}_t = \mathbf{p}_t,$$

$$l_t^j(\mathbf{u}_t^j) = \left\langle \mathbf{c}_t^{(j)}, \mathbf{p}_t^{(j)} \right\rangle, \quad l_t^j(\mathbf{v}) = \left\langle \mathbf{c}_t^{(j)}, \mathbf{v} \right\rangle.$$

Lemma 1 gives, $\forall \mathbf{v} \in \Delta_K$,

$$\frac{1}{M}\sum_{t=1}^{T}\sum_{j=1}^{M}\left\langle \mathbf{c}_t^{(j)}, \mathbf{p}_t^{(j)} - \mathbf{v}\right\rangle$$

$$\leq \sum_{t=1}^{T}\left(\mathcal{D}_{\psi_t}(\mathbf{v}, \mathbf{p}_t) - \mathcal{D}_{\psi_t}(\mathbf{v}, \mathbf{p}_{t+1})\right) + \frac{1}{2}\sum_{t=1}^{T}\mathcal{D}_{\psi_t}(\mathbf{p}_t, \mathbf{q}_{t+1}) + \frac{1}{2}\sum_{t=1}^{T}\mathcal{D}_{\psi_t}(\mathbf{p}_t, \mathbf{r}_{t+1}) + \quad (12)$$

$$\frac{1}{M}\sum_{t=1}^{T}\sum_{j=1}^{M}\left\langle \tilde{\mathbf{c}}_t^{(j)} - \mathbf{c}_t^{(j)}, \mathbf{p}_t - \mathbf{v}\right\rangle.$$

Recalling that

$$\nabla_{\mathbf{q}_{t+1}}\psi_t(\mathbf{q}_{t+1}) = \nabla_{\mathbf{u}_t}\psi_t(\mathbf{u}_t) - 2\sum_{j=1}^{M}\frac{\tilde{g}_t^{(j)} - g_t^{(j)}}{M}, \tag{13}$$

$$\nabla_{\mathbf{r}_{t+1}}\psi_t(\mathbf{r}_{t+1}) = \nabla_{\mathbf{u}_t}\psi_t(\mathbf{u}_t) - \frac{2}{M}\sum_{j=1}^{M}g_t^{(j)}. \tag{14}$$

In (13), we redefine $\Omega = \Delta_K$ and $\psi_t(\mathbf{p}) = \sum_{i=1}^{K}\frac{C_i}{\eta_t}p_i \ln p_i$, and in (14), we redefine $\Omega = \Delta_K$ and $\psi_t(\mathbf{p}) = \sum_{i=1}^{K}\frac{2C_i}{\eta_t}p_i \ln p_i$. Using the results in Section G.1, we can obtain

$$\forall i \in [K], \quad q_{t+1,i} = p_{t,i}\exp\left(-\frac{\eta_t \delta_{t,i}}{C_i}\right), \quad \delta_{t,i} = \frac{2}{M}\sum_{j=1}^{M}\left(\tilde{c}_{t,i}^{(j)} - c_{t,i}^{(j)}\right),$$

$$r_{t+1,i} = p_{t,i}\exp\left(-\frac{\eta_t \hat{c}_{t,i}}{2C_i}\right), \quad \hat{c}_{t,i} = \frac{2}{M}\sum_{j=1}^{M}c_{t,i}^{(j)}. \tag{15}$$

It can be verified that $\delta_{t,i} \in [-2C_i, 2\frac{K-J}{J-1}C_i]$ and $\hat{c}_{t,i} \in [0, 2C_i]$.

Recalling the definition of learning rate $\eta_t$ in Theorem 2. We can obtain $\frac{\eta_t \delta_{t,i}}{C_i} \geq -1$ and $\frac{\eta_t \hat{c}_{t,i}}{2C_i} \geq -1$. Next we use (10) and (15) to analyze the following two Bregman divergences.

$$\sum_{t=1}^{T}\mathcal{D}_{\psi_t}(\mathbf{p}_t, \mathbf{r}_{t+1}) = \sum_{t=1}^{T}\frac{1}{\eta_t}\sum_{i=1}^{K}2C_i \cdot \left(p_{t,i}\ln\frac{p_{t,i}}{r_{t+1,i}} + r_{t+1,i} - p_{t,i}\right)$$

$$= \sum_{t=1}^{T}\frac{1}{\eta_t}\sum_{i=1}^{K}2C_i \cdot \left(\frac{p_{t,i}\eta_t \hat{c}_{t,i}}{2C_i} + p_{t,i}\cdot\exp\left(-\frac{\eta_t \hat{c}_{t,i}}{2C_i}\right) - p_{t,i}\right)$$

$$\leq \sum_{t=1}^{T}\frac{1}{\eta_t}\sum_{i=1}^{K}2C_i \cdot \left(\frac{p_{t,i}\eta_t \hat{c}_{t,i}}{2C_i} + p_{t,i}\cdot\left(1 - \frac{\eta_t \hat{c}_{t,i}}{2C_i} + \left(\frac{\eta_t \hat{c}_{t,i}}{2C_i}\right)^2\right) - p_{t,i}\right)$$

$$\leq \sum_{t=1}^{T}\eta_t\sum_{i=1}^{K}\frac{p_{t,i}}{2C_i}\left(\frac{2}{M}\sum_{j=1}^{M}c_{t,i}^{(j)}\right)^2$$

$$\leq 2\sum_{t=1}^{T}\eta_t \cdot \frac{1}{M}\sum_{j=1}^{M}\sum_{i=1}^{K}p_{t,i}c_{t,i}^{(j)},$$

and

$$\sum_{t=1}^{T} \mathcal{D}_{\psi_t}(\mathbf{p}_t, \mathbf{q}_{t+1}) = \sum_{t=1}^{T} \frac{1}{\eta_t} \sum_{i=1}^{K} C_i \left( p_{t,i} \ln \frac{p_{t,i}}{q_{t+1,i}} + q_{t+1,i} - p_{t,i} \right)$$

$$= \sum_{t=1}^{T} \frac{1}{\eta_t} \sum_{i=1}^{K} C_i \left( \frac{p_{t,i} \eta_t \delta_{t,i}}{C_i} + p_{t,i} \cdot \exp \left( -\frac{\eta_t \delta_{t,i}}{C_i} \right) - p_{t,i} \right)$$

$$\leq 4 \sum_{t=1}^{T} \eta_t \sum_{i=1}^{K} \frac{p_{t,i}}{C_i} \left( \frac{1}{M} \sum_{j=1}^{M} \left( \tilde{c}_{t,i}^{(j)} - c_{t,i}^{(j)} \right) \right)^2,$$

in where we use the fact $\exp(-x) \leq 1 - x + x^2$ for all $x \geq -1$.

Substituting the two upper bounds into (12) gives

$$\forall \mathbf{v} \in \Delta_K, \quad \underbrace{\frac{1}{M} \sum_{t=1}^{T} \sum_{j=1}^{M} \left\langle \mathbf{c}_t^{(j)}, \mathbf{p}_t^{(j)} - \mathbf{v} \right\rangle}_{\Xi_{4,1}}$$

$$\leq \underbrace{\sum_{t=1}^{T} \left( \mathcal{D}_{\psi_t}(\mathbf{v}, \mathbf{p}_t) - \mathcal{D}_{\psi_t}(\mathbf{v}, \mathbf{p}_{t+1}) \right)}_{\Xi_{4,2}} + \underbrace{2 \sum_{t=1}^{T} \eta \sum_{i=1}^{K} \frac{p_{t,i}}{C_i} \left( \frac{1}{M} \sum_{j=1}^{M} \left( \tilde{c}_{t,i}^{(j)} - c_{t,i}^{(j)} \right) \right)^2}_{\Xi_{4,3}} +$$

$$\sum_{t=1}^{T} \frac{\eta}{M} \sum_{j=1}^{M} \sum_{i=1}^{K} p_{t,i} c_{t,i}^{(j)} + \underbrace{\frac{1}{M} \sum_{t=1}^{T} \sum_{j=1}^{M} \left\langle \tilde{\mathbf{c}}_t^{(j)} - \mathbf{c}_t^{(j)}, \mathbf{p}_t - \mathbf{v} \right\rangle}_{\Xi_{4,4}}.$$

**Bounding $\Xi_{4,1}$**

We define a random variable $X_t$ as follows,

$$X_t = c_{t,A_{t,1}}^{(j)} - \left\langle \mathbf{c}_t^{(j)}, \mathbf{p}_t^{(j)} \right\rangle.$$

Let $H_t = \{O_t^{(1)}, \ldots, O_t^{(M)}\}$. Then we have $\mathbb{E}[X_t | H_{[t-1]}] = 0$ and $|X_t| \leq C$ where $C = \max_i C_i$. Thus $X_{[T]}$ is a bounded martingale difference sequence w.r.t. the filtration $H_{[T]}$. The sum of condition variance satisfies

$$\sum_{t=1}^{T} \mathbb{E}\left[ |X_t|^2 | H_{[t-1]} \right] \leq \sum_{t=1}^{T} \mathbb{E}\left[ \left| c_{t,A_{t,1}}^{(j)} \right|^2 | H_{[t-1]} \right] \leq C \cdot \sum_{t=1}^{T} \left\langle \mathbf{c}_t^{(j)}, \mathbf{p}_t^{(j)} \right\rangle \leq C^2 T.$$

The upper bound is a random variable. Lemma 1 yields, with probability at least $1 - M \log(C^2 T)\delta$,

$$\Xi_{4,1} \geq \sum_{t=1}^{T} \sum_{j=1}^{M} c_{t,A_{t,1}}^{(j)} - \sum_{t=1}^{T} \sum_{j=1}^{M} \left\langle \mathbf{c}_t^{(j)}, \mathbf{v} \right\rangle - \frac{2CM}{3} \ln \frac{1}{\delta} - 2 \sqrt{CM \cdot \sum_{j=1}^{M} \sum_{t=1}^{T} \left\langle \mathbf{c}_t^{(j)}, \mathbf{p}_t^{(j)} \right\rangle \cdot \ln \frac{1}{\delta}},$$

where the fail probability comes from the union-of-events, and the lower order term $M \sqrt{\frac{2}{C^2 T} \ln \frac{1}{\delta}}$ is omitted.

**Bounding $\Xi_{4,2}$**

According to (10), we have

$$\Xi_{4,2} \leq \mathcal{D}_{\psi_1}(\mathbf{v}, \mathbf{p}_1) = \frac{1}{\eta} \sum_{i=1}^{K} C_i \left( v_i \ln \frac{v_i}{p_{1,i}} + p_{1,i} - v_i \right) \leq \frac{C_i}{\eta} \ln \frac{1}{p_{1,i}} + \frac{1}{\eta} \sum_{k=1}^{K} C_k p_{1,k} - \frac{C_i}{\eta}.$$

**Bounding $\Xi_{4,3}$**

We define a random variable $X_t$ as follows,

$$X_t = \sum_{i=1}^{K} \frac{p_{t,i}}{C_i} \left( \frac{1}{M} \sum_{j=1}^{M} \left( \tilde{c}_{t,i}^{(j)} - c_{t,i}^{(j)} \right) \right)^2 - \mathbb{E}_t \left[ \sum_{i=1}^{K} \frac{p_{t,i}}{C_i} \left( \frac{1}{M} \sum_{j=1}^{M} \left( \tilde{c}_{t,i}^{(j)} - c_{t,i}^{(j)} \right) \right)^2 \right].$$

It can be verified that $\mathbb{E}[X_t | H_{[t-1]}] = 0$ and $|X_t| \leq \frac{K-J}{J-1} C$. Next we upper bound the sum of condition variance.

$$\sum_{t=1}^{T} \mathbb{E}_t[X_t^2] \leq \sum_{t=1}^{T} \mathbb{E}_t \left[ \left( \sum_{i=1}^{K} \frac{p_{t,i}}{C_i} \left( \frac{1}{M} \sum_{j=1}^{M} \left( \tilde{c}_{t,i}^{(j)} - c_{t,i}^{(j)} \right) \right)^2 \right)^2 \right]$$

$$\leq \sum_{t=1}^{T} \mathbb{E}_t \left[ \sum_{i=1}^{K} p_{t,i} \left( \frac{1}{C_i} \left( \frac{1}{M} \sum_{j=1}^{M} \left( \tilde{c}_{t,i}^{(j)} - c_{t,i}^{(j)} \right) \right)^2 \right)^2 \right]$$

$$\leq \frac{(K-J)^2}{(J-1)^2} \sum_{t=1}^{T} \mathbb{E}_t \left[ \sum_{i=1}^{K} p_{t,i} \left( \frac{1}{M} \sum_{j=1}^{M} \left( \tilde{c}_{t,i}^{(j)} - c_{t,i}^{(j)} \right) \right)^2 \right]$$

$$= \frac{(K-J)^2}{(J-1)^2} \frac{1}{M^2} \sum_{t=1}^{T} \sum_{i=1}^{K} p_{t,i} \mathbb{E}_t \left[ \sum_{j=1}^{M} \left( \tilde{c}_{t,i}^{(j)} - c_{t,i}^{(j)} \right)^2 \right]$$

$$\leq \frac{(K-J)^3}{(J-1)^3 M^2} C \cdot \sum_{t=1}^{T} \sum_{j=1}^{M} \left\langle \mathbf{c}_t^{(j)}, \mathbf{p}_t^{(j)} \right\rangle$$

$$\leq \frac{(K-J)^3}{(J-1)^3 M} C^2 T,$$

where we use the fact $\tilde{c}_{t,i}^{(j)} = \frac{c_{t,i}^{(j)}}{\mathbb{P}\left[i \in O_t^{(j)}\right]} \geq \frac{K-1}{J-1} c_{t,i}^{(j)}$. Lemma 1 yields, with probability at least $1 - \log(C^2 K^3 T / M)\delta$,

$$\Xi_{4,3} \leq \eta \left( \frac{g_{K,J}}{M^2} \sum_{t=1}^{T} \sum_{j=1}^{M} \left\langle \mathbf{c}_t^{(j)}, \mathbf{p}_t^{(j)} \right\rangle + \frac{2C g_{K,J}}{3} \ln \frac{1}{\delta} + 2 \sqrt{\frac{g_{K,J}^3}{M^2} C \sum_{t=1}^{T} \sum_{j=1}^{M} \left\langle \mathbf{c}_t^{(j)}, \mathbf{p}_t^{(j)} \right\rangle \cdot \ln \frac{1}{\delta}} \right),$$

where $g_{K,J} = \frac{K-J}{J-1}$ and the lower order term $\sqrt{\frac{2M}{g_{K,J}^3 C^2 T} \ln \frac{1}{\delta}}$ is omitted.

**Bounding $\Xi_{4,4}$**

We define a random variable $X_t$ as follows,

$$X_t = \left\langle \frac{1}{M} \sum_{j=1}^{M} \left( \tilde{\mathbf{c}}_t^{(j)} - \mathbf{c}_t^{(j)} \right), \mathbf{p}_t - \mathbf{v} \right\rangle = \frac{1}{M} \sum_{j=1}^{M} \left( \sum_{i=1}^{K} (p_{t,i} - v_i) \left( \tilde{c}_{t,i}^{(j)} - c_{t,i}^{(j)} \right) \right).$$

$\{X_t\}_{t=1}^T$ is a bounded martingale difference sequence and $|X_t| \le g_{K,J}C$. We further have

$$
\sum_{t=1}^T \mathbb{E}_t[X_t^2]
$$

$$
= \frac{1}{M^2} \sum_{t=1}^T \mathbb{E}_t \left[ \sum_{j=1}^M \left( \sum_{i=1}^K (p_{t,i} - v_i) \left( \tilde{c}_{t,i}^{(j)} - c_{t,i}^{(j)} \right) \right)^2 \right] +
$$

$$
\frac{1}{M^2} \sum_{t=1}^T \mathbb{E}_t \left[ \sum_{j \neq r} \left( \sum_{i=1}^K (p_{t,i} - v_i) \left( \tilde{c}_{t,i}^{(j)} - c_{t,i}^{(j)} \right) \right) \left( \sum_{i=1}^K (p_{t,i} - v_i) \left( \tilde{c}_{t,i}^{(r)} - c_{t,i}^{(r)} \right) \right) \right]
$$

$$
= \frac{1}{M^2} \sum_{t=1}^T \sum_{j=1}^M \mathbb{E}_t \left[ \left( \sum_{i=1}^K (p_{t,i} - v_i) \left( \tilde{c}_{t,i}^{(j)} - c_{t,i}^{(j)} \right) \right)^2 \right]
$$

$$
= \frac{2}{M^2} \sum_{t=1}^T \sum_{j=1}^M \mathbb{E}_t \left[ \left( \sum_{i=1}^K p_{t,i} \left( \tilde{c}_{t,i}^{(j)} - c_{t,i}^{(j)} \right) \right)^2 \right] + \frac{2}{M^2} \sum_{t=1}^T \sum_{j=1}^M \mathbb{E}_t \left[ \left( \sum_{i=1}^K v_i \left( \tilde{c}_{t,i}^{(j)} - c_{t,i}^{(j)} \right) \right)^2 \right]
$$

$$
\le \frac{2}{M^2} \sum_{t=1}^T \sum_{j=1}^M \mathbb{E}_t \left[ \sum_{i=1}^K p_{t,i} \left( \tilde{c}_{t,i}^{(j)} - c_{t,i}^{(j)} \right)^2 \right] + \frac{2}{M^2} \sum_{t=1}^T \sum_{j=1}^M \mathbb{E}_t \left[ \sum_{i=1}^K v_i \left( \tilde{c}_{t,i}^{(j)} - c_{t,i}^{(j)} \right)^2 \right]
$$

$$
\le 2 \frac{g_{K,J}}{M^2} C \cdot \sum_{t=1}^T \sum_{j=1}^M \left\langle \mathbf{c}_t^{(j)}, \mathbf{p}_t^{(j)} \right\rangle + 2 \frac{g_{K,J}}{M^2} \cdot \sum_{t=1}^T \sum_{j=1}^M \left\langle \mathbf{c}_t^{(j)} \otimes \mathbf{c}_t^{(j)}, \mathbf{v} \right\rangle
$$

$$
\le \frac{4C^2 KT}{M},
$$

where $\left\langle \mathbf{c}_t^{(j)} \otimes \mathbf{c}_t^{(j)}, \mathbf{v} \right\rangle = \sum_{i=1}^K v_i (c_{t,i}^{(j)})^2$.

With probability at least $1 - \log(4C^2 KT/M)\delta$,

$$
\Xi_{4,4} \le \frac{2Cg_{K,J}}{3} \ln \frac{1}{\delta} + 2\sqrt{2\frac{g_{K,J}}{M^2} \ln \frac{1}{\delta}} \cdot \sqrt{C \sum_{t=1}^T \sum_{j=1}^M \left\langle \mathbf{c}_t^{(j)}, \mathbf{p}_t^{(j)} \right\rangle + \sum_{t=1}^T \sum_{j=1}^M \left\langle \mathbf{c}_t^{(j)} \otimes \mathbf{c}_t^{(j)}, \mathbf{v} \right\rangle}.
$$

For simplicity, we introduce some new notations

$$
\bar{L}_T = \sum_{t=1}^T \sum_{j=1}^M \left\langle \mathbf{c}_t^{(j)}, \mathbf{p}_t^{(j)} \right\rangle, \quad \bar{L}_T(\mathbf{v}) = \sum_{t=1}^T \sum_{j=1}^M \left\langle \mathbf{c}_t^{(j)}, \mathbf{v} \right\rangle, \quad \tilde{L}_T(\mathbf{v}) = \sum_{t=1}^T \sum_{j=1}^M \left\langle \mathbf{c}_t^{(j)} \otimes \mathbf{c}_t^{(j)}, \mathbf{v} \right\rangle.
$$

**Combining all**

Combining all gives, with probability at least $1 - \Theta(\log(CKT/M)) \cdot \delta$,

$$
\bar{L}_T - \bar{L}_T(\mathbf{v}) \le \frac{M}{\eta} \left( C_i \ln \frac{1}{p_{1,i}} + \sum_{k=1}^K C_k p_{1,k} - C_i \right) +
$$

$$
\eta \left( \left( 1 + \frac{g_{K,J}}{M} \right) \bar{L}_T + \frac{2MC}{3} g_{K,J} \ln \frac{1}{\delta} + 2\sqrt{g_{K,J}^3 C \cdot \bar{L}_T \cdot \ln \frac{1}{\delta}} \right) +
$$

$$
\frac{2MCg_{K,J}}{3} \ln \frac{1}{\delta} + 2\sqrt{2g_{K,J} \ln \frac{1}{\delta}} \cdot \sqrt{C\bar{L}_T + \tilde{L}_T(\mathbf{v})}.
$$

Rearranging terms gives

$$\left(1 - \eta\left(1 + \frac{g_{K,J}}{M}\right)\right)\bar{L}_T - \left(2\eta\sqrt{g_{K,J}^3 C \ln\frac{1}{\delta}} + 2\sqrt{2g_{K,J}C\ln\frac{1}{\delta}}\right)\sqrt{\bar{L}_T}$$

$$\leq \bar{L}_T(\mathbf{v}) + \frac{M}{\eta}\left(C_i \ln\frac{1}{p_{1,i}} + \sum_{k=1}^{K} C_k p_{1,k} - C_i\right) + \frac{4MCg_{K,J}}{3}\ln\frac{1}{\delta} + 2\sqrt{2g_{K,J}\ln\frac{1}{\delta}} \cdot \sqrt{\tilde{L}_T(\mathbf{v})}.$$

Recalling that, the solution of the following inequality

$$x - a\sqrt{x} - b \leq 0, x > 0, a > 0, b > 0,$$

is $x \leq a^2 + b + a\sqrt{b}$. Solving for $\bar{L}_T$ gives

$$\bar{L}_T - \bar{L}_T(\mathbf{v}) \leq \frac{\left(2\eta\sqrt{g_{K,J}^3 C\ln\frac{1}{\delta}} + 2\sqrt{2g_{K,J}C\ln\frac{1}{\delta}}\right)^2}{\left(1 - \eta\left(1 + \frac{g_{K,J}}{M}\right)\right)^2} + \frac{2\eta\sqrt{g_{K,J}^3 C\ln\frac{1}{\delta}} + 2\sqrt{2g_{K,J}C\ln\frac{1}{\delta}}}{\left(1 - \eta\left(1 + \frac{g_{K,J}}{M}\right)\right)^{\frac{3}{2}}} \cdot$$

$$\sqrt{\bar{L}_T(\mathbf{v}) + \frac{M}{\eta}\left(C_i\ln\frac{1}{p_{1,i}} + \sum_{k=1}^{K} C_k p_{1,k} - C_i\right) + \frac{4MCg_{K,J}}{3}\ln\frac{1}{\delta} + 2\sqrt{2g_{K,J}\ln\frac{1}{\delta}} \cdot \sqrt{\tilde{L}_T(\mathbf{v})} +}$$

$$\frac{\eta\left(1 + \frac{g_{K,J}}{M}\right)}{1 - \eta\left(1 + \frac{g_{K,J}}{M}\right)}\bar{L}_T(\mathbf{v}) +$$

$$\frac{\frac{M}{\eta}\left(C_i\ln\frac{1}{p_{1,i}} + \sum_{k=1}^{K} C_k p_{1,k} - C_i\right) + \frac{4MCg_{K,J}}{3}\ln\frac{1}{\delta} + 2\sqrt{2g_{K,J}\ln\frac{1}{\delta}} \cdot \sqrt{\tilde{L}_T(\mathbf{v})}}{1 - \eta\left(1 + \frac{g_{K,J}}{M}\right)}.$$

Denote by $A_m = \operatorname{argmin}_{i\in[K]} C_i$. Let the learning rate and initial distribution $\mathbf{p}_1$ satisfy

$$\eta = \frac{\sqrt{\ln(KT)}}{2\sqrt{\left(1 + \frac{K-J}{(J-1)M}\right)T}} \wedge \frac{J-1}{2(K-J)},$$

$$p_{1,k} = \left(1 - \frac{\sqrt{K}}{\sqrt{T}}\right)\frac{1}{|A_m|} + \frac{1}{\sqrt{KT}}, k \in A_m, \quad p_{1,j} = \frac{1}{\sqrt{KT}}, j \neq A_m.$$

Then we have

$$C_i\ln\frac{1}{p_{1,i}} + \sum_{k=1}^{K} C_k p_{1,k} - C_i$$

$$\leq C_i\ln(\sqrt{KT}) + \frac{C\cdot(K-|A_m|)}{\sqrt{KT}} + \min_i C_i \cdot |A_m| \cdot \left(\left(1 - \frac{\sqrt{K}}{\sqrt{T}}\right)\frac{1}{|A_m|} + \frac{1}{\sqrt{KT}}\right) - C_i$$

$$\leq C_i\ln(\sqrt{KT}) + \frac{C\sqrt{K}}{\sqrt{T}}.$$

We further simplify $\bar{L}_T - \bar{L}_T(\mathbf{v})$.

$\bar{L}_T - \bar{L}_T(\mathbf{v})$

$\leq 64 g_{K,J} C \ln \frac{1}{\delta} + 11 \sqrt{g_{K,J} C \ln \frac{1}{\delta}} \cdot$

$$\sqrt{C_i TM + \frac{M}{\eta}\left(C_i \ln(\sqrt{KT}) + \frac{C\sqrt{K}}{\sqrt{T}}\right) + \frac{4MCg_{K,J}}{3}\ln\frac{1}{\delta} + 2\sqrt{2g_{K,J}\ln\frac{1}{\delta}} \cdot \sqrt{\tilde{L}_T(\mathbf{v})} +}$$

$$2\eta\left(1 + \frac{g_{K,J}}{M}\right)C_i T + \frac{\frac{M}{\eta}\left(C_i \ln(\sqrt{KT}) + \frac{C\sqrt{K}}{\sqrt{T}}\right) + \frac{4MCg_{K,J}}{3}\ln\frac{1}{\delta} + 2C_i\sqrt{2g_{K,J}\ln\frac{1}{\delta}}\cdot\sqrt{TM}}{\frac{1}{2}}$$

$$\leq \underbrace{(64 + 3M)g_{K,J}C\ln\frac{1}{\delta} + 17\sqrt{Mg_{K,J}CC_i T \ln\frac{1}{\delta}} + \frac{4}{\sqrt{2}}C_i M\sqrt{\left(1 + \frac{K-J}{(J-1)M}\right)T\ln(KT)}}_{\Xi_{4,5}},$$

in which we omit the lower order terms such as $O(T^{\frac{1}{4}})$ and $O(\sqrt{g_{K,J}C\ln\frac{1}{\delta}})$.

Finally, using the upper bound on $\Xi_{4,1}$ gives, with probability at least $1 - \Theta(M\log(CT) + \log(CKT/M))\cdot\delta$,

$\Xi_4$

$$\leq \bar{L}_T - \bar{L}_T(\mathbf{v}) + \frac{2CM}{3}\ln\frac{1}{\delta} + 2\sqrt{CM \cdot \sum_{j=1}^{M}\sum_{t=1}^{T}\left\langle \mathbf{c}_t^{(j)}, \mathbf{p}_t^{(j)}\right\rangle \cdot \ln\frac{1}{\delta}}$$

$$\leq \bar{L}_T - \bar{L}_T(\mathbf{v}) + \frac{2CM}{3}\ln\frac{1}{\delta} + 2\sqrt{CM\cdot\left(\bar{L}_T(\mathbf{v}) + \Xi_{4,5}\right)\cdot\ln\frac{1}{\delta}}$$

$$\leq (64 + 3M)g_{K,J}C\ln\frac{1}{\delta} + 17\sqrt{Mg_{K,J}CC_i T \ln\frac{1}{\delta}} + \frac{4}{\sqrt{2}}C_i M\sqrt{\left(1 + \frac{K-J}{(J-1)M}\right)T\ln(KT)} +$$

$$2M\sqrt{CC_i T \ln\frac{1}{\delta}},$$

where we omit $O(\sqrt{CM\Xi_{4,5}\cdot\ln\frac{1}{\delta}})$ which is a lower order term.

## I.2 Analyzing $\Xi_5$

We also start with Lemma 1.

We just a fixed $i \in \mathcal{F}_i$. We instantiate some notations.

$$\Omega = \mathcal{F}_i, \quad \mathbf{v} = \mathbf{w} \in \mathcal{F}_i,$$

$$\forall t \in [T], \quad g_t^{(j)} = \nabla_{t,i}^{(j)}, \quad \tilde{g}_t^{(j)} = \tilde{\nabla}_{t,i}^{(j)}, \quad \bar{g}_t = \bar{\nabla}_{t,i}, \quad \mathbf{u}_t^{(j)} = \mathbf{w}_{t,i}^{(j)}, \quad \mathbf{u}_t = \mathbf{w}_t,$$

$$l_t^j(\mathbf{u}_t^j) = \ell\left(f_{t,i}^{(j)}(\mathbf{x}_t^{(j)}), y_t^{(j)}\right), \quad l_t^j(\mathbf{v}) = \ell\left(f(\mathbf{x}_t^{(j)}), y_t^{(j)}\right).$$

Lemma 1 gives

$$\forall \mathbf{w} \in \mathcal{F}_i, \quad \frac{1}{M}\sum_{t=1}^{T}\sum_{j=1}^{M}\left[\ell\left(f_{t,i}^{(j)}(\mathbf{x}_t^{(j)}), y_t^{(j)}\right) - \ell\left(f(\mathbf{x}_t^{(j)}), y_t^{(j)}\right)\right]$$

$$\leq \sum_{t=1}^{T}\left(\mathcal{D}_{\psi_{t,i}}(\mathbf{w}, \mathbf{w}_t) - \mathcal{D}_{\psi_t}(\mathbf{w}, \mathbf{w}_{t+1})\right) + \frac{1}{2}\sum_{t=1}^{T}\mathcal{D}_{\psi_{t,i}}(\mathbf{w}_t, \mathbf{q}_{t+1}) +$$

$$\frac{1}{2}\sum_{t=1}^{T}\mathcal{D}_{\psi_{t,i}}(\mathbf{w}_t, \mathbf{r}_{t+1}) + \frac{1}{M}\sum_{t=1}^{T}\sum_{j=1}^{M}\left\langle\tilde{\nabla}_{t,i}^{(j)} - \nabla_{t,i}^{(j)}, \mathbf{w}_t - \mathbf{w}\right\rangle,$$

where the Bregman divergence is

$$\mathcal{D}_{\psi_{t,i}}(\mathbf{w}, \mathbf{v}) = \frac{1}{2\lambda_{t,i}} \|\mathbf{w} - \mathbf{v}\|_2^2.$$

Besides, (13) and (14) can be instantiated as follows

$$\mathbf{q}_{t+1} = \mathbf{w}_t - \lambda_{t,i} \cdot \frac{2}{M} \sum_{j=1}^M \left( \tilde{\nabla}_{t,i}^{(j)} - \nabla_{t,i}^{(j)} \right),$$

$$\mathbf{r}_{t+1} = \mathbf{w}_t - \lambda_{t,i} \cdot \frac{2}{M} \sum_{j=1}^M \nabla_{t,i}^{(j)}.$$

Thus we have: $\forall \mathbf{w} \in \mathcal{F}_i$,

$$\frac{1}{M} \Xi_5$$

$$\leq \sum_{t=1}^T \frac{\|\mathbf{w} - \mathbf{w}_t\|_2^2 - \|\mathbf{w} - \mathbf{w}_{t+1}\|_2^2}{2\lambda_{t,i}} + 2\sum_{t=1}^T \lambda_{t,i} \left\| \frac{1}{M} \sum_{j=1}^M \left( \tilde{\nabla}_{t,i}^{(j)} - \nabla_{t,i}^{(j)} \right) \right\|_2^2 +$$

$$2\sum_{t=1}^T \lambda_{t,i} \left\| \frac{1}{M} \sum_{j=1}^M \nabla_{t,i}^{(j)} \right\|_2^2 + \frac{1}{M} \sum_{t=1}^T \sum_{j=1}^M \left\langle \tilde{\nabla}_{t,i}^{(j)} - \nabla_{t,i}^{(j)}, \mathbf{w}_t - \mathbf{w} \right\rangle$$

$$\leq \frac{2U_i^2}{\lambda_{T,i}} + 2G_i^2 \sum_{t=1}^T \lambda_{t,i} + \underbrace{2\sum_{t=1}^T \lambda_{t,i} \left\| \sum_{j=1}^M \frac{\tilde{\nabla}_{t,i}^{(j)} - \nabla_{t,i}^{(j)}}{M} \right\|_2^2}_{\Xi_{5,1}} + \underbrace{\frac{1}{M} \sum_{t=1}^T \sum_{j=1}^M \left\langle \tilde{\nabla}_{t,i}^{(j)} - \nabla_{t,i}^{(j)}, \mathbf{w}_t - \mathbf{w} \right\rangle}_{\Xi_{5,2}}.$$

Next we separately give a high-probability upper bound on $\Xi_{4,1}$ and $\Xi_{4,2}$.

**Bounding $\Xi_{5,2}$**

We define a random variable $X_t$ as follows,

$$X_t = \left\langle \frac{1}{M} \sum_{j=1}^M \left( \tilde{\nabla}_{t,i}^{(j)} - \nabla_{t,i}^{(j)} \right), \mathbf{w}_t - \mathbf{w} \right\rangle.$$

$X_{[T]}$ is a bounded martingale difference sequence w.r.t. $H_{[T]}$ and $|X_t| \leq 2g_{K,J}G_iU_i$. We further have

$$\sum_{t=1}^T \mathbb{E}_t[|X_t|^2] \leq \sum_{t=1}^T 4U_i^2 \mathbb{E}_t \left[ \left\| \frac{1}{M} \sum_{j=1}^M \left( \tilde{\nabla}_{t,i}^{(j)} - \nabla_{t,i}^{(j)} \right) \right\|_2^2 \right] \leq 4U_i^2 G_i^2 \frac{g_{K,J}}{M} T.$$

The upper bound on the sum of conditional variance is a constant. Lemma 4 gives, with probability at least $1 - \delta$,

$$\Xi_{5,2} \leq \frac{4G_iU_ig_{K,J}}{3} \ln \frac{1}{\delta} + 2G_iU_i \sqrt{2\frac{g_{K,J}}{M}T \ln \frac{1}{\delta}}.$$

**Bounding $\Xi_{5,1}$**

Recalling that

$$\lambda_{t,i} = \begin{cases} \frac{U_i}{2G_i\sqrt{\left(1 + \frac{g_{K,J}}{M}\right) \cdot g_{K,J}^2}} & \text{if } t \leq g_{K,J}^2, \\ \frac{U_i}{2G_i\sqrt{\left(1 + \frac{g_{K,J}}{M}\right) \cdot t}} & \text{otherwise.} \end{cases}$$

It can be found that $\lambda_{t,i} \le \frac{U_i}{2g_{K,J}G_i}$.

**Case 1**: $T > g_{K,J}^2$.
We decompose $\Xi_{5,1}$ as follows,

$$\Xi_{5,1} = \underbrace{\sum_{t=1}^{g_{K,J}^2} \lambda_{t,i} \left\| \frac{1}{M} \sum_{j=1}^{M} \left( \tilde{\nabla}_{t,i}^{(j)} - \nabla_{t,i}^{(j)} \right) \right\|_2^2}_{\Xi_{5,1,1}} + \underbrace{\sum_{t=g_{K,J}^2+1}^{T} \lambda_{t,i} \left\| \frac{1}{M} \sum_{j=1}^{M} \left( \tilde{\nabla}_{t,i}^{(j)} - \nabla_{t,i}^{(j)} \right) \right\|_2^2}_{\Xi_{5,1,2}}.$$

We separately analyze $\Xi_{5,1,1}$ and $\Xi_{5,1,2}$. Let

$$X_t = \lambda_{t,i} \left\| \frac{1}{M} \sum_{j=1}^{M} \left( \tilde{\nabla}_{t,i}^{(j)} - \nabla_{t,i}^{(j)} \right) \right\|_2^2 - \lambda_{t,i} \mathbb{E}_t \left[ \left\| \frac{1}{M} \sum_{j=1}^{M} \left( \tilde{\nabla}_{t,i}^{(j)} - \nabla_{t,i}^{(j)} \right) \right\|_2^2 \right].$$

$X_{[T]}$ is a martingale difference sequence and satisfies $|X_t| \le \lambda_{t,i} \cdot g_{K,J}^2 G_i^2 \le \frac{g_{K,J}U_iG_i}{2}$.
We further have

$$\sum_{t=1}^{g_{K,J}^2} \mathbb{E}_t[|X_t|^2] \le \sum_{t=1}^{g_{K,J}^2} \mathbb{E}_t \left[ \lambda_{t,i}^2 \left\| \frac{1}{M} \sum_{j=1}^{M} \left( \tilde{\nabla}_{t,i}^{(j)} - \nabla_{t,i}^{(j)} \right) \right\|_2^4 \right] \le U_i^2 G_i^2 \frac{g_{K,J}^3}{4M},$$

$$\sum_{t=g_{K,J}^2+1}^{T} \mathbb{E}_t[|X_t|^2] \le U_i^2 G_i^2 \frac{g_{K,J}}{4M} \left( T - g_{K,J}^2 \right).$$

With probability at least $1 - 2\delta$,

$$\Xi_{5,1} \le \sum_{t=1}^{T} \lambda_{t,i} \mathbb{E}_t \left[ \left\| \frac{1}{M} \sum_{j=1}^{M} \left( \tilde{\nabla}_{t,i}^{(j)} - \nabla_{t,i}^{(j)} \right) \right\|_2^2 \right] + \frac{2g_{K,J}G_iU_i}{3} \ln \frac{1}{\delta} + G_iU_i\sqrt{2\frac{g_{K,J}}{M}T \ln \frac{1}{\delta}}$$

$$\le \frac{g_{K,J}}{M} G_i^2 \sum_{t=1}^{T} \lambda_{t,i} + \frac{2g_{K,J}G_iU_i}{3} \ln \frac{1}{\delta} + G_iU_i\sqrt{2\frac{g_{K,J}}{M}T \ln \frac{1}{\delta}}.$$

Combining with all results gives, with probability at leat $1 - 3\delta$,

$$\frac{1}{M}\Xi_5 \le \frac{2U_i^2}{\lambda_{T,i}} + 2G_i^2 \left( 1 + \frac{g_{K,J}}{M} \right) \left( \sum_{t=1}^{g_{K,J}^2} \lambda_{t,i} + \sum_{t=g_{K,J}^2+1}^{T} \lambda_{t,i} \right) + 2g_{K,J}G_iU_i \ln \frac{1}{\delta} +$$

$$3G_iU_i\sqrt{\frac{2g_{K,J}T}{M} \ln \frac{1}{\delta}}$$

$$\le \frac{2U_i^2}{\lambda_{T,i}} + G_iU_i\sqrt{1 + \frac{g_{K,J}}{M}} \left( g_{K,J} + \int_{t=g_{K,J}^2+1}^{T} \frac{1}{\sqrt{t}} \mathrm{d}\,t \right) + 2g_{K,J}G_iU_i \ln \frac{1}{\delta} +$$

$$3G_iU_i\sqrt{\frac{2g_{K,J}T}{M} \ln \frac{1}{\delta}}$$

$$\le 6U_iG_i\sqrt{\left( 1 + \frac{g_{K,J}}{M} \right) T} + 2g_{K,J}G_iU_i \ln \frac{1}{\delta} + 3G_iU_i\sqrt{\frac{2g_{K,J}T}{M} \ln \frac{1}{\delta}}.$$

**Case 2**: $T \le g_{K,J}^2$.
In this case, we do not decompose $\Xi_{5,1}$ and $\lambda_{t,i} = \frac{U_i}{2G_i\sqrt{\left(1+\frac{g_{K,J}}{M}\right)g_{K,J}^2}}$. With probability at least $1 - \delta$,

$$\Xi_{5,1} \le \frac{g_{K,J}}{M} G_i^2 \sum_{t=1}^{T} \lambda_{t,i} + \frac{g_{K,J}G_iU_i}{3} \ln \frac{1}{\delta} + G_iU_i\sqrt{\frac{g_{K,J}}{2M}T \ln \frac{1}{\delta}}.$$

Furthermore, with probability at least $1 - 2\delta$,

$$
\frac{1}{M}\Xi_5 \leq \frac{2U_i^2}{\lambda_{T,i}} + 2G_i^2\left(1 + \frac{g_{K,J}}{M}\right)\sum_{t=1}^{T}\lambda_{t,i} + \frac{5g_{K,J}G_iU_i}{3}\ln\frac{1}{\delta} + 4G_iU_i\sqrt{\frac{g_{K,J}}{M}T\ln\frac{1}{\delta}}
$$

$$
\leq 5U_iG_i\sqrt{1 + \frac{g_{K,J}}{M}}\cdot g_{K,J} + \frac{5g_{K,J}G_iU_i}{3}\ln\frac{1}{\delta} + 4G_iU_i\sqrt{\frac{g_{K,J}}{M}T\ln\frac{1}{\delta}}.
$$

Combining the two cases gives, with probability at least $1 - (M+5)\delta$,

$$
\frac{1}{M}\Xi_5 \leq 6U_iG_i\sqrt{1 + \frac{g_{K,J}}{M}}\left(\sqrt{T} + g_{K,J}\right) + 2g_{K,J}G_iU_i\ln\frac{1}{\delta} + 3G_iU_i\sqrt{\frac{2g_{K,J}}{M}T\ln\frac{1}{\delta}}.
$$

### I.3 Combining all

Combining the upper bounds on $\Xi_4$ and $\Xi_5$ gives an upper bound on the regret.
With probability at least $1 - \Theta\left(M\log(CT) + \log(CKT/M)\right)\cdot\delta$,

$$
\sum_{t=1}^{T}\sum_{j=1}^{M}\ell\left(f_{t,A_{t,1}}^{(j)}(\mathbf{x}_t^{(j)}), y_t^{(j)}\right) - \sum_{t=1}^{T}\sum_{j=1}^{M}\ell\left(f(\mathbf{x}_t^{(j)}), y_t^{(j)}\right)
$$

$$
\leq M\sqrt{1 + \frac{g_{K,J}}{M}}\left(6U_iG_i\left(\sqrt{T} + g_{K,J}\right) + \frac{4}{\sqrt{2}}C_i\sqrt{T\ln(KT)}\right) +
$$

$$
(64C + 3MC + 2U_iG_i)g_{K,J}\ln\frac{1}{\delta} + (3\sqrt{2}G_iU_i + 17\sqrt{CC_i})\sqrt{2Mg_{K,J}T\ln\frac{1}{\delta}} + 2MC_i\sqrt{T\ln\frac{1}{\delta}}.
$$

Omitting the constant terms and lower order terms concludes the proof.

## J Proof of Theorem 3

We first establish a technical lemma.

**Lemma 5.** *Let $X_1, ..., X_K$ be a sequence of independent standard normal random variables. Let $Z_K = \max\{X_1, ..., X_K\}$. If $K \geq 5$, then $\mathbb{E}[Z_K] \geq \left(1 - \frac{1}{\sqrt{e}}\right)\sqrt{2\ln K}$.*

*Proof of Lemma 5.* Proposition 2.1.2 in Vershynin [2018] gives a lower bound on the tail probability of standard normal distribution.

$$
\forall x > 0, \mathbb{P}[X_1 \geq x] = \int_x^{+\infty}\frac{1}{\sqrt{2\pi}}\exp\left(-\frac{\mu^2}{2}\right)d\mu \geq \frac{1}{\sqrt{2\pi}}\left(\frac{1}{x} - \frac{1}{x^3}\right)\exp\left(-\frac{x^2}{2}\right).
$$

Then we have

$$
\begin{aligned}
\mathbb{E}[Z_K] =& \mathbb{E}\left[Z_K | \exists i \in [K], X_i \geq \varepsilon\right]\cdot\mathbb{P}\left[\exists i \in [K], X_i \geq \varepsilon\right] + \mathbb{E}[Z_K | \forall X_i < \varepsilon]\cdot\mathbb{P}\left[\forall X_i < \varepsilon\right] \\
\geq& \mathbb{P}\left[\exists i \in [K], X_i \geq \varepsilon\right]\cdot\varepsilon \\
=& (1 - \mathbb{P}\left[\forall X_i < \varepsilon\right])\cdot\varepsilon \\
=& \left(1 - \prod_{i=1}^{K}\mathbb{P}\left[X_i < \varepsilon\right]\right)\cdot\varepsilon \\
=& \left(1 - \prod_{i=1}^{K}\left(1 - \mathbb{P}\left[X_i \geq \varepsilon\right]\right)\right)\cdot\varepsilon \\
\geq& \left(1 - \left(1 - \frac{1}{\sqrt{2\pi}}\left(\frac{1}{\varepsilon} - \frac{1}{\varepsilon^3}\right)\exp\left(-\frac{\varepsilon^2}{2}\right)\right)^K\right)\cdot\varepsilon.
\end{aligned}
$$

Let $\varepsilon = \sqrt{2\ln K}$. If $K > 5$, then we have

$$\left(1 - \frac{1}{\sqrt{2\pi}}\left(\frac{1}{\varepsilon} - \frac{1}{\varepsilon^3}\right)\exp\left(-\frac{\varepsilon^2}{2}\right)\right)^K = \left(1 - \frac{1}{\sqrt{2\pi}}\left(\frac{1}{\sqrt{2\ln K}} - \frac{1}{\ln^{1.5}K^2}\right)\frac{1}{K}\right)^K$$

$$\leq \left(1 - \frac{1}{K^2}\right)^K$$

$$\leq \frac{1}{\sqrt{e}}.$$

Substituting into the lower bound of $\mathbb{E}[Z_K]$ concludes the proof. $\qquad\square$

## J.1 Proof of the First Lower Bound

*Proof.* Let $d \geq K$, $\mathcal{X} \subseteq \mathbb{R}^d$ and $\mathcal{Y} \in \{0,1\}$. We use the absolute loss function $\ell(f(\mathbf{x}_t), y_t) = |f(\mathbf{x}_t) - y_t|$. Recalling that

$$\mathcal{F}_i = \{f_i(\mathbf{x}) = \langle \mathbf{e}_i, \mathbf{x}\rangle\}, \quad i = 1, 2, ..., K,$$

where $\mathbf{e}_i$ is the standard basis vector in $\mathbb{R}^d$. It is obvious that the time complexity of computing $f_i(\mathbf{x}) = x_i$ is $O(1)$. At each client $j$, let the selected hypothesis be $f_t^{(j)}$ and the prediction be $f_t^{(j)}(\mathbf{x}_t^{(j)})$. Since there are no computational constraints on each client, $f_t^{(j)}(\mathbf{x}_t^{(j)})$ can be a weighted combination of $K$ predictions, i.e., $f_t^{(j)}(\mathbf{x}_t^{(j)}) = \sum_{i=1}^{K} w_{t,i}^{(j)} f_i(\mathbf{x}_t^{(j)})$. The time complexity of computing $f_t^{(j)}(\mathbf{x}_t^{(j)})$ is $O(K)$. We will follow the techniques used in the proof of Theorem 3.1 in Patel et al. [2023] and Theorem 3.7 in Cesa-Bianchi and Lugosi [2006].

Following the proof of Theorem 3.1 in Patel et al. [2023], the adversary gives a sequence of same examples for each client. To be specific, we define

$$\left(\mathbf{x}_t^{(j)}, y_t^{(j)}\right) = (\mathbf{x}_t, y_t), \quad t = 1, ..., T, \quad j = 1, ..., M,$$

where $\mathbf{x}_t = (b_{t,1}, b_{t,2}, ..., b_{t,K}, 0, ..., 0) \in \mathbb{R}^d$, and $b_{t,1}, b_{t,2}, ..., b_{t,K}, y_t$ is a sequence of symmetric i.i.d. Bernoulli random variables, i.e., $\mathbb{P}[y_t = 1] = \mathbb{P}[y_t = 0] = \frac{1}{2}$.

At any round $t$, the minimax regret against the best hypothesis can be simplified as follows

$$\inf_{f_1^{(1)},...,f_T^{(M)}} \sup_{(\mathbf{x}_t^{(j)}, y_t^{(j)}),, j\in[M], t\in[T]} \max_{i\in[K]} \mathrm{Reg}_D(\mathcal{F}_i)$$

$$\geq \inf_{f_1^{(1)},...,f_T^{(M)}} \sup_{(\mathbf{x}_t, y_t), t\in[T]} \max_{i\in[K]} \mathrm{Reg}_D(\mathcal{F}_i)$$

$$\geq \inf_{f_1^{(1)},...,f_T^{(M)}} \mathbb{E}_{(\mathbf{x}_t, y_t), t\in[T]} \left[\sum_{t=1}^{T}\sum_{j=1}^{M} \ell\left(f_t^{(j)}(\mathbf{x}_t), y_t\right) - \min_{i\in[K]}\sum_{t=1}^{T}\sum_{j=1}^{M}\ell\left(f_i(\mathbf{x}_t), y_t\right)\right]$$

$$= \inf_{f_1^{(1)},...,f_T^{(M)}} \mathbb{E}_{(\mathbf{x}_t, y_t), t\in[T]} \left[\sum_{t=1}^{T}\sum_{j=1}^{M}|f_t^{(j)}(\mathbf{x}_t) - y_t| - M\min_{i\in[K]}\sum_{t=1}^{T}|f_i(\mathbf{x}_t) - y_t|\right]$$

$$= \frac{MT}{2} - M\,\mathbb{E}_{(\mathbf{x}_t, y_t), t\in[T]}\left[\min_{i\in[K]}\sum_{t=1}^{T}|f_i(\mathbf{x}_t) - y_t|\right]$$

$$= M\,\mathbb{E}_{(\mathbf{x}_t, y_t), t\in[T]}\left[\max_{i\in[K]}\sum_{t=1}^{T}\left(\frac{1}{2} - f_i(\mathbf{x}_t)\right)\cdot(1 - 2y_t)\right],$$

in which $f_i(\mathbf{x}_t) = b_{t,i}$ is a Bernoulli random variable and

$$\mathbb{E}_{(\mathbf{x}, y_t), t\in[T]}\left[\sum_{t=1}^{T}\sum_{j=1}^{M}|f_t^{(j)}(\mathbf{x}_t) - y_t|\right] = \mathbb{E}_{y_t, t\in[T]}\left[\sum_{t=1}^{T}\sum_{j=1}^{M}y_t\right] = \frac{MT}{2}.$$

We further obtain

$$\inf_{f_1^{(1)},...,f_T^{(M)}} \sup_{(\mathbf{x}_t^{(j)},y_t^{(j)}),j\in[M],t\in[T]} \max_{i\in[K]} \text{Reg}_D(\mathcal{F}_i) \geq \frac{M}{2} \mathop{\mathbb{E}}_{\sigma_t,Z_{t,i},t\in[T],i\in[K]} \left[ \max_{i\in[K]} \sum_{t=1}^{T} Z_{t,i} \cdot \sigma_t \right]$$

$$= \frac{M}{2} \mathop{\mathbb{E}}_{Z_{t,i},t\in[T],i\in[K]} \left[ \max_{i\in[K]} \sum_{t=1}^{T} Z_{t,i} \right],$$

where both $\{Z_{t,i}\}_{t\in[T],i\in[K]}$ and $\{\sigma_t\}_{t\in[T]}$ are i.i.d. Rademacher random variables.

By Lemma A.11 in Cesa-Bianchi and Lugosi [2006], we obtain

$$\lim_{T\to\infty} \mathbb{E}\left[ \max_{i\in[K]} \frac{1}{\sqrt{T}} \sum_{t=1}^{T} Z_{t,i} \right] = \mathbb{E}\left[ \max_{i\in[K]} G_i \right],$$

where $G_1,...,G_N$ are independent standard normal random variables.

By Lemma 5, we obtain

$$\lim_{T\to\infty} \inf_{f_1^{(1)},...,f_T^{(M)}} \sup_{(\mathbf{x}_t^{(j)},y_t^{(j)}),j\in[M],t\in[T]} \max_{i\in[K]} \text{Reg}_D(\mathcal{F}_i) \geq \frac{1}{2}\left(1-\frac{1}{\sqrt{e}}\right)M\sqrt{2T\ln K},$$

which concludes the proof. $\qquad\square$

## J.2  Proof of the Second Lower Bound

We mainly use the techniques in the proof of Theorem 2 in Seldin et al. [2014], but also require a new technique. The idea of our proof is to reduce the online model selection on each client to multi-armed bandit problem with additional observations.

*Proof.* Now we prove the second lower bound in Theorem 3.

Let $d \geq K$, $\mathcal{X} \subseteq \mathbb{R}^d$ and $\mathcal{Y} \in \{0,1\}$. We use a linear loss function $\ell(f(\mathbf{x}_t),y_t) = 1 - y_t f(\mathbf{x}_t)$. Recalling that

$$\mathcal{F}_i = \{f_i(\mathbf{x}) = \langle \mathbf{e}_i, \mathbf{x} \rangle\}, \quad i = 1,2,...,K.$$

It is obvious that the time complexity of computing $f_i(\mathbf{x}) = x_i$ is $O(1)$. Under the constraint that the time complexity on each client is limited to $O(J)$, on each client, any algorithm can only select $J$ hypotheses and then output a prediction.

One of challenges is that the prediction may be a combination of $J$ predictions. To be specific, for each client $j \in [M]$, $f_t^{(j)}(\mathbf{x}_t^{(j)}) = \sum_{i\in O_t^{(j)}} w_{t,i} f_i(\mathbf{x}_t^{(j)})$, where $O_t^{(j)}$ contains the index of selected $J$ hypotheses by some algorithm. To address this challenge, we introduce a virtual strategy that randomly selects a hypothesis $f_{I_t^{(j)}}^{(j)} \in \{f_{A_{t,1}}, f_{A_{t,2}}, ..., f_{A_{t,J}}\}$ following the distribution $(w_{t,A_{t,1}}, w_{t,A_{t,2}}, ..., w_{t,A_{t,J}})$ where $A_{t,a} \in O_t^{(j)}$, $a = 1,...,J$. Since the loss function is a linear function, it is easy to prove that,

$$\mathbb{E}\left[\ell(f_{I_t^{(j)}}^{(j)}(\mathbf{x}_t^{(j)}),y_t^{(j)})\right] = \ell\left(\mathbb{E}\left[f_{I_t^{(j)}}^{(j)}(\mathbf{x}_t^{(j)})\right],y_t^{(j)}\right) = \ell\left(f_t^{(j)}(\mathbf{x}_t^{(j)}),y_t^{(j)}\right), \qquad (16)$$

where the expectation is taken over $I_t^{(j)}$. Assuming that $\ell(f_i(\mathbf{x}_t^{(j)}),y_t^{(j)}) \leq C$ for all $i = 1,...,K$. Lemma A.7 in Cesa-Bianchi and Lugosi [2006] gives, with probability at least $1-\delta$,

$$\sum_{t=1}^{T}\left[\ell(f_{I_t^{(j)}}^{(j)}(\mathbf{x}_t^{(j)}),y_t^{(j)}) - \ell\left(f_t^{(j)}(\mathbf{x}_t^{(j)}),y_t^{(j)}\right)\right] \leq -C\sqrt{\frac{T}{2}\ln\frac{1}{\delta}}.$$

Note that we assume $w_{t,i} \geq 0$ and $\sum_{i\in O_t^{(j)}} w_{t,i} = 1$ for all $t = 1,...,T$. Recalling that Theorem 3 assumes the outputs of algorithm belong to $[\min_{i\in[K],\mathbf{x}\in\mathcal{X}} f_i(\mathbf{x}), \max_{i\in[K],\mathbf{x}\in\mathcal{X}} f_i(\mathbf{x})]$. If $w_{t,i} < 0$ or $\sum_{i\in O_t^{(j)}} w_{t,i} > 1$, we can still find a weight vector $w'_{t,i} \geq 0$ and $\sum_{i\in O_t^{(j)}} w'_{t,i} = 1$, such that

$$f_t^{(j)}(\mathbf{x}_t^{(j)}) = \sum_{i\in O_t^{(j)}} w_{t,i} f_i(\mathbf{x}_t^{(j)}) = \sum_{i\in O_t^{(j)}} w'_{t,i} f_i(\mathbf{x}_t^{(j)}).$$

Then we sample $I_t^{(j)}$ following $(w'_{t,A_{t,1}}, w'_{t,A_{t,2}}, ..., w'_{t,A_{t,J}})$. We can replace $(w_{t,A_{t,1}}, w_{t,A_{t,2}}, ..., w_{t,A_{t,J}})$ with $(w'_{t,A_{t,1}}, w'_{t,A_{t,2}}, ..., w'_{t,A_{t,J}})$.

Since the algorithm is non-cooperative, the total regret can be decomposed into the summation of the regret on each client. With probability at least $1 - M\delta$,

$$\forall i \in [K], \quad \text{Reg}_D(\mathcal{F}_i) = \sum_{j=1}^{M} \left[ \sum_{t=1}^{T} \ell\left(f_t^{(j)}(\mathbf{x}_t^{(j)}), y_t^{(j)}\right) - \sum_{t=1}^{T} \ell\left(f_i(\mathbf{x}_t^{(j)}), y_t^{(j)}\right) \right]$$

$$= \sum_{j=1}^{M} \left[ \sum_{t=1}^{T} \ell\left(f_{I_t^{(j)}}^{(j)}(\mathbf{x}_t^{(j)}), y_t^{(j)}\right) - \sum_{t=1}^{T} \ell\left(f_i(\mathbf{x}_t^{(j)}), y_t^{(j)}\right) \right] +$$

$$\sum_{j=1}^{M} \left[ \sum_{t=1}^{T} \ell\left(f_t^{(j)}(\mathbf{x}_t^{(j)}), y_t^{(j)}\right) - \sum_{t=1}^{T} \ell\left(f_{I_t^{(j)}}^{(j)}(\mathbf{x}_t^{(j)}), y_t^{(j)}\right) \right]$$

$$\geq \underbrace{\sum_{j=1}^{M} \left[ \sum_{t=1}^{T} \ell\left(f_{I_t^{(j)}}^{(j)}(\mathbf{x}_t^{(j)}), y_t^{(j)}\right) - \sum_{t=1}^{T} \ell\left(f_i(\mathbf{x}_t^{(j)}), y_t^{(j)}\right) \right]}_{\overline{\text{Reg}}_D(\mathcal{F}_i)} + C\sqrt{\frac{T}{2}\ln\frac{1}{\delta}}. \quad (17)$$

Note that if we take expectation w.r.t. the randomness of algorithm, then the additional term $C\sqrt{\frac{T}{2}\ln\frac{1}{\delta}}$ in (17) can be omitted following (16).

If the prediction is not a combination of $J$ predictions, but just $f_{I_t^{(j)}}^{(j)}(\mathbf{x}_t^{(j)})$, then we have

$$\forall i \in [K], \quad \text{Reg}_D(\mathcal{F}_i) = \underbrace{\sum_{j=1}^{M} \left[ \sum_{t=1}^{T} \ell\left(f_{I_t^{(j)}}^{(j)}(\mathbf{x}_t^{(j)}), y_t^{(j)}\right) - \sum_{t=1}^{T} \ell\left(f_i(\mathbf{x}_t^{(j)}), y_t^{(j)}\right) \right]}_{\overline{\text{Reg}}_D(\mathcal{F}_i)}. \quad (18)$$

Combining with the two cases, we just need to analyze $\overline{\text{Reg}}_D(\mathcal{F}_i)$.

The adversary first uniformly samples a same $h \in [K]$ for all clients, and then constructs $\{(\mathbf{x}_t^{(j)}, y_t)\}_{t=1}^{T}$ as follows

$$\mathbf{x}_t^{(j)} = \mathbf{x}_t := (b_{t,1}, b_{t,2}, ..., b_{t,K}, 0, ..., 0), \quad y_t^{(j)} = 1, \quad j = 1, ..., M,$$

in which $b_{t,i}$ satisfies

$$\mathbb{P}_h[b_{t,i} = 1] = \frac{1-\rho}{2}, \quad \mathbb{P}_h[b_{t,i} = 0] = \frac{1+\rho}{2}, \quad i \neq h,$$
$$\mathbb{P}_h[b_{t,h} = 1] = \frac{1+\rho}{2}, \quad \mathbb{P}_h[b_{t,h} = 0] = \frac{1-\rho}{2}.$$

Let $\mathbb{E}_h[\cdot]$ and $\mathbb{P}_h[\cdot]$ separately be the expectation and probability operator conditioned on $h$ is selected. Then we have

$$\mathbb{P}_h[\ell(f_i(\mathbf{x}_t), 1) = 1] = \frac{1+\rho}{2}, \quad \mathbb{P}_h[\ell(f_i(\mathbf{x}_t), 1) = 0] = \frac{1-\rho}{2}, \quad i \neq h,$$
$$\mathbb{P}_h[\ell(f_h(\mathbf{x}_t), 1) = 1] = \frac{1-\rho}{2}, \quad \mathbb{P}_h[\ell(f_h(\mathbf{x}_t), 1) = 0] = \frac{1+\rho}{2}.$$

It is obvious that online model selection can be reduced to a $K$-armed bandit problem, in which $f_i$ is the $i$-th arm. At each round $t$, let $I_t^{(j)}$ be the selected arm. Besides, any algorithm can select another $J - 1$ arms. Thus any algorithm can observe $J$ losses. Let $O_t^{(j)}$ be the set of the selected $J$ arms. Note that $f_{I_t^{(j)}}^{(j)} = f_{I_t^{(j)}}$ for any $I_t^{(j)} \in [K]$.

Assuming that the algorithm is deterministic, that is, $O_t^{(j)}$ is determined by $\{O_\tau^{(j)}\}_{\tau=1}^{t-1}$ and the observed losses. Let $N_{T,i} = \sum_{t=1}^{T} \mathbb{I}_{I_t^{(j)}=i}$. Taking expectation w.r.t. $(b_{t,1}, ..., b_{t,K})_{t=1}^{T}$ yields

$$
\mathbb{E}_h \left[ \sum_{t=1}^{T} \ell \left( f_{I_t^{(j)}}^{(j)}(\mathbf{x}_t), 1 \right) - \min_{i \in [K]} \sum_{t=1}^{T} \ell \left( f_i(\mathbf{x}_t), 1 \right) \right]
$$

$$
\geq \mathbb{E}_h \left[ \sum_{t=1}^{T} \ell \left( f_{I_t^{(j)}}^{(j)}(\mathbf{x}_t), 1 \right) \right] - \min_{i \in [K]} \mathbb{E}_h \left[ \sum_{t=1}^{T} \ell \left( f_i(\mathbf{x}_t), 1 \right) \right]
$$

$$
= \rho \cdot \mathbb{E}_h \left[ \sum_{t=1}^{T} \mathbb{I}_{I_t^{(j)} \neq h} \right]
$$

$$
= \rho T \cdot \left( 1 - \frac{1}{T} \mathbb{E}_h [N_{T,h}] \right).
$$

Following the techniques in the proof of Theorem 2 in Seldin et al. [2014], we have

$$
\frac{1}{KT} \sum_{h=1}^{K} \mathbb{E}_h [N_{T,h}] \leq \frac{1}{K} + \sqrt{-\frac{JT}{K} \frac{2\rho^2}{1-\rho^2}}.
$$

Recalling that $T \geq K \geq 5$. Let $\rho = \frac{\sqrt{K}}{3\sqrt{JT}}$. We further have

$$
\frac{1}{K} \sum_{h=1}^{K} \left[ \mathbb{E}_h \left[ \sum_{t=1}^{T} \ell \left( f_{I_t^{(j)}}^{(j)}(\mathbf{x}_t), 1 \right) \right] - \min_{i \in [K]} \mathbb{E}_h \left[ \sum_{t=1}^{T} \ell \left( f_i(\mathbf{x}_t), 1 \right) \right] \right]
$$

$$
\geq \rho T \cdot \left( 1 - \frac{1}{K} - \frac{3}{2} \rho \sqrt{\frac{JT}{K}} \right)
$$

$$
\geq 0.1 \frac{\sqrt{KT}}{\sqrt{J}}. \tag{19}
$$

For any deterministic algorithm, we can prove

$$
\sup_{(\mathbf{x}_t^{(j)}, y_t^{(j)}), t \in [T], j \in [M]} \max_{i \in [K]} \overline{\text{Reg}}_D(\mathcal{F}_i)
$$

$$
\geq \sup_{(\mathbf{x}_t, 1), t \in [T], h \in [K]} \left[ \sum_{t=1}^{T} \sum_{j=1}^{M} \ell \left( f_t^{(j)}(\mathbf{x}_t), 1 \right) - \min_{i \in [K]} \sum_{t=1}^{T} \sum_{j=1}^{M} \ell \left( f_i(\mathbf{x}_t), 1 \right) \right]
$$

$$
= \sup_{(\mathbf{x}_t, 1), t \in [T], h \in [K]} \left[ \sum_{t=1}^{T} \sum_{j=1}^{M} \ell \left( f_t^{(j)}(\mathbf{x}_t), 1 \right) - M \min_{i \in [K]} \sum_{t=1}^{T} \ell \left( f_i(\mathbf{x}_t), 1 \right) \right]
$$

$$
\geq \sup_{h \in [K]} \mathbb{E}_h \mathbf{x}_t, t \in [T] \left[ \sum_{j=1}^{M} \left[ \sum_{t=1}^{T} \ell \left( f_{I_t^{(j)}}^{(j)}(\mathbf{x}_t), 1 \right) - \min_{i \in [K]} \sum_{t=1}^{T} \ell \left( f_i(\mathbf{x}_t), 1 \right) \right] \right]
$$

$$
\geq \sup_{h \in [K]} \sum_{j=1}^{M} \left[ \mathbb{E}_h \mathbf{x}_t, t \in [T] \left[ \sum_{t=1}^{T} \ell \left( f_{I_t^{(j)}}^{(j)}(\mathbf{x}_t), 1 \right) \right] - \min_{i \in [K]} \mathbb{E}_h \mathbf{x}_t, t \in [T] \left[ \sum_{t=1}^{T} \ell \left( f_i(\mathbf{x}_t), 1 \right) \right] \right]
$$

$$
\geq \mathbb{E}_{h \in [K]} \sum_{j=1}^{M} \left[ \mathbb{E}_h \left[ \sum_{t=1}^{T} \ell \left( f_{I_t^{(j)}}^{(j)}(\mathbf{x}_t), 1 \right) \right] - \min_{i \in [K]} \mathbb{E}_h \left[ \sum_{t=1}^{T} \ell \left( f_i(\mathbf{x}_t), 1 \right) \right] \right]
$$

$$
= \sum_{j=1}^{M} \frac{1}{K} \sum_{h=1}^{K} \left[ \mathbb{E}_h \left[ \sum_{t=1}^{T} \ell \left( f_{I_t^{(j)}}^{(j)}(\mathbf{x}_t), 1 \right) \right] - \min_{i \in [K]} \mathbb{E}_h \left[ \sum_{t=1}^{T} \ell \left( f_i(\mathbf{x}_t), 1 \right) \right] \right]
$$

$$
\geq 0.1 M \sqrt{\frac{K}{J}} T,
$$

---

**Algorithm 4** NCO-OMS

---

**Require:** $T, J, \eta_1, \{U_i, \lambda_{1,i}, i \in [K]\}$
**Ensure:** $f_{1,i}^{(j)} = 0, p_{1,i}, i \in [K], j \in [M]$
 1: **for** $t = 1, 2, \ldots, T$ **do**
 2:     **for** $j = 1, \ldots, M$ **do**
 3:        The client samples $O_t^{(j)}$ following (7)
 4:        The client outputs $f_{t,A_{t,1}}^{(j)}(\mathbf{x}_t^{(j)})$
 5:        The client computes $f_{t,A_{t,a}}^{(j)}(\mathbf{x}_t^{(j)})$ for all $a = 2, \ldots, J$
 6:        The client computes $\tilde{\nabla}_{t,i}^{(j)}$ and $\tilde{c}_{t,i}^{(j)}$ for all $i \in O_t^{(j)}$
 7:        The client computes $\mathbf{p}_{t+1}^{(j)}$ and $\mathbf{w}_{t+1,i}^{(j)}, i \in [K]$ following Definition 1
 8:     **end for**
 9: **end for**

---

where the last inequality comes from (19). As claimed in the proof of Theorem 6.11 in Cesa-Bianchi and Lugosi [2006], the lower bound of any randomized algorithm is same with that of any deterministic algorithm, i.e.,

$$
\sup_{(\mathbf{x}_t^{(j)}, y_t^{(j)}), t \in [T], j \in [M]} \mathbb{E}\left[\max_{i \in [K]} \overline{\mathrm{Reg}}_D(\mathcal{F}_i)\right]
$$

$$
= \sup_{(\mathbf{x}_t^{(j)}, y_t^{(j)}), t \in [T], j \in [M]} \left[\mathbb{E}\left[\sum_{t=1}^T \ell\left(f_{I_t^{(j)}}^{(j)}(\mathbf{x}_t^{(j)}), y_t^{(j)}\right)\right] - \min_{i \in [K]} \sum_{t=1}^T \ell\left(f_i(\mathbf{x}_t^{(j)}), y_t^{(j)}\right)\right]
$$

$$
\geq 0.1 M \frac{\sqrt{KT}}{\sqrt{J}},
$$

in which the expectation is taken over the internal randomness of algorithm. Substituting into (17), or (18) concludes the proof. $\qquad\square$

## K    Regret Analysis of NCO-OMS

Algorithm 4 gives the pseudo-code of NCO-OMS.

Following the definition of NCO-OMS and Algorithm 4, it is obvious that the regret bound of NCO-OMS on each client is same with Theorem 2 in which we set $M = 1$. The regret bound on $M$ clients is $M$ times of that of a client. Thus we have Theorem 6.

**Theorem 6** (Regret Bound of NCO-OMS). *Let the learning rate $\eta$, $\lambda_{t,i}$ and the initial distribution $\mathbf{p}_1$ be same for each client $j \in [M]$. The values of $\eta$, $\lambda_{t,i}$ and $\mathbf{p}_1$ follow Theorem 2 in which $M = 1$. With probability at least $1 - \Theta\left(M \log(KT)\right) \cdot \delta$, the regret of NCO-OMS satisfies:*

$$
\forall i \in [K], \ \mathrm{Reg}_D(\mathcal{F}_i) = O\left(M\left(B_{i,1}\sqrt{(1 + g_{K,J})T} + B_{i,2}g_{K,J}\ln\frac{1}{\delta} + B_{i,3}\sqrt{g_{K,J}T\ln\frac{1}{\delta}}\right)\right),
$$

*where $B_{i,1} = U_iG_i + C_i\sqrt{\ln(KT)}$, $B_{i,2} = C + U_iG_i$ and $B_{i,3} = U_iG_i + \sqrt{CC_i}$ and $C = \max_i C_i$.*

## L    Proof of Theorem 4

### L.1    Algorithm

We give the pseudo-code in Algorithm 5.

To implement Algorithm 5, we require one more technique, i.e., the random features [Rahimi and Recht, 2007]. We will use the random features to construct an approximation of the implicity kernel mapping. The are two reasons. The first one is that we can avoid transferring the data itself and thus the privacy is protected. The second one is that we can avoid the $O(T)$ computational cost on the clients.

For any $i \in [K]$, we consider the kernel function $\kappa_i(\mathbf{x}, \mathbf{v})$ that has an integral representation, i.e.,

$$\kappa_i(\mathbf{x}, \mathbf{v}) = \int_\Gamma \varphi_i(\mathbf{x}, \omega)\varphi_i(\mathbf{v}, \omega)\mathrm{d}\,\mu_i(\omega), \ \forall \mathbf{x}, \mathbf{v} \in \mathcal{X}, \tag{20}$$

where $\varphi_i : \mathcal{X} \times \Gamma \to \mathbb{R}$ is the eigenfunctions and $\mu_i(\cdot)$ is a distribution function on $\Gamma$. Let $p_i(\cdot)$ be the density function of $\mu_i(\cdot)$. We sample $\{\omega_j\}_{j=1}^D \sim p_i(\omega)$ independently and compute

$$\tilde{\kappa}_i(\mathbf{x}, \mathbf{v}) = \frac{1}{D}\sum_{j=1}^D \varphi_i(\mathbf{x}, \omega_j)\varphi_i(\mathbf{v}, \omega_j).$$

For any $f(\mathbf{x}) = \int_\Gamma \alpha(\omega)\varphi_i(\mathbf{x}, \omega)p_i(\omega)\mathrm{d}\,\omega$. We can approximate $f(\mathbf{x})$ by $\hat{f}(\mathbf{x}) = \frac{1}{D}\sum_{j=1}^D \alpha(\omega_j)\varphi_i(\mathbf{x}, \omega_j)$. It can be verified that $\mathbb{E}[\hat{f}(\mathbf{x})] = f(\mathbf{x})$. Such an approximation scheme also defines an explicit feature mapping denoted by

$$\phi_i(\mathbf{x}) = \frac{1}{\sqrt{D}}\left(\varphi_i(\mathbf{x}, \omega_1), \ldots, \varphi_i(\mathbf{x}, \omega_D)\right).$$

For each $\kappa_i, i \in [K]$, we define two hypothesis spaces [Rahimi and Recht, 2008, Li and Liao, 2022] as follows

$$
\begin{aligned}
\mathcal{F}_i &= \left\{ f(\mathbf{x}) = \int_\Gamma \alpha(\omega)\varphi_i(\mathbf{x}, \omega)p_i(\omega)\mathrm{d}\,\omega \,\middle|\, \|\alpha(\omega)\| \leq U_i \right\}, \\
\mathbb{H}_i &= \left\{ \hat{f}(\mathbf{x}) = \sum_{j=1}^D \alpha_j \varphi_i(\mathbf{x}, \omega_j) \,\middle|\, |\alpha_j| \leq \frac{U_i}{D} \right\} \\
&= \left\{ \hat{f}(\mathbf{x}) = \mathbf{w}^\top \phi_i(\mathbf{x}) \,\middle|\, \mathbf{w} = \sqrt{D}(\alpha_1, \ldots, \alpha_D) \in \mathbb{R}^D, |\alpha_j| \leq \frac{U_i}{D} \right\},
\end{aligned}
\tag{21}
$$

in which $\mathcal{F}_i$ is exact the hypothesis space defined in (1).

It can be verified that $\|\mathbf{w}\|_2^2 \leq U_i^2$. Let $\mathcal{W}_i = \{\mathbf{w} \in \mathbb{R}^D : \|\mathbf{w}\|_\infty \leq \frac{U_i}{\sqrt{D}}\}$. We replace (9) with (22),

$$
\begin{cases}
\nabla_{\bar{\mathbf{w}}_{t+1,i}}\psi_{t,i}(\bar{\mathbf{w}}_{t+1,i}) = \nabla_{\mathbf{w}_{t,i}}\psi_{t,i}(\mathbf{w}_{t,i}) - \dfrac{1}{M}\sum_{j=1}^M \tilde{\nabla}_{t,i}^{(j)}, \quad i = 1, \ldots, K, \\[2mm]
\mathbf{w}_{t+1,i} = \underset{\mathbf{w} \in \mathcal{W}_i}{\arg\min}\, \mathcal{D}_{\psi_{t,i}}(\mathbf{w}, \bar{\mathbf{w}}_{t+1,i}), \\[2mm]
\psi_{t,i}(\mathbf{w}) = \dfrac{1}{2\lambda_{t,i}} \cdot \|\mathbf{w}\|_2^2.
\end{cases}
\tag{22}
$$

## L.2 Regret Analysis

We first give an assumption and a technique lemma.

**Assumption 2** (Li et al. [2019]). *For any $i \in [K]$, if $\kappa_i$ satisfies (20), then there is a bounded constant $b_i$ such that, $\forall \mathbf{x} \in \mathcal{X}$, $|\varphi_i(\mathbf{x}, \omega)| \leq b_i$.*

Under Assumption 2, we have $|f(\mathbf{x})| \leq U_i b_i$ for any $f \in \mathbb{H}_i$ and $f \in \mathcal{F}_i$. It is worth mentioning that if Assumption 2 holds, then Assumption 1 holds with the same $b_i$.

**Lemma 6.** *For any $i \in [K]$, let $\mathcal{F}_i$ and $\mathbb{H}_i$ follow (21). With probability at least $1 - \delta$, $\forall f \in \mathcal{F}_i$, there is a $\hat{f} \in \mathbb{H}_i$ such that $|f(\mathbf{x}) - \hat{f}(\mathbf{x})| \leq \frac{U b_i}{\sqrt{D}}\sqrt{2\ln\frac{1}{\delta}}$.*

The lemma is adopted from Lemma 5 in [Li and Liao, 2023]. Thus we omit the proof.

Now we begin to prove Theorem 4.

**Algorithm 5** FOMD-OMS for Distributed OMKL

---

**Require:** $U, T, R, J$.
**Ensure:** $f_{1,i}^{(j)} = 0, p_{1,i}, i \in [K], j \in [M]$
1: **for** $r = 1, 2, \ldots, R$ **do**
2:    **for** $t \in T_r$ **do**
3:       **if** $t == (r-1)N + 1$ **then**
4:          **for** $j = 1, \ldots, M$ **do**
5:             Server samples $O_t^{(j)}$ following (7)
6:             Server transmits $\mathbf{w}_{t,i}, i \in O_t^{(j)}$ to the $j$-th client
7:          **end for**
8:       **end if**
9:       **for** $j = 1, \ldots, M$ in parallel **do**
10:          **for** $i \in O_t^{(j)}$ **do**
11:             Computing $\phi_i(\mathbf{x}_t^{(j)})$
12:          **end for**
13:          Outputting $\mathbf{w}_{t,A_{t,1}}^\top \phi_{A_{t,1}}(\mathbf{x}_t^{(j)})$ and receiving $y_t^{(j)}$
14:          **for** $i \in O_t^{(j)}$ **do**
15:             Computing $\nabla_{t,i}^{(j)}$ and $c_{t,i}^{(j)}$
16:          **end for**
17:       **end for**
18:       **if** $t == rN$ **then**
19:          Clients transmit $\{\frac{1}{N}\sum_{t \in T_r} \nabla_{t,i}^{(j)}, \frac{1}{N}\sum_{t \in T_r} c_{t,i}^{(j)}\}_{i \in O_t^{(j)}}$ to server
20:          Server computes $\mathbf{p}_{t+1}$ following (8)
21:          Server computes $\mathbf{w}_{t+1,i}, i \in [K]$ following (22)
22:       **end if**
23:    **end for**
24: **end for**

---

*Proof of Theorem 4.* The regret w.r.t. any $f \in \mathcal{F}_i$ can be decomposed as follows.

$$\sum_{t=1}^{T}\sum_{j=1}^{M} \ell\left(f_{t,A_{t,1}}^{(j)}(\mathbf{x}_t^{(j)}), y_t^{(j)}\right) - \sum_{t=1}^{T}\sum_{j=1}^{M} \ell\left(f(\mathbf{x}_t^{(j)}), y_t^{(j)}\right)$$

$$= \underbrace{\sum_{t=1}^{T}\sum_{j=1}^{M}\left[\ell\left(f_{t,A_{t,1}}^{(j)}(\mathbf{x}_t^{(j)}), y_t^{(j)}\right) - \ell\left(f_{t,i}^{(j)}(\mathbf{x}_t^{(j)}), y_t^{(j)}\right) + \ell\left(f_{t,i}^{(j)}(\mathbf{x}_t^{(j)}), y_t^{(j)}\right) - \ell\left(\hat{f}(\mathbf{x}_t^{(j)}), y_t^{(j)}\right)\right]}_{\text{Reg}_D(\mathbb{H}_i)}$$

$$+ \underbrace{\sum_{t=1}^{T}\sum_{j=1}^{M}\left[\ell\left(\hat{f}(\mathbf{x}_t^{(j)}), y_t^{(j)}\right) - \ell\left(f(\mathbf{x}_t^{(j)}), y_t^{(j)}\right)\right]}_{\Xi_6}$$

$$= \text{Reg}_D(\mathbb{H}_i) + \Xi_6.$$

$\text{Reg}_D(\mathbb{H}_i)$ is the regret that we run FOMD-OKS with hypothesis spaces $\{\mathbb{H}_i\}_{i=1}^{K}$. $\hat{f} \in \mathbb{H}_i$ satisfies Lemma 6. In other words, $\Xi_6$ is induced by the approximation error that we use $\hat{f}$ to approximate $f$.

$\text{Reg}_D(\mathbb{H}_i)$ has been given by Theorem 5. Next we analyze $\Xi_6$.

Using the convexity of $\ell(\cdot, \cdot)$, with probability at least $1 - TM\delta$,

$$
\begin{aligned}
\Xi_6 &\leq \sum_{t=1}^{T} \sum_{j=1}^{M} \frac{\mathrm{d}\,\ell\left(\hat{f}(\mathbf{x}_t^{(j)}), y_t^{(j)}\right)}{\mathrm{d}\,\hat{f}(\mathbf{x}_t^{(j)})} \cdot \left(\hat{f}(\mathbf{x}_t^{(j)}) - f(\mathbf{x}_t^{(j)})\right) \\
&\leq \sum_{t=1}^{T} \sum_{j=1}^{M} \left|\frac{\mathrm{d}\,\ell\left(\hat{f}(\mathbf{x}_t^{(j)}), y_t^{(j)}\right)}{\mathrm{d}\,\hat{f}(\mathbf{x}_t^{(j)})}\right| \cdot \left|\hat{f}(\mathbf{x}_t^{(j)}) - f(\mathbf{x}_t^{(j)})\right| \\
&\leq g_i b_i U_i \frac{MT}{\sqrt{D}} \sqrt{2 \ln \frac{1}{\delta}} \\
&\leq G_i U_i \frac{MT}{\sqrt{D}} \sqrt{2 \ln \frac{1}{\delta}}.
\end{aligned}
$$

Under Assumption 2, there is a constant $g_i$ such that $\left|\frac{\mathrm{d}\,\ell\left(\hat{f}(\mathbf{x}_t^{(j)}), y_t^{(j)}\right)}{\mathrm{d}\,\hat{f}(\mathbf{x}_t^{(j)})}\right| \leq g_i$. The last inequality comes from the definition of Lipschitz constant (see Lemma 2).

Combining the upper bounds on $\operatorname{Reg}_D(\mathbb{H}_i)$ and $\Xi_6$ concludes the proof. $\qquad\square$

