# OpenReview forum: "On the Necessity of Collaboration for Online Model Selection with Decentralized Data"
_NeurIPS.cc/2024/Conference — NeurIPS 2024 poster_

### Official Review · Reviewer_83EK · 2024-07-10

**Soundness:** 3
**Presentation:** 3
**Contribution:** 4
**Rating:** 8
**Confidence:** 2

**Summary:**

The paper studies the problem of regret minimization $ \min_{f_1, \ldots, f_T} \left( \sum_{t=1}^T \ell(f_t(x_t), y_t) - \min_{f \in \mathcal{F}^*} \sum_{t=1}^T \ell(f(x_t), y_t) \right)$ in distributed settings with $M$ clients. The paper proves lower bounds on regret and introduces a federated algorithm with an upper bound analysis. The paper proves collaboration is unnecessary without computational constraints on sites and necessary if the computation per client is bounded by $o(K)$ where $K$ is the number of hypotheses ($F$).

**Strengths:**

The paper made a strong contribution. It proposes a new algorithm that improves previous bounds on computation and communication costs and establishes interesting lower bounds on algorithms. The work presents three new techniques: improved Bernstein's inequality for martingales, a federated online mirror descent framework, and decoupling model selection and prediction. The technical contribution is sold, and the paper is well-written. The paper contains extensive experimental studies that support theoretical bounds.

**Weaknesses:**

I didn't see any significant weaknesses.

**Questions:**

N/A

---

> ### Author Rebuttal · Authors · 2024-08-07
>
> We are grateful to the reviewer for taking the time to evaluate our paper and for providing encouraging feedback. We will continue our research in this area to make more controbutions.

---

### Official Review · Reviewer_EYSG · 2024-07-12

**Soundness:** 3
**Presentation:** 2
**Contribution:** 2
**Rating:** 5
**Confidence:** 3

**Summary:**

This paper discusses the necessity of collaboration among clients in a decentralized data setting for online model selection. The authors analyze this necessity from the perspective of computational constraints and propose a new federated learning algorithm. They prove the upper and lower bounds of this algorithm. The main conclusion is that collaboration is unnecessary if there are no computational constraints on clients, but necessary if each client's computational cost is limited to o(K), where K is the number of candidate hypothesis spaces. The key contributions include establishing conditions under which collaboration is required, designing an improved federated learning algorithm (FOMD-OMS) separating model selection from client prediction, which improves regret bounds and enhances model selection performance at lower computational and communication costs.

**Strengths:**

1. The paper provides a rigorous theoretical analysis of the necessity of collaboration in federated learning for online model selection, considering computational constraints.
2. The algorithm demonstrates improved regret bounds and performance in model selection while maintaining low computational and communication costs.

**Weaknesses:**

1. The motivation for federated learning in the context of online model selection with decentralized data is not clearly articulated. It would be beneficial to provide more details to justify the need for federated learning for practical applications, such as the online hyperparameter tuning mentioned in the example.
2. As shown in Section 4.4, the exact communication cost for $o(K)$ is $O(MJ\log K)$, where $J$ may have the same order with $K$ and introduce a large communication cost. It would be better to discuss the choice of $J$ and provide the exact communication cost in the abstract and introduction part.
3. The paper does not sufficiently discuss the difference from previous works on federated online bandits. Both settings involve selection among K options (arms or hypothesis spaces), and a detailed comparison would highlight the unique aspects of this work.
3. When considering setting $K=1$, [1] achieve results $O(M\sqrt{T}+\sqrt{MT})$ result for federated adversarial linear bandits setting, which has a similar form compared to the result of this work. It would be better to discuss the technique challenge in the proof process.

**Questions:**

Please see the weaknesses part.

Question 1: How does this work specifically differ from and improve upon federated online bandit algorithms? What are the unique challenges introduced with the online model selection setting?

**Limitations:**

The authors have adequately addressed the limitations and societal impact.

---

> ### Author Rebuttal · Authors · 2024-08-07
>
> We thank the reviewer for detailed comments and valuable suggestions. The suggestions will make our presentation more clear. Next we answer the questions.
>
> **[Question 1]** The motivation for federated learning in the context of OMS-DecD is not clearly articulated. It would be beneficial to provide more details to justify the need for federated learning for practical applications.
>
> **[Answer]** In many real-world scenarios, data are generated at the edge (e.g., smartphones, IoT devices). Transmitting vast amounts of data to a central server violates the privacy concerns. Federated Learning (FL) mitigates this by only sharing model updates, thereby perserving data privacy.
>
> In computing scenarios with limited computational resources (e.g.,constrained memory and processing power), online algorithms deployed a single device could only perform approximate model selection and model updating, thereby leading suboptimal solutions. **FL aggregates information from various clients without inducing significant additional computational cost on each client, and thus have the potentials to improve the performance**. Our work further theoretically demonstrate that FL is necessary for OMS-DecD when the computational resources available on each client are limited.
>
> **[Question 2]** In the Introduction part, why the communication cost is $o(K)$ instead of the exact $O(MJ\log(K))$. How to choose the value of $J$.
>
> **[Answer]** As stated in Section 4.5, if $J$ is of the same order as $K$, i.e., $J=\Theta(K)$, then there is no need to use federated learning for OMS-DecD, and consequently, communication is unnecessary. In the case of $J=o(K)$, federated learning is necessary. In this case, the communication of our algorithm is also $o(K)$. Therefore, for convinence of presentation, we state that our algorithm has a $o(K)$ communication cost in the Introduction.
>
> As stated in Section 5, setting $J=2$ is sufficient to achieve the desired improvement in the regret bound, time complexity, or communication complexity compared to previous algorithms. Actually, our algorithm explicitly balances the regret and computational cost by adjusting $J$. The larger the value of $J$, the smaller the regret bound becomes, while the larger the computational complexity increases.
>
> **[Question 3]** The paper does not sufficiently discuss the difference from previous works on federated online bandits. Both settings involve selection among $K$ options (arms or hypothesis spaces), and a detailed comparison would highlight the unique aspects of this work.
>
> **[Answer]** This paper focused on the work most related to OMS-DecD. We are happy to discuss federated bandits, such as federated $K$-armed bandits [1] and federated linear contextual bandits [2], in our revision. Next we detail the differences between federated bandits and OMS-DecD.
>
> (1)	For OMS-DecD, we do not require the examples $({\bf x}^{(j)}_t,y^{(j)}_t)$, $t=1,…,T$, on each client to be i.i.d. (independent and identically distributed), in contrast, the rewards $r^{(j)}_t$, $t=1,...,T$, in both federated $K$-armed bandits or federated linear contextual bandits must be i.i.d. **The assumption of i.i.d. rewards makes collaboration effective for federated bandits (similar to federated stochastic optimization); however, this may not hold true for OMS-DedD**. Therefore, **it is a unique problem of OMS-DecD to study whether collaboration is effective.**
>
> (2)	The challenges posed by the two problems are distinctly different. The hypothesis space $\mathcal{F}_i$, such as a RKHS, enjoys a complex structure, while an arm (or action) does not has a structure. We can measure the complexity of $\mathcal{F}_i$ by $\mathfrak{C}_i>0$, while there is not a complexity defined for an arm. **Thus algorithms for OMS-DecD must adapt to the complexity of individual hypothesis spaces, posing unique challenges on algorithm design and theoretical analysis.**
>
> **[Question 4]** When considering setting $K=1$, [3] achieves the $O(M\sqrt{T}+\mathrm{poly}(d)\sqrt{MT})$ result for federated adversarial linear bandits, which has a similar form compared to the result of this work. It would be better to discuss the technical challenges in the proof.
>
> **[Answer]**
> (1)	Our regret bounds necessitate an analysis for both federated online gradient descent (F-OGD) and federated exponential gradient descent (F-EGD). However, the proof in [3] focuses exclusively on F-OGD, but is not applicable to F-EGD. **The first technical challenge is to analyze the regret of F-EGD**. To this end, our proofs offer a template for analyzing federated online mirror descent, which can be applied to both F-OGD and F-EGD. It is interesting that we decouple the regret bound into two parts: the first part cannot be reduced by federated averaging, while the second part highlights the benefits of federated averaging (see Theorem 1).
>
> (2) Regarding the dependence on problem-specific constants, the regret bound in [3] differs significantly from our results. As elucidated in the answer for Question 3, algorithms for OMS-DecD must adapt to the complexity of individual hypothesis spaces. Thus **the second technical challenge is the derivation of a high-probability regret bound that adapts to the complexity of individual hypothesis spaces.** To address this issue, we establish a new Bernstein’s inequality for martingale.
>
> We will incorporate the discussion in the revision. Thanks for the suggestion.
>
> **[Question 5]** How does this work specifically differ from and improve upon federated online bandit algorithms? What are the unique challenges introduced with the online model selection setting?
>
> **[Answer]** Please refer to the answer for Question 3.
>
>
> **Reference**
>
> [1] Wang et al. Distributed Bandit Learning: Near-Optimal Regret with Efficient Communication. ICLR, 2020.
>
> [2] Huang et al. Federated Linear Contextual Bandits. NeurIPS, 2021.
>
> [3] Patel, K. K., et al. Federated online and bandit convex optimization. ICML, 2023.

---

> > ### Comment · Reviewer_EYSG · 2024-08-12
> >
> > Thanks for the detailed responses. The discussion on motivation and communication costs in the responses is helpful in deepening the understanding of this work. A detailed discussion of the related works also makes the contribution of this work clearer. Overall, I will raise my score to $5$ since the rebuttal addresses my concerns.

---

> > > ### Author Response · Authors · 2024-08-12
> > >
> > > Thank you for taking the time to provide feedback on our response. We are pleased that our reply has addressed your concerns. We also appreciate your positive evaluation of our contributions.

---

### Official Review · Reviewer_eZoF · 2024-07-13

**Soundness:** 3
**Presentation:** 4
**Contribution:** 3
**Rating:** 6
**Confidence:** 4

**Summary:**

The paper proposes a new online multi-kernel federated learning algorithm. The proposed method works based on estimating the gradients by the clients to make collaborations among clients for online federated learning helpful. The paper analyzes the regret bound of the proposed algorithm.

**Strengths:**

1. The paper tries to address the important and interesting problem of benefit of collaboration in online model selection and online multi-kernel learning.
2. The paper provides complete comparisons between regret upper of the proposed algorithm and those of other works.
3. The paper is well-written and clear.

**Weaknesses:**

The proposed algorithm improves the regret of online multi-kernel learning by constant. Maybe adding some assumptions can help to achieve tighter regret bounds. Finding such conditions can be an interesting direction for future research.

**Questions:**

The paper states that some algorithms suffer from $\mathcal O(K)$ download cost. However, based on my understanding download cost may not be that significant. Can you explain the importance of download cost?

**Limitations:**

Limitations adequately discussed.

---

> ### Author Rebuttal · Authors · 2024-08-07
>
> We thank the reviewer for valuable comments and suggestions. Next we answer the questions.
>
> **[Question 1]** The proposed algorithm improves the regret of online multi-kernel learning by constant. Maybe adding some assumptions can help to achieve tighter regret bounds. Finding such conditions can be an interesting direction for future research.
>
> **[Answer]** We appreciate the reviewer’s valuable suggestion. It is interesting to explore conditions under which we can explicitly compare the constants.
>
> **[Question 2]** The paper states that some algorithms suffer from $O(K)$ download cost. Can the authors explain the importance of download cost?
>
> **[Answer]** This is a good question. There are algorithms that enjoy $O(\log{K})$ upload cost, but suffer from $O(K)$ download cost, such as POF-MKL[1]. If $K$ is sufficiently large such that the speed of download is less than $K$ times of the speed of upload, then the download cost becomes the bottleneck in communication. Therefore, it is important to reduce the download cost for such algorithms.
>
> **Reference**
>
> [1] Pouya M. Ghari and Yanning Shen. Personalized online federated learning with multiple kernels. NeurIPS, 2022.

---

### Official Review · Reviewer_N9xw · 2024-07-22

**Soundness:** 2
**Presentation:** 2
**Contribution:** 3
**Rating:** 6
**Confidence:** 1

**Summary:**

This paper looks at the problem of Online Model Selection in a Decentralized context (OMS-DECD). The authors show the relationship of this problem class with multiple problem classes included Online Model Selection, federated learning, online multi-kernel learning etc. They consider two aspects of the problem 1. whether they can get a minimized regret for the OMS-DECD problem and 2. whether collaboration is necessary (between clients) to get privacy preserving version of regret.
The authors show initially a federated version of online mirror descent with no Local update which provides a bound on the regret. Additionally they develop a federated OMS algorithm which provides an upper bound on the regret and recreates the result of historical work. They also develop lower bound results for cooperative and non-cooperative algorithms. Finally they show that collaboration is unnecessary ( the best bounds are obtained without collaboration) when there are no computational constraints on the clients, thus generalizing the results of distributed online learning and collaboration is necessary if the computation cost is constrained to o(K) where K is the number of clients. Along the way they develop a new Berstein's inequality for a Martingale to show the high probability bounds on the regret making it a stronger bounds than in expectation.

**Strengths:**

I have not checked the mathematical accuracy of the proofs but the results look novel and are well motivated in the context of recent work and connection with other recent results. This helps us provide a unique perspective about when collaboration is actually necessary (the authors claim that they are not required if we focus on only the statistical perspective and not the computational aspects of the algorithms) and also gives us high probability bounds on the regret.

**Weaknesses:**

The paper is difficult to follow at times due to the mathematical density. It would be better presented by motivating the actual theorems in the paper and pushing the mathematical proofs to the appendix possibly. In particular, it was a bit difficult to follow the connection between FOMD-no-LU and FOMD-OMS and why the authors promoted the first algorithm. Having these details summarized in a table/section would make the paper much more readable.

**Questions:**

Addressed in the previous section.
Additionally I had a slightly naive question regarding how to compare the regret bound of the three algorithms in table 1. It seems like the FOMD-OMS is adding an additional term and can potentially have poorer regret if C and C_i are comparable. Are these regret bounds not to be compared as they might be due to different function classes?

**Limitations:**

No specific limitations.

---

> ### Author Rebuttal · Authors · 2024-08-07
>
> We thank the reviewer for offering valuable comments and suggestions. Next we answer the questions.
>
> **[Question 1]** The paper is difficult to follow at times due to the mathematical density.  It would be better presented by motivating the actual theorems in the paper.
>
> **[Answer]**  We will provide more insights into the theorems. We appreciate the reviewer’s valuable suggestions.
>
> **[Question 2]** It was a bit difficult to follow the connection between FOMD-No-LU and FOMD-OMS and why the authors promoted the first algorithm.
>
> **[Answer]** At a high level, **FOMD-No-LU is a federated algorithmic framework** that establishes the update rules in a general form, **from which we derive FOMD-OMS**. To be specific, FOMD-OMS updates hypotheses by federated online gradient descent (F-OGD), and updates sampling probabilities by federated exponential gradient descent (F-EGD). Both F-OGD and F-EGD can be derived from FOMD-No-LU, which is a federated mirror descent framework.
>
> Therefore, **in order to provide more concise presentations and theoretical analyses, we introduce FOMD-No-LU and establish a general regret bound for it**. We believe that both FOMD-No-LU and the general regret bound might be of independent interest.
>
>
> **[Question 3]** Are these regret bounds in Table 1 not to be compared as they might be due to different function classes? It seems like the FOMD-OMS is adding an additional term and can potentially have poorer regret if $\mathfrak{C}$ and $\mathfrak{C}_i$ are comparable.
>
> **[Answer]** Thanks for raising the question. Indeed, the regret bounds in Table 1 are comparable.
> Recalling that, for each $i=1,...,K$, we define $\mathfrak{C}_i=\Theta(U_i G_i+C_i)$ where $U_i$, $G_i$ and $C_i$ are problem-dependent constants, and $\mathfrak{C}=\max_i\mathfrak{C}_i$. We have $\mathcal{C}_i\leq \mathcal{C}$. Next we separately compare FOMD-OMS with POF-MKL [1] and eM-KOFL [2].
>
> **FOMD-OMS vs POF-MKL**.
>
> From Table 1, it can be verified that
> $$
>     \mathfrak{C}_i M\sqrt{T\ln{K}}+\sqrt{\mathfrak{C}\mathfrak{C}_iMKT} +\frac{\mathfrak{C}_iMT}{\sqrt{D}}
>     <
>     \mathfrak{C}M\sqrt{KT}+\frac{\mathfrak{C}_iMT}{\sqrt{D}}.
> $$
> Thus **the regret bound of FOMD-OMS is better than that of POF-MKL**.
>
> **FOMD-OMS vs eM-KOFL**.
>
> (1) In the case of $K\leq\mathfrak{C}M/\mathfrak{C}_i\cdot\ln{K}$, we have
> $$
>     \mathfrak{C}_i M\sqrt{T\ln{K}}+\sqrt{\mathfrak{C}\mathfrak{C}_iMKT}
>     +\frac{\mathfrak{C}_iMT}{\sqrt{D}}
>     \lesssim \mathfrak{C}M\sqrt{T\ln{K}}+\frac{\mathfrak{C}_iMT}{\sqrt{D}},
> $$
> where we use the notation $\lesssim$ to hide constant factors. Thus **the regret bound of FOMD-OMS is better than that of eM-KOFL**.
>
> (2) In the csae of $K\geq\mathfrak{C}M/\mathfrak{C}_i\cdot\ln{K}$, we have
> $$
>     \mathfrak{C}M\sqrt{T\ln{K}}+\frac{\mathfrak{C}_iMT}{\sqrt{D}}
>     \lesssim
>     \mathfrak{C}_i M\sqrt{T\ln{K}}+\sqrt{\mathfrak{C}\mathfrak{C}_iMKT}
>     +\frac{\mathfrak{C}_iMT}{\sqrt{D}}.
> $$
> Thus **the regret bound of FOMD-OMS is worse than that of eM-KOFL**. If $K$ is sufficiently large, the regret bound of eM-KOFL is better than that of FOMD-OMS by a factor of $O(\sqrt{K})$.
>
> From Table 1, we observe that FOMD-OMS significantly improves the computational complexity of POF-MKL and eM-KOFL (by a factor of $O(K)$), either maintaining a better regret bound or incurring only a small expense in the regret bound.
>
> Furthermore, the **collaboration is ineffective in POF-MKL and eM-KOFL**, while **it is effective in FOMD-OMS**.
>
>
>
> **Reference**
>
> [1] Pouya M. Ghari and Yanning Shen. Personalized online federated learning with multiple kernels. NeurIPS, 2022.
>
> [2] Songnam Hong and Jeongmin Chae. Communication-efficient randomized algorithm for multi-kernel online federated learning. IEEE Transactions on Pattern Analysis and Machine Intelligence, 2022.

---

### Author Rebuttal · Authors · 2024-08-07

We thank the reviewers for taking the time to review our paper and offering valuable comments and suggestions.

In this paper, we address the fundamental problem of whether collaboration is necessary for online model selection with decentralized data (OMS-DecD). All reviewers acknowledged our theoretical contributions, which include the conditions under which collaboration is necessary for OMS-DecD and a clarification of the unnecessary nature of collaboration in previous federated algorithms for distributed online multi-kernel learning, and technical contributions, consisting of improved Bernstein's inequality for martingales, a federated online mirror descent framework, and decoupling model selection and prediction, all of which might be of independent interest.

The reviewers also rose some valuable questions. We have provided detailed answers to all of the questions. The reviewers also provided constructive suggestions that will make the presentation more clear and make our paper easier to follow. We will revise our paper following the suggestions.

---

### Decision · Program_Chairs · 2024-09-25

**Decision:**

Accept (poster)

**Comment:**

This paper took a nice framework, federated learning, and provided some nice results around the problem of collaborating on model selection in a decentralized setting. The reviewers all agreed this was an interesting problem, had solid results, and deserved to be published. I would strongly recommend the authors take into account the critical feedback when preparing the paper for the camera-ready version.